# History of the Terminal Cataclysm Paradigm: Epistemology of a Planetary Bombardment That Never (?) Happened

**William K. Hartmann**

Planetary Science Institute, 1700 East Fort Lowell Road, Suite 106, Tucson, AZ 85719, USA; hartmann@psi.edu

**Abstract:** This study examines the history of the paradigm concerning a lunar (or solar-system-wide) terminal cataclysm (also called "Late Heavy Bombardment" or LHB), a putative, brief spike in impacts at ~3.9 Ga ago, preceded by low impact rates. We examine origin of the ideas, why they were accepted, and why the ideas are currently being seriously revised, if not abandoned. The paper is divided into the following sections:

1. Overview of paradigm.
2. Pre-Apollo views (1949–1969).
3. Initial suggestions of cataclysm (ca. 1974).
4. Ironies.
5. Alternative suggestions, megaregolith evolution (1970s).
6. Impact melt rocks "establish" cataclysm (1990).
7. Imbrium redux (ca. 1998).
8. Impact melt clasts (early 2000s).
9. Dating of front-side lunar basins?
10. Dynamical models "explain" the cataclysm (c. 2000s).
11. Asteroids as a test case.
12. Impact melts predating 4.0 Ga ago (ca. 2008–present.).
13. Biological issues.
14. Growing doubts (ca. 1994–2014).
15. Evolving Dynamical Models (ca. 2001–present).
16. Connections to lunar origin.
17. Dismantling the paradigm (2015–2018).
18. "Megaregolith Evolution Model" for explaining the data.
19. Conclusions and new directions for future work.

The author hopes that this open-access discussion may prove useful for classroom discussions of how science moves forward through self-correction of hypotheses.

**Keywords:** terminal cataclysm; late heavy bombardment; LHB; lunar impact basin formation; regolith; megaregolith; paradigms in planetary science; cratering; lunar impact basin ages; lunar origin

---

## 1. Overview and Origin of Terms

In the field of planetary science, a "classic" paradigm (Kuhn, 1962 [1]) of terminal cataclysm was initiated in 1973–1974. This consisted of a late, heavy, inner-solar-system bombardment, around 3.9 Ga ago, creating many of the lunar multi-ring impact basins within intervals of ~150–200 Ma, and with

minimal cratering prior to ~4.0 Ga ago. Note that in this paper, Ma and Ga are used for $10^6$ years and $10^9$ years, respectively, except where original language is used for time designation quotes from earlier papers. (In addition, statements in quotes that include earlier stylistic forms are left in their original wording). The paradigm was based on interpretations of lunar rock samples returned from the Apollo and Luna landings. After a history of mostly wide acceptance for four decades (with sporadic, relatively ineffectual critiques), contradictory evidence became apparent around 2000. The existence of such a cataclysm began to be seriously questioned circa 2015 at certain topical conferences and in review papers. At the same time, however, the paradigm (along with subtly shifting versions of it) is still being invoked under the original names ("terminal cataclysm," or "late heavy bombardment") among other scientific communities including biologists, martian and terrestrial geologists, and dynamicist's modeling of planetary system formation. Kuhn (1962 [1]) famously discussed the importance to scientific history of what he called "paradigm shifts," noting that jumps in progress are often associated with such collapses of accepted hypotheses. Given the wide application of the terminal cataclysm paradigm, from terrestrial geologic and biologic evolution to Solar System dynamicists, its 45-year rise and fall warrants study from Kuhn's point of view.

As described in Section 2, the idea of a very early period of intense bombardment of the Moon in its first few hundred Ma can be traced back at least to pre-Apollo lunar studies in the mid-1960s. The more specific concept of a brief "terminal cataclysm" dates from circa 1973–1974 (Tera et al. 1973 [2]; 1974 [3,4]; Turner and Cadogan 1973 [5]; Turner et al. 1973 [6]), as described in Section 3. The term Aterminal cataclysm@ appears to have been first used in the titles of the three Tera et al. papers. An additional phrase, "late heavy bombardment," appears to have been first used by Wetherill (1975 [7]) in the title of his paper on asteroid bombardment, and used again in the text both of Wetherill's follow-up papers (1977a, b) [8,9]). This second term (often abbreviated now as "LHB") has been widely adopted in later years, and in the period roughly 1975 to 2000, was often used interchangeably with "terminal cataclysm" (Turner and Cadogan (1975 [10]). However, as we will see in Section 15, some writers began to use "LHB" to refer merely to a more extended tail end of a long decline in impact cratering, a practice that conflicted with the original definition of a short-lived cataclysm and brief (~150 Ma) spasm of basin formation. For this reason, it is important to review exactly what Wetherill meant by "late heavy bombardment." Wetherill (1975 [7] p. 866), responding to the abovementioned papers, discussed the possibility of "storage" places, " ... in which large bodies [he later mentions 200-km diameters] can be 'stored' for hundreds of millions [of years]" prior to their impacting the Moon or planets. An unnamed reviewer of this paper disputed that Wetherill ever meant his term "late heavy bombardment" to refer to a spike in impacts. Admittedly, there is some ambiguity in his discussion, because, to clarify his intent, Wetherill (1975 [7] p. 866) goes on to explain that "The concept of 'storage places' has been used in two senses. In the first sense the bodies are thought of as constrained from impact for a long period of time after which they become available." This is the option that he discusses in the most detail, and it clearly leads to a late spike. The "second sense" of the storage concept allowed for continuous attrition of the stored objects, giving a ~600 Ma continuously declining impact rate with no particular late, cataclysmic spike.

Wetherill (1977a [8] p. 10) described two "storage" (Wetherill's word) places that included "storage" where "bodies can be preserved for several hundred million years and still have a high probability of returning to Earth orbit." Among his examples of this "first sense" of his term "storage places," he lists "evolution of Mars-crossing asteroids ... with initial perihelia near Mars and aphelion far into the asteroid belt ... In the case of an initial swarm of such bodies in nearly identical orbits, no members of the swarm will become Earth-crossing until after a fairly distinct time interval of 500–1500 m.y. has elapsed." This "first sense" example by Wetherill clearly relates to the Turner and Cadogan (1973 [5]) and Turner et al. (1973 [6]) suggestion of "probably six" major lunar basins forming at 3.88 to 4.05 Ga ago and the second of the Tera et al. (1973 [2] p. 725) two choices: "Either the Imbrium blanket has dominated all the materials so far or we must conclude that the major impacts peaked in a relatively short period near 4.0 (Ga ago)." Wetherill's (1977b, [9] p. 1) follow-up paper

refers in its abstract to "the lunar heavy bombardment 4 b.y. ago" and relates it to the work of Tera et al. (1974 [3,4]).

The problem here is that the present epistemological paper attempts to study not only what researchers stated in a given paper, but what was perceived by the outside community as the operative meaning—in this case the meaning of "late heavy bombardment" or "LHB." My assertion is that most post-1975 lunar researchers focused on the Tera et al. (1973) [2] suggestion of several multi-ring basins forming in a cataclysmic bombardment at ~3.9 Ga ago (rather than the Tera et al. (1973 [2] suggestion that the 3.9 Ga-old Imbrium impact alone distributed material over much of the Apollo collecting area). Additional influences in that direction (lesser-noticed in the USA?) came from the English lab of Turner and Cadogan (1973 [5]) and Turner et al. (1973 [6]) who concluded that "at least three and probably six" of the front-side multi-ring basins were formed in a 170-Ma interval ~3.9 Ga ago. In the same way, I suggest that most later readers of Wetherill (1975 [7]; 1977, [8]) focused on the "first sense" of his storage mechanism, which allowed a delayed "terminal," "cataclysmic" spike in cratering (especially multi-ring impact basin formation) at about 3.9 Ga ago, rather than his "second sense" of the storage mechanism, which allowed a continual, declining impact flux (with modest spikes and surges?).

These terms, "terminal cataclysm" and "late heavy bombardment" ("LHB"), were soon accepted in the various scientific communities mentioned above. The most common interpretation of both terms was a unique, basin-forming bombardment cataclysm at 3.9 Ga. Ryder (1990 [11]) solidified this view with a study of Apollo impact melt rocks that showed a strong spike at ~3.9 Ga, with very few impact melts before that, which Ryder interpreted as indicating very few, if any, major impacts before ~4.0 Ga ago. The post-Wetherill perceived equation of LHB with a bombardment spike is explicitly clear in certain later papers, as we will discuss in more detail in Section 15. For example, Levison et al. (2001 [12]) defined LHB as the period "roughly 4.0 to 3.8 Gyr ago" when "the lunar basins with known dates were formed." In addition, Bottke et al. (2010, [13]) stated, "The Late Heavy Bombardment (LHB) is defined as a period 3.96–3.75 Ga when many lunar basins (e.g., Serenitatis, Imbrium) and impact melts were produced . . . "

"Terminal cataclysm", thus, became the dominant paradigm in discussing the first 600–700 Ma of the Earth, Moon, and inner Solar System impact cratering environment. Here, we will use the term "classic" to refer to the long-accepted version of lunar (or solar-system-wide) evolution, involving ~400–500 Ma of relatively low-impact cratering, followed by ~170 Ma-long "late heavy bombardment" that included formation of many of the large lunar basins, around 3.9 Ga ago, followed by a declining rate ever since.

A number of criticisms of the paradigm appeared along the way, starting in the 1970s (Section 5), but with little effect until recent years. Especially, since about 2015, the number of critiques has increased, along with contrary empirical evidence (Sections 6–12 and Section 14). As noted in Section 17, at two significant conferences in 2015, a number of current researchers agreed that the "classic" concept of a cataclysmic terminal bombardment episode is now "off the table," a phrase accepted by some (probably most) attendees, including dynamicists, at both conferences. As we will see also in Section 17, several review articles in more recent years have maintained the cataclysm/LHB terminology, but only by changing their definitions beyond recognition.

Given that the four-decade acceptance of the "classic terminal cataclysm" paradigm is declining, if not collapsing, it is disturbing to find that papers from biological, climatic, and terrestrial geological communities, not to mention textbooks, popular-level articles, and press releases continue to invoke the putative cataclysm as a constraint on their interpretations and models of life's origin and geologic/climatic evolution. Thus, analysis focusing on the epistemology and history of the terminal cataclysm paradigm seems timely, if not overdue. How did the cataclysm ideas arise? Why were they so widely accepted? When did contrary ideas arise? Why did they have little effect?

Since we are investigating the origin and history of the idea, this paper cites abstracts in addition to major papers in order to demonstrate more clearly the evolution of the idea. In the interest of the historical record and given that the author was an eyewitness and participant in the unfolding of these

events from pre-Apollo studies of lunar/terrestrial impact history onward, this paper includes some anecdotal descriptions of certain illustrative events. In addition, in contrast to listing only the first author of papers with >2 authors, the present paper occasionally lists all additional authors when it is helpful for the reader to track what notable researchers were working together.

To summarize, this study, unlike some current review articles on the topic, aims not just at listing or debating current scientific data, but at investigating larger issues regarding the practice of scientific research itself, the effects of our accumulating new data, and the nature of paradigms. As science gels into paradigms, human culture, by definition, begins to influence the paradigm along with the science. As we will see, as a paradigm becomes influential, new research begins to become influential not only for the quality of the work, but also because it fits within the paradigm.

## 2. Pre-Apollo/Luna Evidence for "Intense Early Bombardment" of the Moon, 1949–1969

Ralph Baldwin (1949 [14]), holder of a Ph.D. in astronomy, and later, President of the Oliver Machinery Company, utilized World War II bomb crater statistics and telescopic lunar crater measurements to analyze the formation process of lunar craters. His 1949 book demonstrated that lunar craters were probably explosion craters formed by meteorite impact, not volcanic features, as many were still arguing in the 1940s and into the 1950s. In 1961, when the present author arrived in Gerard Kuiper's new "Lunar and Planetary Lab" at the University of Arizona, the volcanic theory was still being argued (e.g., Fielder 1961 [15]). However, populations of ancient impact craters were being discovered in the Canadian Shield and elsewhere (Beals et al., 1963 [16]). These not only established impact cratering as a geologic process but allowed crude estimates of the current rate of crater formation in the Earth–Moon system. Those data, in turn, led to a pre-Apollo estimate that the lunar mare lava plains dated, on average, from ~3.6 Ga ago (Hartmann 1965 [17]), consistent with dates of lunar mare samples returned by Apollo astronauts.

It was then noted that the pre-mare lunar highland surfaces have ~32 times the crater density of the lava plains. Applying the estimated ~3.6 Ga mare average age, it was argued that with 32 times as many craters forming in roughly 1/5 the subsequent time, the pre-mare cratering rate (the first 700–800 Ma) had to be at least ~160 times higher than the more recent average cratering rate (Hartmann 1966 [18]). Taken from that paper, Figure 1 shows a schematic sketch of the cratering rate versus time, relative to the mare-forming period. The sketch was published without numerical scales on either axis, but illustrates the much higher pre-mare cratering rate due to the declining flux of asteroid and cometary bodies, and perhaps due to the presence of some other planetesimal source as well. That paper referred to this higher average rate (by two orders of magnitude) as "Early Intense Bombardment" or "EIB."

This logic is still essentially correct. The estimate of the pre-mare average cratering rate was a lower limit, because the lunar highland areas are mostly saturated with craters. Thus, in the densest-cratered highland, early craters could be obliterated by later ones, to the point of establishing a "saturation equilibrium" size–frequency distribution (SFD), consistent with densest SFDs being found on other planets and satellites (Hartmann 1984 [19]). Here, we note that some high-albedo areas, typically classified as "highlands," include lower crater-density lava patches covered by a thin veneer of more recent bright crater ejecta (see Hartmann and Wood, 1971 [20], and further discussion in Section 5). On average, however, many more impacts/km$^2$ could have occurred than we can see in the "saturation equilibrium" parts of the highlands, in which case, the cratering rates before ~3.9 Ga ago could have been many times higher than what we can measure directly. These data left unresolved the time dependence of the cratering during the first 800 Ma or so. That time dependence is the major issue in all discussions of "terminal cataclysm" or "LHB."

Baldwin (1969 [21]), for example, continued to analyze crater populations and argued that the lunar uplands had developed a "steady state" of crater production versus destruction (saturation), leading to the prescient statement in his 1969 abstract, referring to crater populations and topography:

Even the largest craters do not last indefinitely. Probably no portion of the Moon dates back as far as 4.5 billion years.

This statement was soon to be supported by analyses of the first samples "brought home" from the Moon.

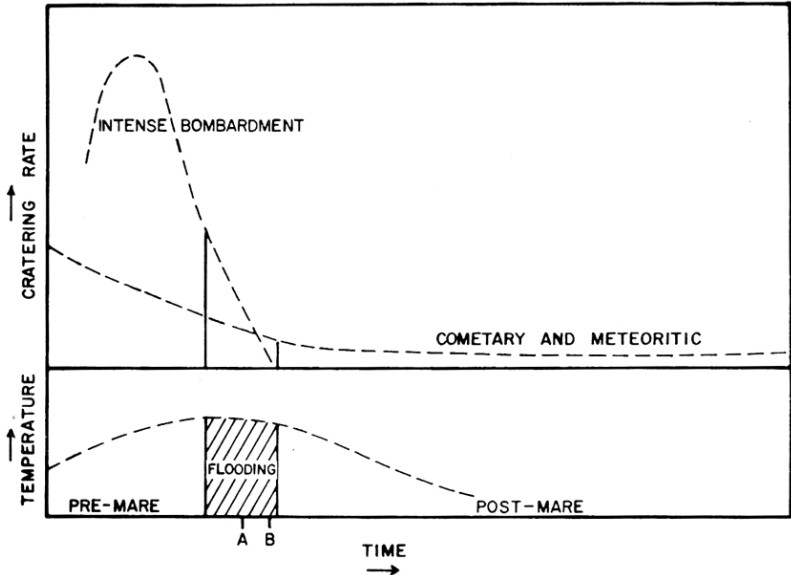

**Figure 1.** A 1966 visualization of lunar bombardment history, from a 1966 paper regarding "early intense bombardment." The curve is based on a 1965 estimate of lunar maria having ~3.6 Ga age, and pre-mare uplands having ≳32 times the crater density as the maria. This required an average cratering rate, during pre-mare time, of ≳160 times the average post-mare cratering rate. The time-dependent structure of this early intense bombardment ("EIB") remains unknown, even today. Original caption noted: "Two maria of nearly the same age, formed at A and B, could have appreciably different crater densities because of the decline in cratering." (Hartmann 1966, his Figure 5 [18]).

## 3. The Search for Lunar Genesis Rocks and Initial Suggestions of "Terminal Cataclysm," circa 1974

During the 1960s and planning for Apollo flights and lunar sampling, researchers widely assumed, almost subliminally, in spite of Baldwin's work, that because the Moon lacked Earth's erosion/deposition processes, rocks lying on the lunar surface would offer a sampling of the entire history of the Solar System. Astronauts' collections of such rocks would give us a direct record of Earth/Moon history since 4.5 Ga ago. The cumulative destructive (as well as creative) effects of cratering were, thus, grossly underestimated, since cratering was an unfamiliar process to most terrestrially trained geologists, with the notable exception of Shoemaker (e.g., 1962 [22]).

An anecdote is instructive here. During the author's 22 April 2009 interview with Johannes Geiss (first director of the International Space Science Institute, radiometric dating expert, and student of Harold Urey, who in turn was an advisor during the planning of the Apollo missions), Professor Geiss spoke about how Urey helped sell the lunar Apollo program by saying (as paraphrased by Geiss) "give me one rock from the Moon and I can give you the history of the Solar System." As stated in Urey's 1952 [23] book, "The Planets," Urey thought the Moon would likely be a vast reservoir of primitive, chondritic material.

As a result of this mindset, analysts of the earliest Apollo samples were astonished by the paucity of rocks older than 4.0 Ga. Apollo astronauts in later missions were, thus, trained in petrologic properties that might allow them to identify and retrieve the long-sought primordial rocks. Voice tapes during an Apollo 15 excursion in 1971 recorded the two astronauts reacting strongly to the discovery of what turned out to be a pristine, unbrecciated anorthosite rock, sample 15415:

"Guess what we just found! Guess what we just found!"

"I think we found what we came for."

During a press conference held during the return flight, an inquiry relayed to the astronauts was, "Near Spur Crater, you found what may be a 'Genesis Rock,' the oldest yet collected on the Moon. Tell us more about it." (Quotes from NASA Johnson Space Center Curatorial Document [24]).

The term "Genesis Rock" spread rapidly in the media and among scientists, as the Apollo and Russian Luna sample investigators searched for additional primordial specimens in the lunar sample collection. Researchers, thus, continued to express surprise as radiometric dating of the Apollo sample collections revealed relatively few rock specimens dating from before ~3.9 Ga ago. As a result, two lunar sample research groups simultaneously developed the hypothesis that some sort of cataclysm occurred on the Moon (or in the Earth–Moon system) around 3.9 Ga ago.

The Wasserburg/Tera/Papanastassiou group at Caltech, prominent in radiometric dating of the Apollo samples, developed the idea of a global metamorphic event at 3.9 Ga ago, referring to it in their titles as a "terminal cataclysm" (Tera et al. 1973 [2]; 1974 [3,4]). These three publications record the evolution of their early reasoning, starting with Tera et al. (1973 [2]), who discussed their Rb–Sr and U–Th–Pb analyses of rocks from Apollos 12, 14, 15, and 16 samples. Using the then-current AE abbreviation for "aeon," referring to $10^9$ years, their second sentence states:

*"These data point to a period of extensive melting and metamorphism at ~3.95 AE (possibly produced by the Imbrium event at 3.95–4.00 AE?)"*

Their penultimate sentences conclude:

*"Contamination of rocks by the dispersed materials in the complex plasma produced by the Imbrium impact have altered some of the isotopic characteristics . . . Since Imbrium was a relatively young basin forming event, this type of process must have been even more prevalent prior to 4.0 AE. Either the Imbrium [ejecta] blanket has dominated all the materials so far or we must conclude that the major impacts peaked in a relatively short period near 4.0 AE."*

A stronger, simultaneous suggestion along the same lines was made in 1973 by a second group in England (Turner and Cadogan 1973 [5]):

The $^{40}$Ar–$^{39}$Ar ages of samples from the lunar highlands clustered in the interval 3.88 to 4.05 aeons and were taken to indicate that at least three and probably six of the major lunar basins were formed by impact in this period. Whether these impacts represent the final stages of a continuous accretion process for the Moon or whether they represent an episodic burst of bombardment is not yet clear.

Turner et al. (1973 [6]) repeated this conclusion and discussed the "severe problems" in understanding the formation of the Serenitatis, Nectaris, Humorum, Crisium, Imbrium, and Orientale basins in an interval as short as 170 Ma, some 600 Ma after the planets formed.

During the next three Apollo flights, mission planners focused on the effects of the giant Imbrium impact. Apollo 14 landed in 1971 at Fra Mauro, on Imbrium ejecta striated radially from the Imbrium center 1230 km away. The landing site was 550 km from the southern Carpathian Mountain rim of the Imbrium basin (Swann et al. 1977 [25]). Later in 1971, Apollo 15 collected Imbrium samples with similar chemical and age signatures inside the Imbrium basin near the southeastern rim-ring formed by the Apennine Mountains. Apollo 16 landed in 1972, intending to investigate upland plains thought to be volcanic. As understood today, that site was at the outer edge of southeast ejecta striated radially from Imbrium, near the Nectaris Basin, thus yielding additional, plausibly Imbrium samples. Taken together, the sample collections from various sites showed that the Imbrium basin formed at a then-estimated 3.7 to 3.9 Ga ago, but a more precise date of the Imbrium basin was long debated. For example, a 2001 review by Stöffler and Ryder [26] noted that "an age of 3.85–3.90 had been generally accepted before 1980," but they themselves presented two possible ages, 3.77 ± 0.02 and 3.85 ± 0.02.

Tera et al. (1974 [3,4]) moved perhaps a bit further away from their 1973 Imbrium-specific interpretation toward the more global concept, "terminal cataclysm." It is useful to note that they used the term "cataclysm" in the titles of all three papers, and "terminal cataclysm" in the titles of both 1974 papers. The U–Pb and Rb–Sr data from several landing sites revealed formation of a primordial lunar

crust at around 4.4–4.5 AE, and in the 1974 paper, Tera et al. [3] stated that "the Pb–U fractionation was essentially due to the Pb volatilization during the metamorphic events." They associated this "metamorphism" with bombardment at "~3.9 AE." In all three papers, however, they were careful to suggest that the cataclysmic event could be either the formation of Imbrium, which could have spread debris over much of the front side, or a clustering of impacts around 3.9 Ga ago. (Interestingly, none of their three papers refer to the Turner et al. [6] (1973) conclusion that three to six basins were formed between 3.88 and 4.05 Ga ago).

The introductory section of the Tera et al. [3] (1974a) paper describes their argument:

" ... *we shall present evidence for wide-spread shock metamorphism and element redistribution [at] approximately 3.9 AE which resulted from large-scale impacts on an ancient lunar crust ... both the predominance of ~3.9 AE ages and the Pb systematics are the result of extremely heavy, possibly sporadic, bombardment of the Moon from its formation until ~3.9 AE. The intensity of the bombardment was extremely high at ~3.9 and was sharply reduced to a relatively low level after 3.9 AE ... we have now accumulated sufficient independent evidence to propose a ~3.9 AE "cataclysm" on an observational basis.*"

The abstract of the Tera et al. [3] (1974) paper clarifies their view that:

"*This cataclysm is associated with the Imbrium impact and very possibly the formation of Crisium and Orientale and possibly several other major basins in a narrow time interval (~2 × 10$^8$ yr or less).*"

In their 1974 LPSC abstract, Tera et al. (1974 [4]), state that:

"*We interpret this Pb–U fractionation to be largely the result of major impacts ... This cataclysmic event or cluster of events, which may have occurred over a 0.1 AE interval, is most reasonably associated with the formation of major lunar basins ... *".

Again, they are careful to allow the observations to refer either to the singular Imbrium event or a cluster of events within about 100 Ma, but their sentence construction begins to lean more toward "events" instead of one "event."

By the end of the Apollo missions in 1974 and even the Russian Luna sample collection missions in 1976, the results were often discussed as if they seemed clear to most researchers: the Apollo and Luna samples revealed not simply a widespread geochemical metamorphic event, or effects of the Imbrium impact, but rather a climactic "late" bombardment that formed many of the largest impact basins in <200 Ma. Some researchers remained cautious, however. Turner (1977 [27]), for example, correctly noted that, "What is not yet clear is whether this 'cataclysm' represents simply the effective termination of an approximately monotonic decrease in the early cratering or ... a period of increased influx to close to 4.0 Ga."

## 4. Ironies

Another personal anecdote is perhaps amusing and instructive about subliminal concepts when the Apollo program was underway. While I referred in 1966 to an intense average cratering rate before ~3.6 Ga ago as "*early* intense cratering," by 1973 the concept of intense cratering at 3.9 Ga ago came to be called "*terminal* cataclysm" or "*late* heavy bombardment." As a graduate student in 1966, I was thinking of pre-mare bombardment in the 4.5 to 3.5 Ga interval as "early," relative to the history of the Earth–Moon system. To the researchers accustomed to measuring ages of meteorites and planetary formation, however, the putative 3.9 Ga event was late, compared to the era of planet formation. Should we call it "early" or "late?" Paradoxically, both of the mutually exclusive semantic constructs were correct. It was "early" in the sense of lunar history, but to researchers focusing on planet formation it seemed strangely late. Hence, we are made aware of the mischievous role of words, our tools in creating and discussing scientific paradigms.

In another irony, the Tera et al. group, usually credited with linking the name "terminal cataclysm" to a burst of basin-forming impacts, initially seemed to lean toward their other hypothesis, that the Imbrium impact was the main source of the 3.9 Ga signature in the samples. Thus, the current trend of associating Apollo/Luna front-side sample ages with Imbrium ejecta takes us back toward the repeated early suggestions by Tera et al. (1973 [2]; 1974 [3,4]), that the Imbrium might be involved in spreading altered material over much of the front-side Apollo/Luna-sampled area—a suggestion that most of the planetary science community gradually abandoned.

Nonetheless, as early as 1977, Schaeffer and Schaeffer [28] attempted to return the discussion of lunar sample ages to the 1973–1974 ideas of Tera et al. [2–4], that debris from Imbrium might be responsible for the clustering of lunar sample ages around 3.9 Ga. They (1977 [28] abstract) wrote that "Histograms for the $^{39}$Ar–$^{40}$Ar plateau ages strongly suggest that the peak in ages at Apollo 14, 16, and 17 of about 4 billion years is caused by the dominance of Imbrium ejecta at these landing sites."

Another illustration of an irony, or perhaps we should say the seriousness of the current problem, comes in a 2018 issue of *Icarus*, where Strom, Marchi, and Malhotra (2018 [29] p. 107) interpret the crater populations on the largest asteroid, Ceres, from the Dawn orbiter imagery, in terms of "late heavy bombardment" (which, as we have seen in Section 1, had already been defined as equivalent to the terminal cataclysm of Tera et al. (1974 [3,4]). Strom et al. (2018 [29] p. 107) write:

> *"Since Ceres is in the middle of the main asteroid belt it must have been impacted by Main Belt asteroids both during and after the Late Heavy Bombardment ... The Ceres cratering record is consistent with an impact history that includes the period of the Late Heavy Bombardment ... caused by projectiles originating from Main Belt asteroids."*

The irony and indication of our problem is that this paper appeared on the page following a paper in the same issue whose final paragraph states that " ... our general conclusion is that the terminal cataclysm proposed by Tera et al. (1973, 1974) [2–4] ... did not occur" (Michael, Basilevsky, and Neukum 2018, [30] p. 101). All the researchers involved are highly respected, but some difference in view of the LHB is evident.

## 5. Alternative Suggestions from the Cratering Record, and the Role of Megaregolith: 1970s–1980s

As soon as the certain lunar mare surfaces and upland sites were assigned dates based on Apollo and Russian Luna samples, measurements of crater densities at those sites, versus reported ages from those sites, shown in Figure 2, revealed that the lunar cratering rate at ~3.8 Ga ago was ~100–200 times higher than at present. These data showed that the cratering rate was declining rapidly after that, with a gradually lengthening half-life of the cratering rate (i.e., decline in the rate by a factor two) estimated to be of the order 50–150 Ma to roughly the present cratering rate by ~3.0–2.5 Ga ago (Hartmann 1970a [31]; 1970b [32]; 1972 [33]). This fits with the 1966 evidence, mentioned in Section 2, that the average lunar cratering rate before ~3.6 Ga ago (i.e., from lunar crust formation to ~3.6 Ga) must have been at least 160 times the current rate. Schematic diagrams, in the 1970a [31] paper (also in Hartmann 1980 [34] and Hartmann et al. 2000 [35]) pointed out that the decline in the cratering rate might not have been smooth at all sizes, and probably included a saw-tooth structure, associated with asteroid collisional breakup events (cf. Section 13). Such events could have produced episodic bursts of cratering below certain sizes (controlled by the size of the largest fragment that reaches the Earth–Moon system from a particular fragmentation event). The technique of measuring the flux versus time curve was extended more carefully by Neukum (1983 [36]), and by Neukum and Ivanov (1995, [37]), as discussed also by Neukum et al. (2001 [38]). This work has been further extended by Quantin et al. (2004 [39]; 2007 [40]), Robbins (2013 [41]), and others. While the flux versus time curve fitted by Neukum (1983, [36]) shows a virtually constant impact rate after ~3.0 Ga ago, the curve proposed by Quantin et al. (2007 [40], and a companion paper by Hartmann et al. 2007 [42]) suggests a continuing decline after 3.0 Ga by a factor ~3. Such a continuous declining trend renders common

mentions of "the end of the late heavy bombardment" entirely ambiguous, depending on the meanings assigned to "late" and "heavy."

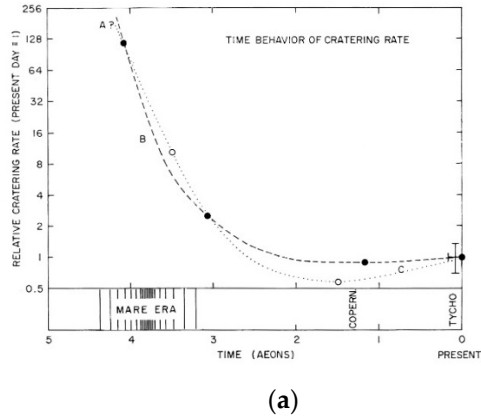
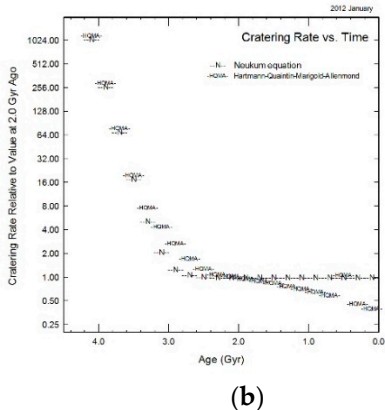

(**a**) (**b**)

**Figure 2.** (**a**) A 1970 diagram of the decline in the impact cratering rate, relative to the present-day cratering rate, based on data from the earliest Apollo landing sites. Two different interpretations are shown, but both agree on a decay by roughly 6 half-lives between 4.0 and 3.0 Ga ago (Hartmann 1970b [32]). (**b**) Two post-Apollo solutions for cratering rate as a function of time, by Neukum et al. (2001, [38]) and Hartmann et al. (2007 [42]).

The time dependence of cratering rates prior to ~3.9 to ~4.0 Ga ago could not be directly determined by these techniques because so many craters have accumulated in the lunar highlands that saturation equilibrium effects occur in most highland areas. In this situation, crater densities fluctuate by a factor ~2 around a size distribution equation (Hartmann 1984 [19]). This prevents a clear deciphering of the cratering rate prior to ~4.0 Ga ago (Hartmann 1984 [19], Hartmann and Gaskell 1997 [43]). What we can say, given the nature of the lunar crater size distribution and the measured very high early impact rate, is that pre-mare impact fragmentation and pulverization of initial crustal rock units could have reached depths of kilometers. As early as 1971, Nash et al. [44] pointed out this effect in a paper describing issues to be investigated with the Apollo/Luna samples:

*"Beneath the mare fill and in the highlands we might expect a "mega-regolith" perhaps kilometers in thickness, created by the final stages of the lunar accretion flux. This large regolith may be compatible with the essentially saturated distribution of 10–100 km craters on the highlands and far-side . . . "*

Nash et al. (1971 [44]) apparently first published the important term "mega-regolith" (now usually "megaregolith") for the putative deep pulverized layer, far deeper than the 10–20 m dusty regolith layer that the early astronauts had measured on the surface. Hartmann (1973 [45]) suggested that:

*" . . . mega-regolith of depth on the order 2 km should exist in the terrae (possibly with cementing at depth). The column between the mare surface and basin-floor basements contains highly variable amounts of fragmental material depending on the formation dates of the specific basins."*

The proposed "cementing" process would explain that the fragmental material was commonly welded into the ubiquitous upland breccias.

In a little-cited, two-paragraph abstract, Jack Hartung (1974 [46]) proposed an additional effect that could explain the lunar rock age distribution in a way contrary to the terminal cataclysm model. He pointed out that if lunar rocks were subjected to an early, declining destruction process (such as early intense bombardment), then a peak can easily be created in the age signature of those rocks. This occurs around the time that the destruction process eventually drops to a level where production rate gets the upper hand. Twenty or forty years ahead of his time, he succinctly wrote:

> *"The 4 b.y. peak in the Ar 39/40 age distribution for lunar highland rocks may be caused by: 1. the Imbrium event; 2. a sharp decrease in the rate of formation of craters. This latter possibility calls for a cratering rate so high before 4 b.y. that few rock ages survive and a rate so low after 4 b.y. that few rock ages are produced."*

A crude bell curve in age distribution was then produced around some critical age bin with the maximum number of surviving specimens. Hartung estimated the half-life associated with the declining cratering rate at ~4 Ga ago as ~70 Ma.

Hartung's idea may be a key to a post-cataclysm-paradigm understanding of the lunar sample ages. Interestingly, however, Hartung apparently wrote no more on the topic, in spite of his later work on lunar samples and other planetary geology topics. While the lack of positive reception of his abstract might be a factor, we note that Koeberl and Anderson (2015 [47]) in an obituary on Jack Hartung (1933–2015) pointed out that he had a reputation as a "Jack of all trades", and worked fruitfully on a wide variety of problems at NASA/Houston and at various European labs, without being pinned down to one topic.

Baldwin (1974 [48]), in a paper titled "Was there a 'Terminal Lunar Cataclysm' 3.9–4.0 × $10^9$ Years Ago?" examined different morphologies or preservation states of craters and basins as related to impact flux and crustal viscosity. He concluded that the impact flux had been declining during early lunar history with a half-life of 88 Ma, and that:

> *"There was no major series of events which produced the "terminal lunar cataclysm" approximately 3.95 × $10^9$ y ago. The magnitude and timing of the Imbrium collision was the single overwhelming event at that time."*

Hartmann (1975 [49]; 1980 [34]) pursued the idea of the declining impact flux as the destruction mechanism. Those papers assumed that the planetary accretion process at 4.5 Ga ago left abundant interplanetary debris, while producing a high planetesimal collision rate (as per Safronov (1972 [50])), and an extremely high initial cratering rate (≳160× present value). In that view, a high but declining cratering rate, with lengthening half-life, continued from ~4.4 Ga ago to ~4 Ga ago. This early, intense, declining bombardment fragmented near-surface lunar materials and created a megaregolith of upland breccias. Half-lives lengthened to values observed at ~3.9–3.8 Ga ago (Hartmann (1972 [33]; 1975 [49])). For example, Hartmann (1972 [33]) showed a schematic diagram with an average half-life ~80 Ma before 4.1 Ga ago and ~300 Ma from 4.1 to 3.0 Ga ago. In that view, the significance of the 3.9 Ga era was not that a cataclysmic global spike in cratering occurred, but rather that materials formed in the upper kilometers before time T years ago (as measured looking back from today) were pulverized into unmeasurable dust and breccia particles from intervals <T, whereas materials placed in the upper kilometers after T = 3.9 Ga had a good chance of surviving until today, hence a chance of being excavated by later impacts and placed on the surface recently enough to be collected. Hartmann (1975 [49]) proposed that:

> *"The scarcity of old (>4 AE) lunar rocks is here derived as a natural consequence of known paleocratering chronology. <u>Explosive mega-regolith formation</u> prior to 4 AE brecciated and heated most earlier material."*

This 1975 paper posited a "stonewall" effect, such that most impact melts that formed before ~4.0 Ga did not survive in our sample collection because of intense pulverization during megaregolith formation, and/or were reset in age, so that we could not "see" back beyond ~4.0 Ga ago. (The term "stonewall" was adopted from the contemporaneous American journalistic term for destruction of politically incriminating documents in President Nixon's White House, retarding investigation of events before a certain time.) The 1975 paper was flawed by too-facile use of the concept of "resetting" rock ages and the idea that cratering alone could reset ages.

While the Hartung (1974 [46]) abstract was not widely known, and was not cited by Hartmann (1975 [49]), it was cited and supported by Wetherill (1975 [7]), who demonstrated Hartung's mechanism

in two diagrams. The first showed a theoretical bell curve produced by (1) early destruction and (2) later impact rate decline with 70 Ma half-life (Wetherill's [7] Figure 3). The second diagram showed the empirical bell curve of then-available lunar highland sample ages (Wetherill's [7] Figure 4). As shown in our Figure 3, the two curves were remarkably similar, though Wetherill [7] wisely cautioned that the spread of radiometric ages in the latter diagram might be affected by "interlaboratory differences" and impact-caused transport of young mare basalt debris into the highlands. As we will see, Hartung's [46] mechanism has had something of a resurgence, being cited by Grinspoon (1989 [51], in a Ph.D. thesis with more sophisticated statistical and physical analysis), and by Hartmann et al. (2000 [35]) and Hartmann (2003 [52]). Boehnke and Harrison (2016 [53]), while not citing Hartung, suggested a Hartung-style "illusory" peak in Ar–Ar ages at around 3.9 Ga, in a similar mechanism based on survival versus disturbance of rock samples.

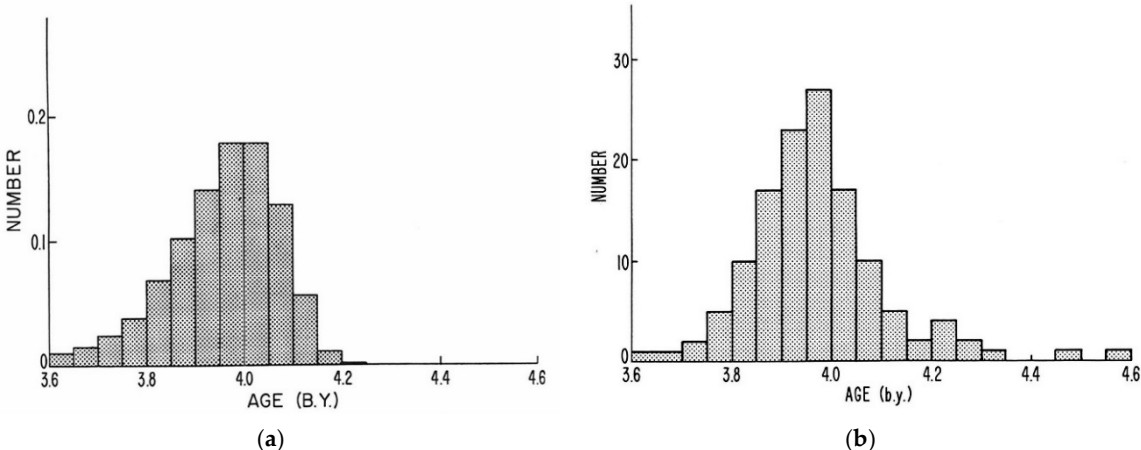

(a) (b)

**Figure 3.** Wetherill's (1975, [7]) diagrams comparing the calculated "Hartung effect" (**a**) and lunar upland rock ages (**b**). The left diagram models the competition between destruction of samples by early intense cratering and survival of samples as the cratering rate smoothly declined. The right diagram shows observed Rb–Sr and Ar–Ar age distribution from lunar highland rock samples. (Note that time proceeds from right to left, contrary to most of our diagrams).

Scattered, additional writers in that early period urged some caution about the cataclysm hypothesis. For example, Taylor (1982 [54] p. 242) noted that the cataclysm is a "non-unique interpretation of the data." Later, Taylor (1992 [55]) listed pro and con arguments about the cataclysm, but summarized that:

> " … *the record is consistent with a major spike of basin forming collision between about [4.00 and 3.85 Ga ago]. Whether impacts spread over 200 m.y. constitute a "cataclysm" may be left to experts in semantics; the term is now too thoroughly entrenched in the lunar literature to be changed.*"

## 6. Impact Melt Rock Age Statistics "Establish" a Cataclysm, circa 1990

By the early 1980s, improved criteria had been developed for distinguishing impact melt rocks (formed during impacts) from non-impact igneous rocks. These criteria emerged from both terrestrial experience and impact modeling (cf. Taylor et al. (1991 [56])). Spray (2016, [57] Figure 6) noted that among the Apollo samples of total mass ~382 kg, ~33% are breccia rocks, ~24% fines, ~23% basalt rocks, but only ~6 or 7% are impact melt rocks, and ~3% are anorthosite crustal rocks.

Dating of lunar samples showed that while few impact melts survived from before ~4.0 Ga ago, crustal rock fragments from ~4.4 Ga did survive, as shown in Figure 4. As a result, the Hartung-type impact-based models for explaining age distributions, as described in Section 5, met with little favor. They were criticized, reasonably, for not explaining the difference between survival of impact melts and early crustal igneous rocks. Why, critics asked, did sample collections show almost no impact

melts dating before 4.0 Ga, while some igneous rock fragments dated from as far back as magma ocean solidification and crust formation, around 4.4 Ga ago? (The author recalls struggling unconvincingly and incorrectly during those years with his own arguments about impact melts possibly being weaker than crustal igneous rocks).

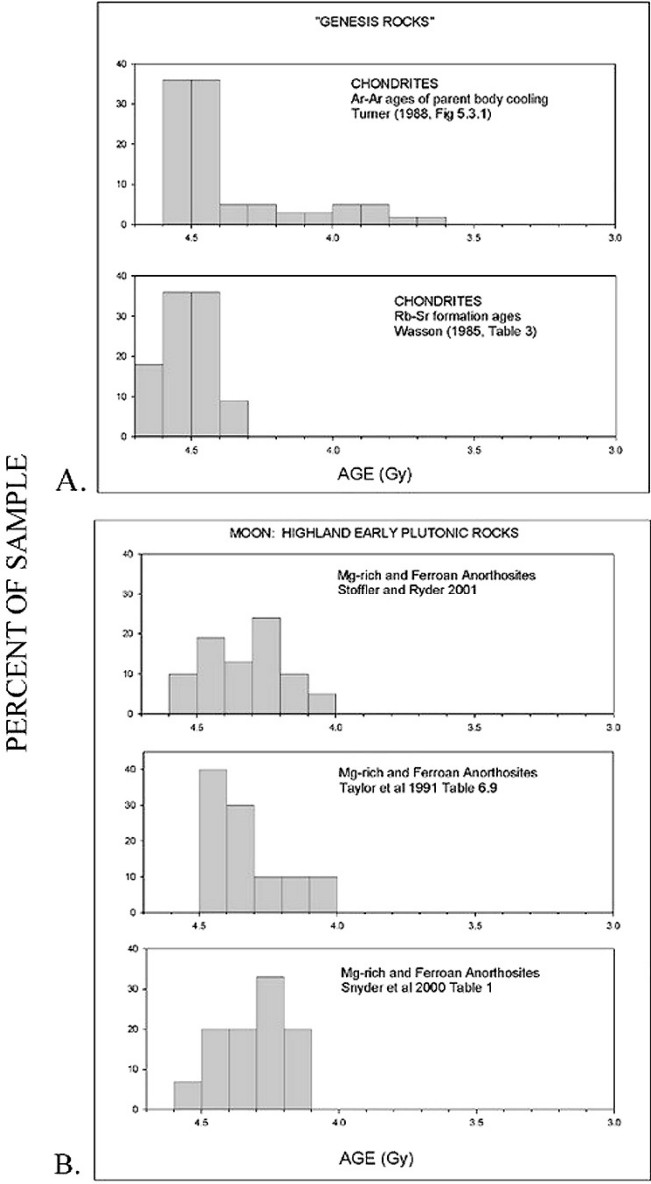

**Figure 4.** Age distribution of primordial rocks, comparing (**A**) chondrite ages with (**B**) lunar highlands crustal anorthosite rocks. Lunar crustal samples survive from the entire period of at least 4.5 to 4.0 Ga ago, with fewer younger than 4.2 Ga.

Although some impact melts are formed during impacts of modest-sized craters, numerical impact models indicate that largest volumes of lunar impact melts come from impact melt lenses in the largest multi-ring basins (Melosh, 1989 [58], Stöffler et al. 2006 [59], Artemieva and Shuvalov 2008 [60]). Small craters vastly outnumber the big basins, but the basins are believed cumulatively to produce more impact melt volume than all small craters combined. As we will see, issues have been raised, also, about the stability of radiometric ages of rocks formed at a given time, but later buried in hot debris aprons of later basins (Fernandez and Artemieva 2012 [61].

Lunar impact melt rocks were thus assumed to offer a straightforward, best available history of impact basins near the various collecting sites. Using data from various sources on relevant lunar

impact melt rocks, Graham Ryder (1990 [11]) produced a pivotal paper that seemed, once and for all, at least to many researchers, to establish the reality of a cataclysmic spike in impacts around 3.9 Ga ago. Ryder's opening summary established his main arguments:

> *"Among lunar samples there are no impact melts dated older than about 3.9 Ga, [whereas] a heavy bombardment of the Moon from its birth until 3.9 Ga should have produced many melts . . . The absence of older impact melts cannot be explained by continued isotopic resetting because ejecta are mainly cold and ancient igneous rock [still] exist. The common 3.85-Ga melt ages cannot be ascribed to a single or even a few basin events (e.g., Imbrium) because the samples show wide differences in chemistry and real, if small, differences in ages; the ages also appear in lunar meteorites and are thus Moon-wide."*

Interestingly, Ryder (1990 [11]) also argued that:

> *" . . . the separate concept of a late intense 'terminal lunar cataclysm' cogently advocated by Tera et al. (1974) has fallen into almost universal disfavor [following] arguments against it by Baldwin (1974, 1981, 1987) and Hartmann (1975)."*

Ryder's idea that the 1974 concepts of Tera et al. [2–4] had fallen into disfavor was not my perception at the time, since I perceived that the Tera et al. (and Turner and Cadogan [5]) models of a cataclysmic episode at 3.9 Ga had carried the day, and that it was the Baldwin [48] and Hartmann [45,49] papers that had been dismissed—but perhaps we scientists on both sides of the issue were overly defensive.

Ryder insisted vigorously that *absence of impact melts* before ~3.9 Ga indicates *near-absence of impacts* in the 3.9 to 4.45 Ga interval:

> *"The lunar data are consistent with only light bombardment in the first 600 m.y. and then an intense cataclysmic bombardment . . . The arguments . . . suggest that the rapid accretion of the Moon was over by 4.45 Ga at the latest, and that it . . . was fairly undisturbed by exogenic processes from ~4.45 to ~3.9 Ga ago. During this period cratering was light and did not produce a megaregolith more than several hundred meters thick . . . No impact melts from this period have been recognized. This period cannot, therefore, be interpreted as one of heavy bombardment. At ~3.9 Ga a catastrophic cratering period started, and it was over by ~3.81 Ga. During this time, not only about a dozen multi-ring basins formed, but also almost all the visible craters of pre-Orientale age."*

Interestingly, this conclusion, that the cratering rate dropped to "only light" or near-zero values between planet formation and the spike at 3.9 Ga, seemed hard to reconcile with then-favored theoretical studies of planetary accretion by workers such as Safronov (1972 [50]) and Wetherill (1975 [7]; 1977a [8]; 1977b [9]), and it dramatically conflicted with the early view of Tera et al. (1974a [3]), who stated:

> *" . . . both the predominance of ~3.9 AE ages and the Pb systematics are the result of extremely heavy, possibly sporadic, bombardment of the Moon from its formation until ~3.9 AE."*

Figure 5 shows the kinds of impact melt age data that Ryder used. It shows a huge, narrow spike in impact melt rock ages from ~3.8 to 4.0 Ga, containing most of the impact melts among the Apollo samples that were studied.

Ryder's cogent, concise arguments about impact melt ages were widely accepted as establishing the "classic" version of the terminal cataclysm paradigm, sometimes called the "strong form" of the concept. An anecdotal aside: At the 1990 Perth meeting of the Meteoritical Society, one session featured what amounted to an informal "debate" about these issues between Ryder and Hartmann. I would say that Graham, whom I knew as a friend, won.

Ryder's logic had two parts: (1) the uncontested presence of a spike at ~3.9 Ga in Apollo impact melt rock dates, and (2) the lack of impact melts dating from before 4.0 Ga ago, which he took as indicating the lack of impacts in that period. We will refer to this second premise as "Ryder's rule":

lack of observed impact melts from a given period = lack of impacts during that period. As discussed in later sections, this rule was adopted as an empirical constraint on theoretical dynamical models for the next two decades. Impact rates had to be low in the "Ryder gap," between ~4.4 Ga and 4.0 Ga.

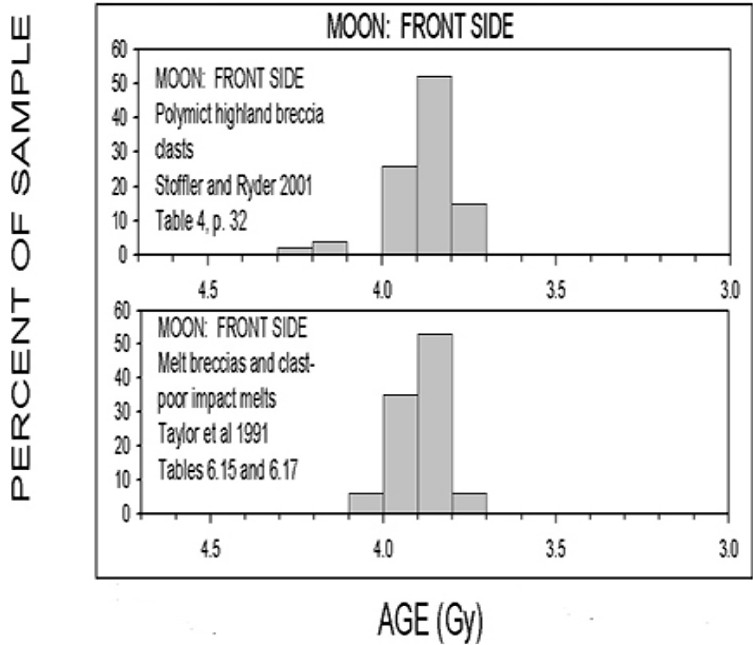

**Figure 5.** Two diagrams showing impact melt rock age statistics from Apollo samples. Ryder (1990, [11]) presented such data as proof of the existence of cataclysmic bombardment for ~150 Ma around 3.9 Ga ago. Ryder also argued that the relative lack of impact melts from before 4 Ga ago indicated lack of impacts during that time. This "Ryder gap" argument was widely used by later researchers as a base for sample interpretation and dynamical models.

## 7. Imbrium Redux, circa 1998

Haskin, Korotev, and colleagues (1998 [62]) proposed a return to what was (ironically, as noted above) the initial view of Tera, Papanastassiou, Wasserburg, and the Caltech group in 1973 [2] and 1974 (1974 [3,4]). Namely, that the spike in 3.9 Ga ages among the Apollo impact melt samples could be due to the pervasive effects of Imbrium basin formation, i.e., Imbrium ejecta emplacement in the sampling areas. For example, they wrote:

*"The origin of mafic impact melt breccias bears on many lunar problems: the nature of the late meteoroid bombardment (cataclysm); the special distribution of KREEP [materials with enhanced K, rare earth elements, and P], both near the surface and at depth; the ages of the major basins . . . Thus it is crucial that [their] origin . . . be accurately understood . . . We suggest that the narrow range of ages of 3.7–4.0 Ga for all successfully dated mafic impact-melt breccias may reflect a single event whose age is difficult to measure precisely, rather than a number of discrete impact events closely spaced in time."*

This was the antithesis of the prevailing conception of global, cataclysmic formation of many basins within a 150 Ma spasm of large impacts. The Haskin et al. (1998 [62]) conclusion was based on newly mapped geochemical signatures considered to be associated with widespread Imbrium ejecta. As shown in Figure 6, such signatures suggested that Imbrium ejecta were present over much of the near-side, Apollo-sampled region. In that view, the scatter among petrologic rock types, and among reported impact melt ages from ~3.85 to 4.00 Ga, would be interpreted at least partly as "noise" *within* the Imbrium ejecta blanket, rather than evidence of multiple separate basins. Mapping of "light plains" deposits radial to the Orientale basin shows a similar, strikingly widespread, distribution of materials associated with an Imbrium-scale impact basin (Figure 6b). To add an updated view, such "noise"

in the Imbrium ejecta could involve several phenomena: error bars of individual measurements; modest resetting of some isotopic systems such as Ar–Ar in rocks buried in warm ejecta blankets (Fernandez and Artemieva 2012 [61], Boehnke and Harrison 2016 [53]); additional entrainment of different petrologic types of igneous rocks and pre-existing impact melts from the enormous transient cavity of Imbrium, which may have tapped into various intrusions and one or more large, pre-Imbrium craters or small basins, which would not have been unlikely in an area as large as the Imbrium basin (Hartmann 2003 [52] pp. 589–590); plus excavation and additional entrainment into the Imbrium ejecta blanket of local diverse materials during Imbrium ejecta impacts at the collection sites. As an example of the last phenomenon, Nectaris basin ejecta at the Apollo 16 site could be mixed into the Imbrium ejecta, as Imbrium material rained down (Petro and Pieters 2006 [63]). Other factors in the "noise" of the sample dates could include variable dwell times at high pressures and temperatures, and even small ongoing corrections in the decay constants of various isotopic systems used by investigators in different decades. All these effects were presented and discussed at the Workshop on "The First 1 Ga of Impact Records" at the 2015 Meteoritical Society meeting in Berkeley, California. For example, regarding attempts to date possible earliest impact basins, Warren (2015 [64]) concluded that "the *true upward uncertainty* in any very ancient (4.35–4.5) lunar crystallization age is *never less than about 50 Ma*" (Warren's italics).

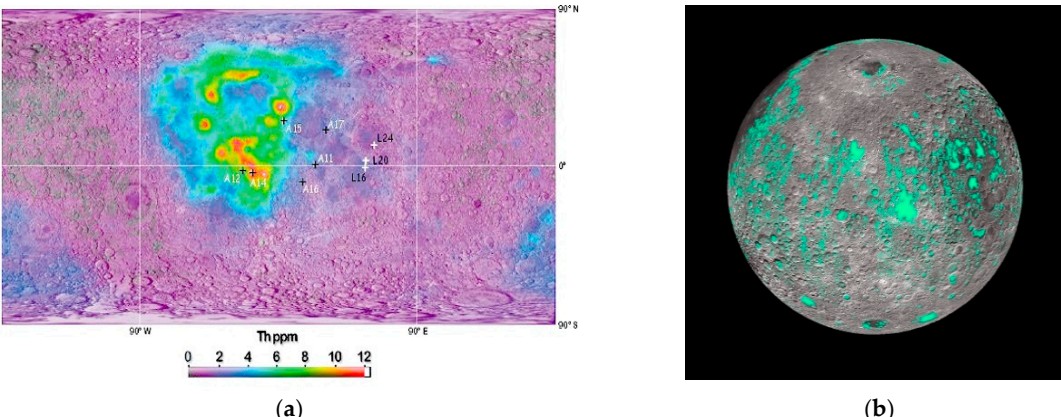

(a)  (b)

**Figure 6.** (**a**) Map showing thorium concentrated in the area of Imbrium impact basin, but also widely distributed on the front side of Moon (central half of diagram), including Apollo landing sites. (**b**) Lunar global image showing mapping of "light plains" in patterns radial to Orientale impact basin (at top) and interpreted as deposits of uncertain character associated with the formation of Orientale (Meyer et al. 2018 [65]). This image supports the very widespread distribution of surface materials associated with an Imbrium-sized basin (Courtesy Heather Meyer).

At this point, one might also ask if some large fragments of Imbrium ejecta escaped into heliocentric orbit and might have impacted at scattered sites around the Moon tens of Ma later, creating new impact melts, and possibly adding to the noise level in the impact melt data being used to date Imbrium. As a partial answer, Gladman et al. (1995 [66]) modeled the history of ejecta from lunar impacts (focusing mostly on history of lunar meteorites impacting Earth). In a sample of 10 lunar meteorites, they noted that most lunar meteorites on Earth have cosmic ray exposure (CRE) ages <1 Ma, but one is measured at 9 ± 2 Ma. Eugster (2003 [67]) plotted CRE ages of lunar meteorites of which three had ages >1 Ma, one listed at ~1.3 Ma, another at ~3 Ma, and the third at ~8 Ma. The Gladman et al. (1995 [66]) models of ejection from the Moon and later impacts in the Earth–Moon system gave similar results. Their conclusion was that most of the Imbrium ejecta was re-accreted within a few Ma onto Earth and Moon (and a minor fraction on Venus!). This would be too short a timescale to add much noise to the Imbrium impact signature on the Moon. Furthermore, most of the objects ejected from Imbrium into heliocentric orbits would be relatively small, so that their impacts would produce relatively small amounts of impact melt. Still, it is plausible from the observations and the models that some few large Imbrium

fragments might have impacted the Moon a few tens of Ma years later, creating an enhancement of impact melts scattered around the Moon around 3.9 Ga ago and "fuzzing out" the signal from the initial Imbrium impact by a few Ma or a few tens of Ma.

As a more important example of noise in the reported Imbrium impact dates, Fernandes and Artemieva (2012 [61]) discussed age resetting in the Ar–Ar isotope system as a result of burial in warm impact ejecta blankets. They argued that a number of lunar samples show "partial to total resetting of the K–Ar system [ages] even at low shock pressures," and suggest that the likely cause was burial in warm ejecta. Their models of Imbrium-scale impacts (their Figures 3 and 4 captions) indicate distant Imbrium ejecta reaching 400–600 K and depths of 200–400 m or more. This fits the fact that kilometer-scale topography at the Apollo 16 landing site near the Nectaris basin is radial to Imbrium, not Nectaris—which means that dates near 3.8–3.9 Ga at the Apollo 16 site may reflect primarily Imbrium material, not the age of the nearby, older Nectaris basin. Note that the work of Fernandes and Artemieva (2012 [61]) reverses Ryder's statement, quoted above, that ejecta blankets are "mainly cold" and cannot reset any isotopic chronometers. With a similar argument, Boehnke and Harrison (2016 [53]) discussed diffusive loss of $^{40}$Ar creating significant uncertainties in Ar–Ar ages. (See further discussion of such effects in Section 9).

As a personal recollection, my sense in the late 1990s was that the Haskin et al. (1998 [62]) proposal, the 3.9 Ga spike involved Imbrium, rather than a cataclysmic bombardment, provoked vigorous and relatively hostile debate at LPSC meetings and other venues. It seemed to be a return to the Imbrium hypothesis of Tera et al., circa 1973–1974. One counter-argument was that the impact melt ages differed by amounts outside the claimed error bars for Imbrium (so that there must have been distinct basin-forming impacts around 3.8–4.0 Ga ago, other than Imbrium). Another counter-argument was that the chemistry and petrology of the impact melt samples differed among themselves, which was interpreted as indicating separate impacts in distinct geologic regions. The argument about different petrologies carried much weight at the time, but it assumed that all ejecta from a given impact, arriving at a given site, must be petrologically identical, ignoring the above arguments that the untidy processes of transient cavity creation during the Imbrium impact probably sampled and chaotically mixed a range of plutonic and surface materials, not to mention earlier basin impact melt lenses. The Imbrium transient cavity diameter has been estimated at 700 to 850 km by Schultz and Crawford (2016 [68]).

Because the Haskin et al. (1998 [62]) work was not widely embraced, the terminal cataclysm continued to be discussed as if it were a well-documented spasm of large impactors creating most of the major lunar multi-ring basins within ~150 Ma.

## 8. Lunar Impact Melt Data from a New Source: Clasts in Lunar Meteorites, circa 2000

Starting in 2000, Cohen and various co-authors (2000 [69]; 2002 [70]; 2005 [71]) published innovative, important data aimed at clarifying the above issues. Using improved measurement techniques, they reported measurement of impact ages among small impact melt clasts contained in lunar breccia meteorites. In order to increase the chance of sampling a wider region of the Moon, including the lunar far side, Cohen et al. chose samples that avoided the geochemical "KREEP" signature, associated with samples that had been identified with Imbrium (as in Figure 7).

Their first paper (Cohen, Swindle, and Kring 2000 [69], in *Science*) was titled "Support for the Lunar Cataclysm Hypothesis from Lunar Meteorite Impact Melt Ages." It defined the lunar cataclysm in what we have labeled the "classic" terms, as:

" ... *surprising wide-spread isotopic disturbances at 3.9 × 10⁹ years ago ... attributed to ... an enormous number of asteroid and/or cometary collisions in a brief pulse of time ... This single event would have created the large basin structures and resurfaced much of the moon.*"

The Cohen et al.'s (2000 [69]) abstract states their support for:

" ... *a short, intense period of bombardment ... at ~3.9 Ga. This was an anomalous spike of impact activity on the otherwise declining impact–frequency curve?*"

Kring and Cohen (2002 [72]) followed up in the *Journal of Geophysical Research* with an overview of "Cataclysmic bombardment throughout the inner solar system, 3.9–4.0 Ga" (their title), making an explicit case that the terminal cataclysm paradigm had been confirmed. Their abstract begins,

> *"Cohen et al. [2000] recently confirmed the hypothesis that the Moon was resurfaced by an intense period of impact cratering ~3.9 Ga ago and, by inference, that the Earth also sustained bombardment."*

In spite of the titles and assertions of these articles, however, the actual data revealed no anomalous Ryder-like spike in impact melt dates in a 150 Ma pulse at 3.9 Ga ago, as shown in our Figure 7, taken from Cohen et al. (2000 [69]). The ages of 31 impact melt clasts, taken from four lunar meteorites, ranged from 2.43 to 4.12 Ga (an interval of ~1.6 Ga, with error bars mostly tens to hundreds of Ma). Their published ideogram of age distribution (probabilities of impact events, taking into account the error bars) showed four peaks. In order of height, these peaks occurred at 3.6 Ga ago (highest probability), 3.0, 3.4, and 3.9 Ga (lowest probability). Cohen et al. (2000 [69]) suggested that they had detected melts from seven impacts, which they dated at 2.76, 3.00, 3.05, 3.35, 3.43, 3.87, and 3.90 Ga ago. Thus, only 2 out of 7 occurred in a 200 Ma range from 3.8 to 4.0 Ga ago, the range assigned to the putative cataclysm.

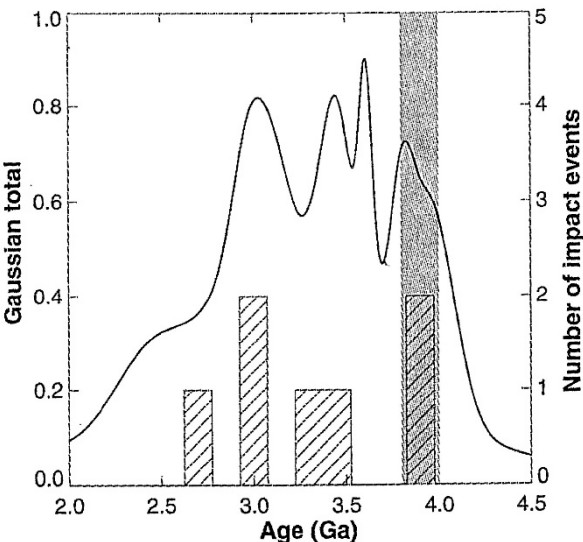

**Figure 7.** Age distribution data from 31 lunar meteorite impact melt clasts discussed by Cohen et al. (2000 [69]), and proposed in their title "support for the lunar cataclysm hypothesis." Cross-hatched data are histogram of best estimates for dates of seven large "impact events." Shaded area is a 200 Ma interval "representing lunar cataclysm." The curve is an ideogram adding Gaussian curves representing uncertainty of dates for the 31 samples. Following "Ryder's rule," the support for terminal cataclysm was proposed because none of the impact melt clasts pre-dated the proposed cataclysm. Note that time flows from right to left, contrary to most of our diagrams.

Why, then, did authors in this period claim "support for the lunar cataclysm" (or even that they had "confirmed the hypothesis that the Moon was resurfaced by an intense period of impact cratering")? The answer goes back to "Ryder's rule," the assertion that lack of impact melts equals absence of impacts. Cohen et al. (2000 [69]), echoing the logic of Tera et al. (1974 [3]), emphasized "lack of impact melt older than 3.92 Ga"—i.e., the "Ryder gap"—as evidence of an anomalous event at ~3.9 Ga.

Cohen, Swindle, Taylor, and Nazarov (2002 [70]) added more data and then Cohen, Swindle, and Kring (2005 [71]) gave a more complete analysis with dating of 42 clasts (including the earlier ones, from the same four lunar meteorites). The conclusions are essentially the same as in the first paper, but the support for a terminal cataclysm is slightly more muted. As expressed in Cohen et al. (2005 [71] abstract):

> *"We interpret these samples to have been created in at least 6, and possibly 9 or more, different impact events. One ... may be consistent with the Apollo impact-melt rock age cluster at 3.9 Ga, but the meteorite impact-melt clasts at this age are different in chemistry from the Apollo samples, suggesting ... a lunar-wide phenomenon. No meteorite impact melts have ages more than 1σ older than 4.0 Ga. This observation is consistent with, but does not require, a lunar cataclysm."*

These papers in prestigious journals, along with conference presentations, tended to solidify the belief that the terminal cataclysm paradigm had been confirmed, as perceived by the broader scientific community. To be more precise, however, these papers affirmed only that few impact melts were found from before 4.0 Ga ago. This could have been either because they never existed (as per Ryder's rule), or because they had been created but had not survived. Nonetheless, the perception was created that lunar meteorite impact melts confirmed the classic, brief era of intense impact around 3.9 Ga, with negligible impact flux in the preceding 400 Ma. This became a basis for more detailed interpretations of radiometric ages, petrologic data, and theoretical dynamical models regarding the first 600 Ma of Solar System history, as discussed in the next two Sections. The general perception is confirmed by the fact that dynamical modelers explicitly set to work ca. 2000–2005 to explain the putative cataclysmic bombardment spike at 3.9 Ga ago, and the paucity of impacts (impact melts, to be more precise) before then.

## 9. Do Apollo/Luna Samples Date 5 Different Front-Side Basins? The Radiometric/Petrologic Approach, circa 2001

The growing confidence that many of the lunar multi-ring basins formed ~3.9 Ga ago, during a ~150 Ma cataclysmic cratering episode, was followed by more assertive interpretations of ages reported for lunar samples. For example, certain Apollo and Luna impact melt samples were interpreted as coming from specific, different distant basins, on the basis of different petrologies, chemical compositions, and isotopic compositions, as well as slightly different ages scattered one, two, or three error bars apart in the ~3.72 to ~3.92 Ga interval. We will call this the "radiometric/petrologic" approach to lunar basin dating. Dates were, thus, assigned to various basins in spite of the absence of direct sampling from bedrock outcrops at collecting sites—a practice that might raise eyebrows in terrestrial geology. Wilhelms (1987 [73]) and Stöffler and Ryder (2001 [26]) give important examples of this technique. Table 1 gives selected, illustrative results. The Nunes et al. (1974 [74]) U–Th–Pb results are cited here as a striking early interpretation of radiometric dating results. Their Table 5 compared $^{207}Pb/^{206}Pb$ ages of "non-mare" rocks from all six Apollo landing sites, then took into account the ejecta blanket extents of all basins based on estimates by McGetchin et al. (1973 [75]). Their results allowed for much earlier Serenitatis, Nectaris, and Crisium dates than proposed by later radiometric/petrologic researchers, and more like the ages being proposed in recent years as the terminal cataclysm/LHB paradigm came under question (see Section 14).

The radiometric/petrologic approach was not without problems. In retrospect, we see an unstated, unexamined, but seemingly reasonable attitude during that 1974–2001 period: namely, that all interpretations of rocks *brought back from the Moon* provided ground truth that trumped any earlier, crude, "telescopic" interpretations, which lacked lunar ground truth.

An anecdotal, eyewitness report may illustrate the atmosphere at the time. The well-respected Caltech radiometric dater of the first lunar samples, Gerald Wasserburg, gave an invited public talk that filled a major auditorium at the University of Arizona, circa 1970–1972, about the meaning of the first rocks brought back from the Moon. As a young assistant professor, having written my dissertation on lunar crater counting and having published a pre-Apollo paper using such counts to predict successfully a characteristic 3.6 Ga age for lunar maria (Hartmann 1965 [17]), I attended with enthusiasm. I was soon mortified as Gerry jovially remarked that "Now you can flush crater counts down the toilet" (quoted from memory, but with the last three words still ringing in my ears).

Perhaps these words were aimed more at well-known Gene Shoemaker than at an obscure assistant professor. Shoemaker (1962 [22]), as a pioneer of much of modern planetary geology, had examined

some early US Department of Defense data on rate of fireball explosions in Earth's atmosphere, which had been thought to indicate a higher impact rate than expected. As it turned out, early DoD researchers had miscalibrated the total energy per event and resulting crater size on the Moon. As a result, Shoemaker delivered a number of highly visible, but little remembered, talks in the last pre-Apollo/Luna years, announcing a surprising new result that crater formation rates indicated a characteristic age for lunar maria on the order some few $10^8$ years or so. This turned out to be wrong by at least an order of magnitude—a rare error in Shoemaker's exemplary career, an error that Wasserburg was cheerfully correcting with his new lunar samples. Whatever the intent of Wasserburg's bon mot ridiculing crater-count chronometry, it still seemed to me that nature had handed us nicely circular landforms being created on planets at a potentially measurable rate, and that we humans could extract useful information from that situation.

Given the attitude about the primacy of the lunar samples, many dates and interpretations were published by the radiometric community without much comparison to the earlier, presumably outmoded lunar studies. Such studies included detailed stratigraphic mapping and analysis of *relative* ages by the US Geological Survey "astrogeology" group in Flagstaff, Arizona, as well as quantitative comparisons of impact crater densities on various lunar landforms. We will refer to such work as morphological studies. Here, we look retrospectively at conflicts between radiometric/petrologic interpretations and the earlier morphological analyses of the large, multi-ring lunar impact basins. Table 2 shows results of four (mostly early) independent morphological studies of age-related properties of five multi-ring basins, including quantitative measurements of crater densities. Most of the table entries date from before the Apollo 16 and 17 missions, and are thus, independent of the radiometric/petrologic interpretations of those sites.

As a preliminary caution to this discussion, note that while Table 2 gives a wide range of relative age rankings for basins such as Imbrium, Crisium, Nectaris, and Serenitatis, a wide range alone of *relative* crater-count ages does not argue against a terminal cataclysm. If a terminal cataclysm showered the Moon with impactors, forming most basins in a 150 Ma interval, then a basin only a few percent older than another (by tens of Ma) could have a crater density many times higher—which was exactly the hypothesis proposed by the early radiometric dating community. However, as we have seen in Section 8, radiometric ages of impact melt clasts in lunar meteorites now appear to refute the classic terminal cataclysm paradigm, i.e., a global, basin-forming, 150 Ma spike in impact rates. That refutation of cataclysmic cratering at 3.9 Ga ago suggests that the significantly higher crater densities shown in Table 2 actually do reveal significantly greater ages.

**Table 1.** Examples of radiometric-based estimates of lunar multi-ring basin ages 1974–2001.

| Basin | Nunes et al. (1974, [74]) * No Error Bars Cited | Wilhelms (1987, [73]) ** (Table 14.1, p. 278; Except Orientale) | Hiesinger, Jaumann, Neukum, and Head (2000, [76]) | Stöffler and Ryder (2001, [26]) Estimate 1 *** | Stöffler and Ryder (2001, [26]) Estimate 2 **** |
|---|---|---|---|---|---|
| Orientale | ~3.85 Ga | 3.8 Ga; 3.72–3.85 (p. 224) | — | 3.72–3.85 Ga | 3.72–3.77 Ga |
| Imbrium | ~3.99 Ga | 3.86 ± 0.04 Ga (p. 201); ≲3.87 Ga (p. 212); Well-constrained from 3.82 to 3.87; 3.85 was adopted (no error bar cited) (pp. 224 & 278) | 3.91 ± 0.1 Ga (citing Neukum and Ivanov) | 3.85 ± 0.02 Ga | 3.77 ± 0.02 Ga |
| Crisium | ~4.13 Ga | 3.84 ± 0.4 Ga; either Crisium age or Imbrium material (p. 171) | Not discussed. Ave. from 6 authors in their Table 7 = 4.01 Ga | 3.89 ± 0.02 Ga | 3.84 ± 0.04 Ga |
| Serenitatis | ~4.45? Ga | 3.86 ± 0.04 or 3.87 ± 0.04 Ga (p. 178) | 3.87 ± 0.04 Ga (K–Ar), 3.98 ± 0.05 Ga (Neukum crater counts) | 3.89 ± 0.01 Ga | 3.87 ± 0.03 Ga |
| Nectaris | ~4.2 Ga | 3.92 ± 0.03 Ga | 4.1 ± 0.1 Ga | 3.92 ± 0.03 Ga | 3.92 ± 0.03 Ga or 3.85 ± 0.05 Ga |

* U–Th–Pb results by Nunes et al. (1974 [74]); ** Wilhelms (1987 [73]) cites Ar–Ar and/or Rb–Sr ages and/or U/Pb/Th ages.; *** Stöffler and Ryder (2001 [26]) combined with Ryder and Spudis (1987 [77]), Wilhelms (1987 [73]); **** Stöffler and Ryder (2001 [26]) combined with Deutsch and Stöffler (1987 [78]), Stöffler et al. (1985 [79]), Jessberger et al. (1977 [80]), Staüdermann et al. (1991 [81]).

Let us discuss four examples of controversies resulting from differences between Tables 1 and 2. First is the convoluted story of ages assigned to the Serenitatis basin. In Table 1, some radiometric authors list Serenitatis as one of the youngest basins, close to Imbrium in age, or possibly even younger (within error bars); whereas in Table 2, Serenitatis was ranked as one of the oldest visible basins, based in part on quantitative crater density measurements. Figure 8 gives some idea of the scene during which the Apollo astronauts were collecting samples that may or may not have dated the underlying, original Serenitatis impact structure.

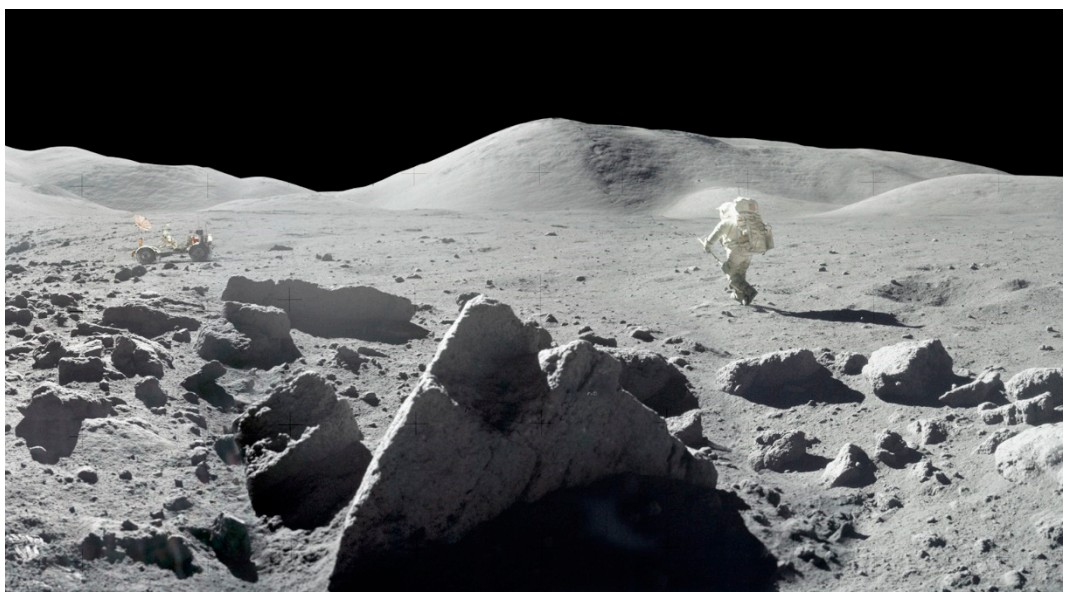

**Figure 8.** Collection of lunar rock samples at the Apollo 17 landing site, at an embayment in mountains at the western edge of Mare Serenitatis. (Photo mosaic from Lunar and Planetary Institute web site).

It should be noted here that measuring the crater densities corresponding to formation of the older basins such as Serenitatis (as well as Crisium, and Nectaris) is not as easy as making those measurements for younger basins like Imbrium and Orientale. In the latter cases, we see radially striated material, in which densities of craters postdating/interrupting the textures presumably date the creation of the ejecta blanket, but for the former cases, the ejecta blankets are less well-defined. It is significant, then, that in spite of the problems, Table 2 shows several independent crater density counts with comparably high ratios of Serenitatis/Imbrium crater densities. We will return to this point in a moment.

We note here also that Serenitatis, compared to the prominent, bulls-eye, ring structure of Orientale and the less well preserved (or developed?) rings of Imbrium, Nectaris, Humorum, and other basins (Hartmann and Kuiper 1962 [82]), appears so lacking in a clear concentric-ring pattern rim structure that one might even question whether it *is* an impact basin. However, gravitational data from GRAIL (Gravity Recovery and Interior Laboratory) studies (Zuber et al. 2013 [83], Wieczorek et al. 2013 [84]) show a well-defined mascon beneath Serenitatis, presumably supporting an impact origin. If Serenitatis is significantly older, the lack of clarity in its multi-ring structure might imply at first glance that the prominent rings were simply eroded by impacts at the small-scale end of the impactor size distribution. Remnants of sharply defined fault scarps, such as the Altai arc around Nectaris or the Apennine arc in Imbrium, are not to be found, however. Another possibility, then, is that the ring formation was not uniform through time but was favored in the last-formed basins. Conceivably, mechanical properties of the crust evolved during the 600 Ma basin-forming period. For example, the lithosphere presumably thickened during cooling of the primordial magma ocean, producing a more brittle crust overlying semi-molten layers, allowing the faulting and slumping process that seems to have created the rings after formation of the transient cavity. The very ancient South Pole–Aitken basin, for example, lacks

a clearly developed ring system, and even lacks a mascon. Serenitatis lacks well-developed rings but has a mascon. At the other extreme, Orientale, considered the youngest multi-ring basin, has the best developed ring system and a well-defined mascon. Young-to-middle-age basins, such as Imbrium, Nectaris, and even Humorum have well-traceable fragments of rings, partly buried by mare lavas (Hartmann and Kuiper 1962 [82]; Hartmann and Wood, 1971 [20]; Hiesinger et al. 2000 [76]), and they all show mascons (Zuber et al. 2013 [83]). If the ring structure correlates with age, and the age correlates with cooling and formation of crustal structure, then the ring structure requires long geologic periods of crustal evolution. In that case, long periods are implied for basin formation, longer than a 150 Ma spasm 600 Ma after lunar formation. In this view, Serenitatis would have to be assigned an age long before Imbrium, rather than within the error bars of the Imbrium date.

To cite a contrary view, published in 2000 when the terminal cataclysm or LHB paradigm was at its height, the well-respected researchers Hiesinger, Jaumann, Neukum, and Head (2000 [76]), in a study of ages primarily of lunar mare basalts, referred to "the youngest basins, Serenitatis and Imbrium" (abstract, p. 29, 239). They mention (p. 29, 261) a K–Ar Serenitatis basin formation age from Staudacher et al. (1978 [85]) of 3.87 ± 0.04 Ga, which would make Serenitatis about one error bar younger than their Imbrium age of 3.92 Ga, which they accept in their Figure 4. Wilhelms (1987 [73], p. 171), however, in his discussion of Serenitatis, concluded that Imbrium ejecta completely covered and obliterated the northern parts of the original Serenitatis ring structure, between Imbrium and Nectaris, ruling out a Serenitatis age younger than Imbrium. This would be consistent with the hypothesis that surface impact melt samples in that region north of Serenitatis (including the Apollo 16 site) would show predominantly the Imbrium age. Hiesinger et al. (2000 [76], p. 29, 261) went on to cite a crater-count age from co-author Neukum, of 3.98 ± 0.05 Ga "for the Serenitatis Impact," and in their Figure 4, they cite "3.98 b.y." for their proposed age of the Serenitatis basin, along with "3.92 b.y." for Imbrium. A different interpretation would be that the K–Ar age they cite for Serenitatis is really from Imbrium ejecta blanketing Serenitatis.

Neukum's chronometric system deserves further comment here. Emphasizing precision in counts, it involved least squares fit of crater counts to his proposed lunar production size-frequency distribution, and also used Neukum's analytic equation relating the size-frequency distributions to time. As a result, ages were cited often to three significant figures. The present author has argued, however, that crater counts are not perfect statistical data, since crater identification must be judged, not to mention that the calibration of lunar surface age versus crater density is still poorly measured. Thus, the crater-chronometric system involves intrinsic uncertainties of roughly 10% or more in relative age, and even more in *absolute age*, so that crater retention *absolute* ages are certainly not good to three significant figures, although *relative* ages can probably be judged to 10% accuracy (Hartmann et al. 1981 [86], Hartmann 2003 [52], Hartmann and Daubar 2017 [87]). As a result, ages in the Neukum system, while giving excellent *relative* ages, should be seen only as *model absolute* ages, probably not valid to three significant figures. In any case, the 60-Ma year difference in these model ages between Imbrium and Serenitatis, listed by Hiesinger et al. (2000 [76]), is within typically 1, 2, or 3 cited radiometric error bars, cited in Table 1 for basin ages.

Table 2 offers a way to test the absolute difference in ages of Serenitatis and Imbrium, based on the *rate of change* in the early impact flux. As noted in Section 5, crater count data, when coupled with Apollo and Luna sample dating, allowed approximate reconstruction of cumulative crater densities versus surface age. These data extended at least as far back as 3.8–3.9 Ga ago. (As mentioned above, the ~3.8–3.9 Ga boundary appears because surfaces older than ~3.9 Ga are saturated with craters, causing specific dating information from before ~3.9 Ga to be lost). In one of the most detailed studies of cratering rate, Neukum (1983 [36]) and Neukum et al. (2001 [38]) developed their curve of cumulative crater density versus age, mentioned in the previous paragraph. This curve suggests that the half-life for decline in presently observed crater density, in the era around Imbrium formation, was ~100 Ma. This fits with the 1970s empirical estimates of gradually lengthening half-life around this time, such as 50–150 Ma, and 70 Ma, mentioned at the beginning of Section 5.

From Table 2 we find best-estimates of the Serenitatis/Imbrium ratio of crater densities are 11.2 (Hartmann and Wood 1971 [20]) and 9.9 (Fassett et al. N(20), 2012 [88]). Using the Neukum half-life, this would represent ~3.31 to ~3.49 half-lives, implying that Serenitatis formed on the order of 331 to 349 Ma before Imbrium. If we adopt 3.85 Ga as the fairly well-determined date of Imbrium from Table 1, both these figures would predict a Serenitatis age of ~4.2 Ga. Indeed, as early as 1975 [10], Turner and Cadogan reported an Apollo 17 Ar–Ar rock age of 4.22 Ga (Serenitatis landing site). If, instead, we adopt a half-life of 50 Ma for this time period as an example (probably too short), the results of this logic give a minimal Serenitatis age of ~4.0 to ~4.1 Ga. If we adopt a 150 Ma half-life, we find a maximal Serenitatis age of ~4.35 Ga. To conclude, the crater chronometry system suggests a few-hundred Ma older age for Serenitatis than for Imbrium, measurably outside the error bars typically cited for Imbrium. Consistent with this, Liu et al. (2019 [89]) adopt an age of 4.13 Ga, consistent with the above estimates (see discussion a few paragraphs below).

**Table 2.** Morphological and crater-chronometric relative ages of three basins (from four independent papers).

| Basin | Relative Age Rank by "Rim Morphology Class," Where 0 = Fresh and 10 = Barely Visible [1] | Relative Age Rank Out of 27 Basins [2] | Relative Age Rank Out of 31 Basins [3] | Crater Density Relative to Average Mare [3] | Crater Density N(20) (Cumul. craters > 20 km per $10^6$ km²) [4] | Crater Density N(64) (Cumul. Craters > 64 km per $10^6$ km²) [4] |
|---|---|---|---|---|---|---|
| Orientale | Class 2 | 1st of 27 | 1st of 31 | 2.4 | 21 ± 4 | 1 ± 1 |
| Imbrium | Class 3 | 3rd of 27 | 2nd of 31 | 2.5 | 30 ± 5 | 4 ± 4 |
| Crisium | Class 4 | 4th of 27 | 18th of 31 | 17 | 117 ± 11 | 8 ± 3 |
| Nectaris | Class 7 | 11th of 27 | 14th of 31 | 16 | 135 ± 14 | 17 ± 5 |
| Serenitatis | Class 8 | 17th of 27 | 28th of 31 | 28 | 298 ± 60 | 28 ± 20 |

[1] Baldwin (1969b [48]), [2] Stuart–Alexander and Howard (1970 [90]), [3] Hartmann and Wood (1971 [20]), [4] Fassett et al. (2012 [88]).

Such data conflict with radiometric/petrologic assertions that the Serenitatis impact date has been directly dated within error bars of 30 Ma, for example at 3.87 ± 0.03 Ga (Table 1). The error bars may be legitimate for certain rock samples, but the question is what those rock samples represent.

With this in mind, we note that the radiometric dates for Serenitatis come primarily from Apollo 17 samples, collected at Serenitatis rim hillsides at the far side of Serenitatis from Imbrium. The rocks are petrologically different from most Imbrium rocks. Still, these rocks came from more or less Imbrium-facing hillsides, and were potentially exposed to effects of the Imbrium ejecta. The question thus emerges: What was the effect of Imbrium ejecta on the neighboring Serenitatis basin system? Do the Apollo 17 samples really date Serenitatis?

McGetchin et al. (1973 [75]) developed an early theoretical model of ejecta blanket thickness versus distance from source, and estimated that in the Apollo 17 area, the ejecta deposits from Imbrium (postdating the Serenitatis basin surface) would be 26–102 m thick, depending on parameters used for the radius of the Imbrium excavation, which was estimated theoretically from the observed multi-ring system of Imbrium. In addition to Imbrium, however, other basins thought to postdate Serenitatis would also contribute their own ejecta to the blanket overlying Serenitatis, even if all the basins formed in a 150-Ma cataclysm. McGetchin et al. (1973 [75]) tabulated accumulation from all major basins at various Apollo sites, assuming that Orientale, Imbrium, Crisium, Humorum, and Nectaris all postdate Serenitatis, so that the total of ejecta *deposited upon the Apollo 17 site* jumps to 54–228 m of material postdating Serenitatis. The question arises: Would this total estimate of 26–228 m suppress Serenitatis material and favor Imbrium material as being exposed at the surface for collection by astronauts?

In answer to this question, Petro and Pieters (2006 [62]) caution that as masses of distant basin ejecta fall at near-escape velocity onto a lunar surface, boulders entrained in the ejecta form craters, so that considerable amounts of the *original* near-surface material are excavated and mixed into the newly forming ejecta blanket. In this view, even the last blanket might contain material from the original surface (Serenitatis). As a schematic thought experiment, again citing the McGetchin et al. (1973 [75]) calculations, suppose 6–26 m of Nectaris material are dumped on the Apollo 17 site, and then 22–97 m

of Crisium material arrive, and finally 26–102 m of Imbrium material arrives (here we neglect effects of the thin layers from Humorum and Orientale layers at this distance). Let us now allow mixing during each layer addition and assume that each subsequent layer ends up with 30% of its material coming from the underlying layer (a plausible percentage from Petro and Pieters 2006 [63], although they also discuss more complete mixing, such as 50%). The 6–26 m Nectaris ejecta layer now contains 30% Serenitatis material and grows to ~8–34 m thickness. Then the Crisium ejecta layer arrives and is mixed with the Nectaris layer, so it contains only 9% Serenitatis material (assuming the mixing does not penetrate through the Nectaris layer), and foreign ejecta becomes ~29–126 m thick. In the same way, the Imbrium layer now absorbs material from the Crisium layer and contains 3% Serenitatis material and becomes 34–133 m thick. In this schematic scenario, the Serenitatis basin would now have a shield of ejecta blankets some 71 to 294 m thick and with a top layer containing only 3% Serenitatis samples and much more abundant Imbrium samples—not to mention post-Imbrium lava covering its floor.

An additional source of Serenitatis material exists: During the entire post-Imbrium period, a gradually declining flux of exogenic meteoroid impacts (whether being the tail end of an LHB spike or being the tail end of a longer-term process) would also excavate craters deep enough to pepper each blanket's surface with rocks from Serenitatis material. Such impacts, however, would have to penetrate through the Imbrium ejecta blanket to reach the Serenitatis basement; much ejecta from these craters would be from the Imbrium ejecta blanket, thus diluting any Serenitatis ejecta.

A more recent estimation of the Serenitatis impact melt ejecta, published after the above text was drafted, comes from a numerical model of Serenitatis, Crisium, and Imbrium ejecta blanket evolution by Liu et al. (2019 [89]). For those three basins, they assume ages of 4.13, 4.09, and 3.88 Ga, respectively. They included impact destruction processes in the interval between the Serenitatis basin's formation and deposition of ejecta from later basins, as well as the masking and dilution of all those materials by the Imbrium basin ejecta. Their model, similar to our above reasoning but with more detail, leads to this summary in their conclusion:

> *"The survival probability of basin melt at the Apollo and Luna sampling sites is quantitatively assessed . . . The relatively young Imbrium melt might be abundant at Apollo 14–17 sampling sites with a fraction ranging from 0.3 to 0.6; Crisium melt could be found at Luna 20 and Apollo 17 sampling sites with a similar fraction about 0.05, each. The relatively old Serenitatis melt was exposed to heavy subsequent gardening, and its abundance should be much less or zero at these sampling sites. The observed prominent peak around 3.88 Ga, the lower values around 4.09 Ga, and the general absence around 4.13 Ga in the K–Ar isotopic ages from Apollo and Luna highland samples are consistent with our simulation results. We...conclude that, particularly for the case of Imbrium, the clustered radiometric ages around 3.9–4.0 Ga for Apollo and Luna highland samples supports a sample bias, rather than the cataclysm scenario."*

Acting against survival of Serenitatis material on the post-Imbrium Moon (and making reality still more complicated) are destruction processes that limit survival time of any boulders placed on the lunar surface by recent impact-excavation events. Basilevsky et al. (2013 [91]) found that, in the current lunar meteorite impact environment, 99% of boulders of diameter >2 m ejected from impact craters onto the crater-rim surfaces (from whatever depth) are reduced to sizes <2 m in 130–300 Ma, and rocks of ~10 cm size are estimated to be mostly converted into regolith in about 50 Ma. Similarly, Ghent et al. (2005 [92]: 2014 [93]) observed from visual, radar, thermal infrared imagery that meter-scale rock abundance, not only on ejecta blanket surfaces but also in the near-surface ejecta layers, declines by a factor of ten (to near disappearance in surface images) in about 800 Ma. Consistent with this, Apollo rock samples' cosmic ray exposure ages are rarely more than a few hundred million years (e.g., Eugster 2003 [67]).

Thus, the rocks that astronauts can collect on the surface do not represent the unblemished impact history (as often assumed during initial Apollo planning), but rather are strongly filtered by regolith formation processes. Small, collectible-sized rock specimens ejected from Serenitatis material,

if buried under (or in) the Imbrium ejecta, would have had to be ejected onto the Apollo 17 surface as collectible-sized specimens within the last few hundred Ma for astronauts to pick them up. For a collectible-sized ejected specimen to have survived on the surface from the time of the Serenitatis impact, or even the Imbrium impact, it would have to be the last remnants of a rare, room-sized boulder. However, during a typical rock fragmentation event, the size distribution of fragments is such that an increase in fragment size by a factor 10 involves a decrease in frequency by factors of 100 to 1000 (Hartmann 1969 [94]), so that recently ejected hand-sized specimens would probably outnumber remnants of ancient, giant boulders.

To consider another approach to the Serenitatis age/sample problem, we cite more recent, relevant, radiometric results. Thiessen et al. (2017 [95]) addressed the age of Serenitatis by looking at U–Pb ages of small grains in four breccia samples collected from two massifs at the Apollo 17 landing site. Assuming that the different chemistries and textures were associated with different impacts, they inferred that the four rocks gave dates from three different impacts, at $3.920 \pm 0.003$, $3.922 \pm 0.005$, and $3.930 \pm 0.005$ Ga ago ($2\sigma$ error bars). They discussed various interpretations, including that the data represent a $3.930 \pm 0.005$ Ga age for Serenitatis and $3.922 \pm 0.005$ Ga for Imbrium. However, they also noted (considering error bars) that this would require the "about 13–25" basins believed to have formed between the Serenitatis and Imbrium events to have formed in "an extremely narrow time interval of only a few million years." Based on additional discussion, their abstract suggests this time interval would have to be "<11 Ma." This, they stated, "highlight(s) serious contradictions between global stratigraphic constraints, sample interpretation, and chronological data."

Thiessen et al. (2017 [95]) pointed out that the McGetchin et al. (1973 [75]) models suggested substantial Imbrium ejecta at the Apollo 17 site, and that incomplete resetting of the U–Pb system in the ejecta may produce differences in measured ages. A justification usually raised at this point, in favor of 3.9 Ga Apollo 17 ages representing the Serenitatis impact, is that petrologic differences exist between A17 samples and most samples collected in or near Imbrium. The counterargument, mentioned above, is that ejecta from different parts of Imbrium's transient cavity, >700 km across, could have produced different mixes of material (Hartmann 2003 [52] pp. 589–591; Thiessen et al. 2017 [95]).

Another justification for interpreting A17 samples as giving Serenitatis ages is that some of the dates come from rocks that rolled down hill slopes adjacent to the Apollo 17 sites, with an interpretation that they came from outcrops of Serenitatis rim formations. It appears equally plausible that these boulders derived from the mass of Imbrium ejecta striking the slopes above the A17 site, only to roll down into the landing area explored by the astronauts. Spudis et al. (2011 [96]) examined Lunar Reconnaissance Orbiter imagery and concluded:

*"1. The Sculptured Hills material of the Montes Taurus [in and around Serenitatis] is a distal facies of Imbrium basin ejecta and is not directly related to the Serenitatis basin forming impact. 2. The relative age of Serenitatis is pre-Nectarian ... 3. Impact melt samples returned by the Apollo 17 mission may not be derived from the Serenitatis basin forming impact but could instead be from Imbrium."*

Our convoluted discussion of Serenitatis does not claim to give a final interpretation of Apollo 17 sample issues, but rather illustrates the problems still associated with the radiometric/petrologic approach. Four decades after Apollo 17, the issues have not been resolved, and one must begin to question whether samples from the Apollo 17 surface, with dates within 2 or 3 error bars of the better-determined Imbrium age, can really be interpreted as having nothing to do with Imbrium.

A second major example of a problem with the radiometric/petrologic approach comes from the reported dating of the Serenitatis impact not as compared with Imbrium, but *vis-à-vis* the Nectaris impact. Most researchers cited in Table 1 asserted from radiometric data that the Serenitatis basin is *younger* than the neighboring Nectaris basin, or at least of similar age within one error bar in reported ages. This conclusion, based on the radiometric/petrologic approach, conflicts with the morphological evidence. In the first place, Nectaris, with its well-defined Altai scarp (part of an Orientale-like multi-ring pattern), has a much "crisper" structure than Serenitatis, which lacks such obvious ring

structure. The well-defined rings or ring arcs of Nectaris are most typical of the youngest basins, like Orientale and Imbrium. To be more specific, Table 2 includes six independent estimates of the relative ages of Nectaris and Serenitatis, including two sets of quantitative crater counts. In *every* study, the Serenitatis basin was ranked as being significantly *older* than Nectaris. The crater-count age estimates involve crater densities measured, of course, not on the later mare lava fill, but on the rim zones and close-in ejecta blankets. Using data in Table 2 to compare the two basins, we found that Serenitatis is listed by various authors, relative to Nectaris, as:

- In class 8 versus class 7 for Nectaris, according to 10 classes based on rim morphology (oldest = 10) defined by Baldwin (1969 [97]), who combined crater counts with morphological traits;
- 17th oldest versus 11th oldest for Nectaris, out of 27 basins listed in "approximate relative age sequence" based on "degree of modification" which included rim and ejecta clarity, size of largest superimposed crater, and subjective judgments by Stuart–Alexander and Howard (1970 [90]);
- 28th oldest versus 14th oldest for Nectaris, out of 31 basins based on measured crater densities measured by Hartmann and Wood (1971 [20]);
- *Oldest* versus 11th oldest for Nectaris, out of 30 basin structures, according to the N(20) crater densities (cumulative density at D >20 km), measured by Fassett et al. (2012 [88]);
- Tied for 16th oldest versus 1st–3rd oldest for Nectaris, out of 29 measured basins in N(64) 64-km craters listed by the same authors (Fassett et al. 2012 [88]). N(64) had poorer statistics than N(20), due to the smaller number of large craters.

In addition to the morphological indicators listed above, Fassett et al. (2012 [88] Figure 8) proposed evidence of sculptured ejecta emanating from Nectaris, superposed on the Serenitatis system.

Amidst the quantitative statistics of Table 2, the ratios of crater densities the Serenitatis basin to those of Nectaris are notably consistent among the different observers:

- Hartmann and Wood (1971 [20]) fitted their crater counts to isochron curves over wide range of diameters and found Serenitatis has 1.8 times higher impact crater density than Nectaris.
- Fassett et al. (2012 [88]) found Serenitatis has 2.3 times higher crater density than Nectaris using their statistic for craters larger than 20 km.
- Fassett et al. (2012 [88]) found Serenitatis has 1.6 times the crater density of Nectaris, using their statistic for craters larger than 64 km.
- (An aside: Stuart–Alexander and Howard (1970 [90]) did not list or apparently measure crater densities).
- (Another aside: Baldwin (1969 [97]) provided fewer clear data. He listed logarithmic values of crater densities, but the value listed for Serenitatis appears to be a misprint or typo, since it is markedly *lower* than the value for Imbrium or Nectaris, contradicting his own table, which indicates that the crater density range that correlates with Serenitatis's rim morphology is his oldest class, class 8. If we use the range of crater densities he gives for that class, Baldwin's (1969 [97]) value for the ratio of Serenitatis to Nectaris crater densities would be 1.1 to 1.8, which overlaps with the range of the later measurements listed above).

In spite of the fact that these independent morphologic measurements make Serenitatis *older* than Nectaris—and raise questions about the radiometric/petrologic age interpretations—many contemporary papers perpetuate the impression that Serenitatis is a well-dated, young basin, even a very typical multi-ring impact basin. For example, Martellato et al. (2017 [98]), reviewing Serenitatis basin geology, cite a "constrained" 3-significant-figure 3.89 Ga age for Serenitatis, and cite Head (1979 [99]) as establishing that Serenitatis is "characterized by four rings with diameters of 410, 620, 1300, and 1800 km"—giving little indication of the above 40-year controversies or the obscure nature of the rings. In short, the "Serenitatis case" highlights the frustrating limitations of the existing lunar site sampling, and the radiometric/petrologic interpretations that led to the terminal cataclysm paradigm.

A third example of radiometric/petrologic problems is a controversy involving the Apollo 16 samples vis-à-vis the age of the Nectaris basin relative to the Imbrium basin. Table 1 illustrates the common assertion (at least at the height of the terminal cataclysm paradigm) that Apollo 16 samples from a landing site near the Nectaris basin measured an age for the Nectaris impact within 70–190 Ma of age of Imbrium, i.e., within 2 cited error bars. An important early study of the Apollo 16 samples was that of James (1981 [100]), who proposed a Nectaris impact age of ≲3.92 Ga, fitting the terminal cataclysm paradigm and influencing the ages shown in Table 1, e.g., Stöffler and Ryder (2001 [26], estimate 2). Three problems arise here. First, McGetchin et al. (1973 [75]) calculated a 12 to 50 m depth of Imbrium ejecta deposited upon 44 to 202 m of Nectaris ejecta in the Apollo 16 region, suggesting the possibility that the Apollo 16 samples sampled mostly Imbrium ejecta. Second, as shown in Figure 9 and mentioned in Section 3, the Apollo 16 site shows striations created by kilometer-wide grooves and ridges lying radial to *Imbrium*, not Nectaris, clearly establishing the dominance of at least some patches or effects of Imbrium ejecta in the Apollo 16 region. Third, a significant amount of material dating to ~4.2 Ga has been reported amidst the Imbrium-age Apollo 16 samples. Interestingly, as early as 1974, this age was suggested by Nunes et al. (1974 [74]) as an age for Nectaris (without much acceptance) (see Table 1). To cite more recent examples, starting around 2011, Fischer–Godde and Becker (2011 [101]) stated that "The new age data [of 4.2 Ga] for 67935 and 67955, and their different HSE composition show that there was a significant pre-4.0 Ga bombardment history affecting the highlands near Nectaris . . . These new age data supports previous notions that the Nectaris impact basin may be as old as 4.2 Ga." Norman and Nemchin (2012 [102]) described sample 67955 as an "impact melt breccia" and obtained an independent (U–Pb) date of 4.20 ± 0.07 Ga. Norman and Nemchin (2012 [102]), along with Norman et al. (2015 [103]), suggested on the basis of KREEP-like petrology, however, that this rock may have been entrained in the local Imbrium debris, rather than coming from Nectaris itself (see Section 12). Figure 10 gives a sense of the scene at the Apollo 16 site, where the local surface rocks may have included a wide variety of materials.

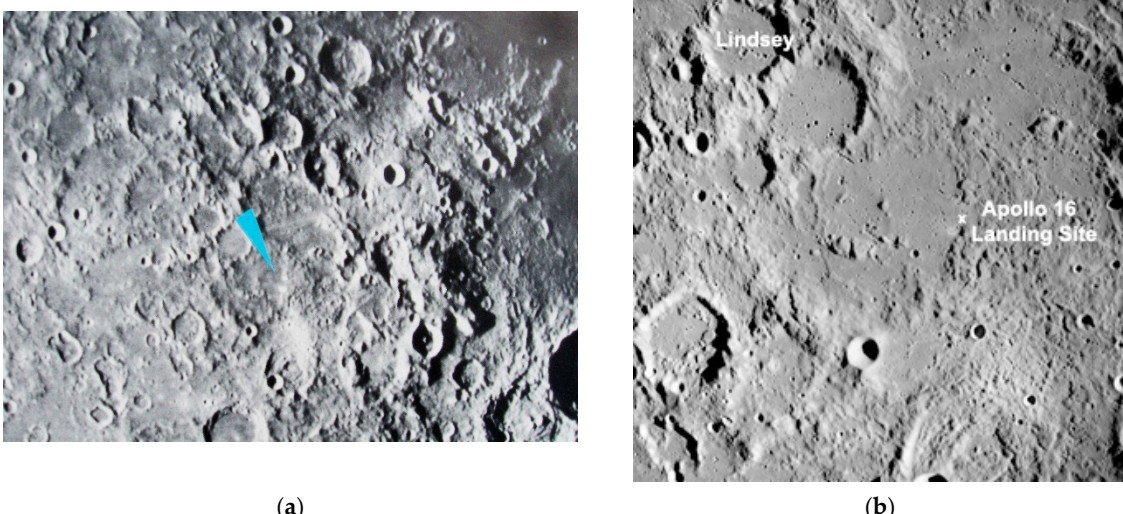

(**a**)　　　　　　　　　　　　　　　　　　　　　(**b**)

**Figure 9.** Lunar surface surrounding Apollo 16 landing site at two resolutions. North is at the top. The region on both sides of the landing site is laced by diagonal northwest-to-southeast linear ridges and valleys, which are part of lineament pattern radial to the Imbrium impact basin, parallel to blue arrow near center of image. Imbrium lies far offstage to the upper left. Nectaris impact basin is just out of frame to right, but Imbrium-produced modifications of the landscape dominate over any radial pattern from Imbrium. (**a**) Broad area showing Imbrium-radial lineaments of ~2–4 km width. Width of image is 550 km. Near-center arrow shows landing site. ("Overhead" image, from Rectified Lunar Atlas; Whitaker et al. 1964 [104]) (**b**) Narrower areas showing Imbrium-radial detail at scales of ~1–2 km width. Width of image is 175 km (Orbital image).

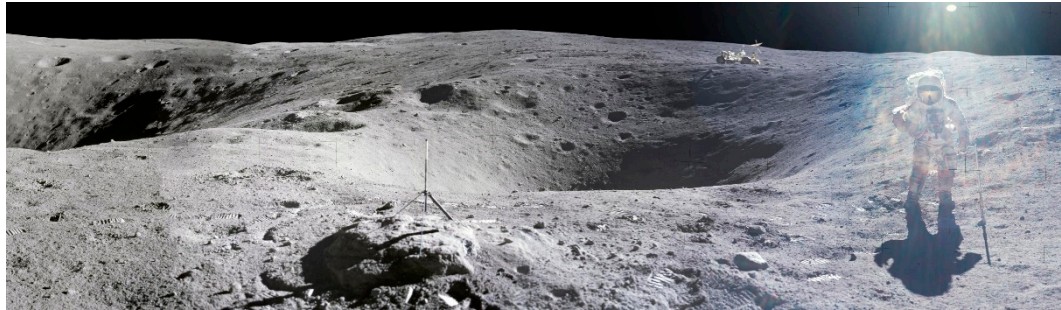

**Figure 10.** The landscape at the Apollo 16 landing site, near the Nectaris basin, but also amidst the topographic features radial to Imbrium basin. As suggested here, craters may have excavated Imbrium blanket material and/or underlying Nectaris materials. (Glare from the sun appears at right. Photo mosaic from Lunar and Planetary Institute web site).

In short, there is growing evidence that Nectaris (or at least some basin-scale impacts sampled by Apollo 16) may date from well before the alleged basin-forming spike around 3.9 Ga ago. In that case, the "traditional" radiometric/petrologic methodology, assuming that Apollo sample ages at ~3.8 to ~4.0 Ga (a range of 1, 2, 3 investigator-cited error bars) refers to many individual nearby basins (rather than Imbrium ejecta) has been too simple.

A fourth example of problems with the radiometric/petrologic approach comes from Wilhelms (1987 [73]), who acknowledged problems with the methodology but listed the Crisium basin as younger than the Imbrium basin (within an error bar). This radiometric result seems dubious, based on the data in Table 2 and the near absence of a visible multi-ring system around Crisium. In Table 2, all but one of the rankings list Crisium as older than Imbrium. The ratios of Crisium/Imbrium crater densities, as listed from Hartmann and Wood (1971 [20]) and from Fassett et al. (2012 [88]) N(20) and N(64) counts, are high—6.8, 3.9, and 2.0, respectively—indicating an older age for Crisium (though we note again, that if most basins formed during a 150 Ma bombardment, short time intervals would produce high crater density ratios). In contrast, the ratio of best estimate Crisium/Imbrium radiometric ages cited by Wilhelms (1987, [73], Table 1) is 1.0.

The Crisium crater counts and radiometric ages each have their problems. Crater densities on the Crisium ejecta blanket would presumably date the basin's formation. But, as mentioned above, in the case of (older?) basins like Crisium, defining the ejecta blanket is an issue. It is significant, then, that the independent estimates of the Crisium/Imbrium crater density ratio agree as well as they do, indicating that Crisium has experienced some 2 to 7 times more cratering than Imbrium.

Wilhelms' (1987 [73]) radiometric age for Crisium, in Table 1, was based on dating by Podosek et al. (1973 [105]) of two rock fragments found in a small drill core sample returned by the Russian Luna 20 craft, which landed on the upland rim of Crisium. Wilhelms, to his credit (p. 171), is cautious:

> *"Most of the fragments are of [plutonic anorthosite/norite/troctolite] rock not unlike the Apollo 16 suite; some are of low-K KREEP composition . . . They have been dated at 3.84 ± 0.04 aeons . . . an age consistent either with a [pre-Imbrium] age of Crisium or with an Imbrian age . . . In view of the small size of the sample and the difficulties encountered in dating the larger Apollo 16 and 17 sample suites, no firm conclusions should be based on this age despite its apparent analytical precision."*

Wilhelms allowed that the Luna 20 material might relate to Imbrium, but nonetheless offers it as giving a plausible date for Crisium impact, 10 ± 40 Ma younger than his own date for Imbrium.

To summarize this arduous Section 9, the radiometric/petrologic approach to Apollo and Luna samples in the period around 1970 to early 2000s, in spite of seeming very straightforward, assigned questionable impact ages to several basins that had not been adequately explored on the ground, and these ages were within few-error-bar range of the age of Imbrium ejecta. The sample interpretations were less reliable than they seem at first glance. Both radiometric and crater chronometric ages, used

to assert that lunar basin ages fit within a 150-Ma-wide spike, were probably being cited to unrealistic levels of precision. Table 3 shows estimates of ages, from 1987 to 2001, for four major basins. The top line shows the ~1σ age range for Imbrium, cited by various authors (see Table 1 for details). The other lines compare 1σ, 2σ, and 3σ age ranges cited by the various authors for Serenitatis, Crisium, and Nectaris. The Table shows that in 10 out of 12 available cases, the 2σ uncertainty cited for Serenitatis, Crisium, and Nectaris overlapped with the 1σ uncertainty being cited for Imbrium. While such a result could fit the terminal cataclysm scenario, it should have raised more interest in the possible dominance of Imbrium ejecta. That concern is currently exacerbated by detection of pre-Imbrium impact melt age clusters around 4.2 and 4.3 Ga in age (as discussed further in Section 12). All of which supports the current, widespread desire for a new (internationally coordinated?), carefully targeted lunar sample return program, designed to get more reliable ages for a variety of multi-ring impact basins.

**Table 3.** Reported radiometric ages of Imbrium (Showing 1-error-bar age ranges) of Imbrium, compared to radiometric awithin 1, 2, 3 reported error bars (circa 2000) for three other lunar basins (Ga). (Underlined values overlap with Imbrium as listed in top line by the given author).

| Basin | Wilhelms (1987 [73]) | Hiesinger et al. (2000 [76]) | Stöffler and Ryder (2001 [26], Estimate 1) | Stöffler and Ryder (2001 [26], Estimate 2) |
|---|---|---|---|---|
| Imbrium (~1 error bar) | 3.82 to 3.87 Ga "well constrained" | 3.90 to 3.91 Ga | 3.83 to 3.87 Ga | 3.75 to 3.79 Ga |
| Serenitatis (1 error bar) | 3.82 to 3.91 Ga | 3.83 to 3.91 Ga | 3.88 to 3.90 Ga | 3.84 to 3.90 Ga |
| Serenitatis (2 error bars) | 3.78 to 3.95 Ga | 3.79 to 3.95 Ga | 3.87 to 3.91 Ga | 3.81 to 3.93 Ga |
| Crisium (1 error bar) | 3.80 to 3.88 Ga | - | 3.87 to 3.92 Ga | 3.80 to 3.88 Ga |
| Crisium (2 error bars) | 3.76 to 3.92 Ga | - | 3.85 to 3.93 Ga | 3.76 to 3.92 Ga |
| Nectaris (1 error bar) | 3.89 to 3.95 Ga | 4.0 to 4.2 Ga | 3.89 to 3.95 Ga | 3.89 to 3.95 Ga or 3.80 to 3.90 Ga |
| Nectaris (2 error bars) | 3.86 to 3.98 Ga | 3.9 to 4.3 Ga | 3.86 to 3.98 Ga | 3.86 to 3.98 Ga or 3.75 to 3.95 Ga |

## 10. Dynamical Models "Explain" the Terminal Cataclysm, circa 2005

By the early 2000s, the then widely accepted "classic" cataclysmic spike in impact rate at 3.9 Ga ago presented a problem for Solar System dynamicists. Did significant numbers of the planet-forming planetesimals last until 4.0 Ga ago? Or were they all swept up by that time, so that virtually no impacts happened from ~4.4 Ga ago until 4.0 Ga ago? If the latter case, could there be a source for planetesimals to appear suddenly at 4.0 Ga to explain the classic cataclysm concept? Such questions led to what we will call the "dynamical model" methodology of interpreting lunar evidence.

As we have seen in Section 1, Wetherill (1975 [7]; 1977 [8,9]) investigated decay of planetesimal populations during the tail end of planetary accretion and looked for a "storage" process (his word) that might explain either a delayed start to a "late heavy bombardment," or a decaying intense cratering rate as late as ~3.9–~3.6 Ga ago. As also noted in Section 1, Wetherill (1975 [7]) related his term "Late Heavy Bombardment" to the Tera et al. (1974 [3,4]) cataclysm at 3.9 Ga ago, whether due the to Imbrium or Imbrium plus other contemporaneous "events" (Tera et al. term).

Morbidelli et al. (2001 [106]), in a paper titled "A Plausible Cause of the Late Heavy Bombardment," proposed that " . . . at the end of the main accretional period of the terrestrial planets, a few percent of the initial planetesimal population . . . (are) left on highly-inclined orbits." Their model produced a leftover population several times more massive than the present asteroid belt, which, in turn, could explain an extended decay of the impact rate. They noted that "after a rapid drop in the first 10 Ma, the decay of the population slows, with a median lifetime on the order of 50–60 Ma." This would produce a long-lasting impact flux, with decline still happening from ~3.9 to ~3.6 Ga ago, as observed in the lunar data. The model was similar to earlier, less sophisticated suggestions of a long decay with increasing half-lives for the planetesimals, such as Hartmann (1970b [32]) and Wetherill (1975 [7] "sense" #2).

Morbidelli et al. (2001 [106]) concluded that their model would not explain a Ryder-like spike, and it is important to note that they were suggesting that a "plausible" scenario would include "an exponentially decaying late heavy bombardment . . . " until ~3.6 Ga ago, with no spike. This definition of the phrase "late heavy bombardment" incidentally contradicted the contemporaneous definition of

"LHB" by Levison et al. (2001 [12]) as the period "roughly 4.0 to 3.8 Gyr ago" when "the lunar basins with known dates were formed." As is clear from our Section 1, and to be fair, the Morbidelli et al. usage of "LHB" did fit one of the "senses" that Wetherill originally considered for LHB (a poorly-defined "end" of a declining impact flux—when does a declining curve end?), but it violated the other sense, which, as argued here, came to dominate the perception of LHB, namely, the Levison et al. usage. The ambiguity of terms introduced a kind of semantic chaos into the evolution of the terminal cataclysm paradigm. "LHB" was no longer equivalent to a spike in impacts or formation of most basins between ~4.0 Ga ago and ~3.85 Ga ago, as required by some usages by Wetherill (1975 [7]; 1977b [9], based on Tera et al. (1973 [2]; 1974 [3,4])) and by the descriptions of bombardment history by Ryder (1990 [11]). Rather, "LHB" was now used to refer to any scenario in which elevated, declining levels of cratering lasted until ~3.9 or ~3.6 Ga ago. In other words, the "LHB," instead of being a radical departure from pre-Apollo concepts, as were the Tera et al. (1973 [2]) or Turner and Cadogan (1973 [5]), and Ryder (1990 [11]) cataclysm concepts, now began to resemble the pre-Apollo models—a product of the gradual decline of the early period of intense bombardment.

Even more varied definitions now became common when the terms "late heavy bombardment" or "LHB" were used. This shift in definitions is crucial to the evolution of the paradigm, since the original definitions of LHB explicitly suggested that many or most the prominent front-side multi-ring basins formed at ~4.0 to ~3.85 Ga ago, whereas the subtly evolving new definition of LHB allowed a much wider time-interval of basin formation.

Continuing the movement in this direction, the dynamical models now began a radical new phase. Around 2001, newer accretion models of planetary origin tended to show Uranus and Neptune forming last (accretion being slowed by their longer orbital periods). Levison et al. (2001 [12]), thus, expanded the dynamical model approach with a paper titled "Could the Lunar Late Heavy Bombardment Have Been Triggered by the Formation of Uranus and Neptune?" They proposed that the late formation of Uranus and Neptune led to scattering of outer Solar System planetesimals (having substantial ice content and extremely non-terrestrial, non-lunar ratios of at least some isotopes, including O, Cr, Ti, and Ni) into the inner Solar System, creating the late heavy bombardment. They defined the LHB with the original definition, as the period "roughly 4.0 to 3.8 Gyr ago" when "the lunar basins with known dates were formed." (Here, we must re-assert that the basin formation dates were, in fact, not "known.") The transport of Uranus–Neptune planetesimals, in the Levison et al. model, caused Jupiter and Saturn to migrate in position, destabilizing the Jupiter Trojans and the asteroid belt. This in turn also added rocky asteroidal material (with more Earth/Moon-like isotope ratios) into the mix of outer Solar System Earth/Moon impactors. Levison et al. (2001 [12]) suggested that main belt asteroids, "in principle" could have dominated the LHB. They added, however, that their model did not explain the delay in these events until 700 Ma after inner Solar System formation, and thus "must be viewed with some skepticism until formation models of Uranus and Neptune are available that are consistent with this late arrival" (of the LHB).

This revolutionary new dynamical paradigm, with giant planet formation history controlling the inner Solar System bombardment history, took firm root around 2005 with the introduction of the "Nice model" (associated with colleagues at Observatoire de la Côte d'Azur in Nice, France). The Nice model was described by a single group of authors in three back-to-back Letters in *Nature* (Morbidelli et al. 2005 [107]; Tsiganis et al. 2005; [108], Gomes et al. 2005 [109]). The authors found that a migration of giant planets in the early Solar System, associated with resonance effects, could trigger a "sudden massive delivery of planetesimals into the inner solar system" (Gomes et al. 2005 [109]). The authors explained clearly that the timing of this event was (as with Levison et al. 2001 [12]) unconstrained by the model, but noted that if they explicitly chose conditions to set the time of scattering at "700 million years after the planets formed" (Gomes et al. 2005 [109] abstract), they could account for the "Origin of the cataclysmic Late Heavy Bombardment period of the terrestrial planets" (Gomes et al. 2005 [109] title). They emphasized, with convincing effect, that the Nice model, for the first time, accounted for the cataclysm "within a self-consistent framework of solar system evolution" (from their abstract).

It is important to note that this early presentation of the Nice model was directly tied (in the Gomes et al. 2005 [109] paper) to its explanation of the classic terminal cataclysm—a brief (≲150 Ma-long) spike in cratering at 3.9 Ga ago, and *vice versa*. The Nice model supported the cataclysm paradigm; the cataclysm paradigm supported the model. The concluding paragraphs of Gomes et al. (2005 [109]) reinforce this with several points:

> *"Our results support a cataclysmic model for the lunar LHB . . . our simulations reproduce two of the main characteristics attributed to this episode: (1) the 700 Ma delay between the LHB and terrestrial planet formation, and (2) the overall intensity of lunar impacts. Our model predicts a sharp increase in the impact rate at the beginning of the LHB . . . Our model predicts the LHB lasted from between ~10 Ma and ~150 Ma."*

Certain earlier work seemed to support aspects of the basic Nice model concepts. For example, Wilkening (1977 [110]) had pointed out that carbonaceous chondrite clasts (thought to represent the low-albedo outer asteroid belt and outer Solar System objects) were common in polymict breccias meteorites, implying collisions of inner belt asteroids with outer Solar System objects. Also, Hartmann (1987 [111]; 1990 [112]) had suggested that a large flux of bodies must have been scattered from or among the outer Solar System objects, in order to explain the fact that all of the apparently captured satellites (both retrograde and prograde) in the Saturn, Jupiter, and Mars systems have extremely low albedos and spectral characteristics of outer Solar System interplanetary bodies such as Centaur and Trojan asteroids and comet nuclei—a fact that was being discovered just in the 1980s (cf. Cruikshank et al. 1985 [113]). The logic in the Hartmann papers was that while single capture events have low probability, the scattering of a large enough number of outer Solar System bodies would be required to insure the observed, modest number of captured "black" satellites. Based on pre-Nice-model empirical early work, then, scattered bodies from the outer Solar System moved inward at least as far as Mars. The pre-Nice suggestions, however, left open the question of when the scattering occurred, and whether the flux was sufficient in number and distribution to create the kind of *inner Solar System* spike in large impactors that would form many of the large, multi-ring basins 3.9 Ga ago, in order to fit the "classic" Earth/Moon terminal cataclysm.

Note that the language in Gomes et al. (2005 [109]) continued to apply the LHB term to a short-lived spike in the impact rate, not merely a long-decaying flux of planetesimals. The use of that concept to support the Nice model exemplifies the wide acceptance of the classic terminal cataclysm model in the 2000–2010 decade. The problem with this putative strength of the Nice model, however, was that the global lunar spike had already been refuted by the Cohen et al. (2000 [69]; 2005 [71]) data, which showed that regions far from Apollo sample-collection areas showed no spike at that time, hence, offering no evidence of the global 10 Ma to 150 Ma event proposed by Gomes et al. (2005 [109]). As an epistemological observation, therefore, we suggest that a weakness of the papers in that decade was a lack of full engagement of the various research communities with each other's actual data and methods. For example, most communities, including the dynamicists, were influenced by the title and venue of the initial Cohen et al. *Science* paper in 2000 [69], which asserted "Support for the Lunar Cataclysm Hypothesis . . . ," not to mention the Cohen et al.'s later papers' additional support for the cataclysm paradigm. This encouraged a widespread sense that the cataclysm had been confirmed by radiometric data and needed to be explained by dynamical data. A welcoming stage had thus been set for the Nice model, and of course the radiometric community was encouraged that the dynamicist community could "explain" the radiometric finding of a spike in impacts at 3.9 Ga ago. In the author's view, the paradigm was maintained in part because various communities thought that other communities had confirmed it. (A subtle sub-issue applicable to this period may be the competition among major journal editors to publish papers that appear to confirm or refute major paradigms, without adequate examination of the actual implications of the evidential data.)

As will be discussed in later sections, the dynamical models continued to evolve, leading to "Grand Tack" models in which Jupiter migrated through the belt into the inner Solar System, before reversing

course. These models, initially, continued the attempt to create theoretical dynamical explanations of the terminal cataclysm (and "LHB" under varying definitions of that term).

## 11. The Asteroids as Another Test Case, circa 2003–2013

The predictions of a solar-system-wide bombardment spike at ~3.9 Ga ago, as described by the Gomes et al. (2005 [109]) model, were tested by looking for a Ryder-like spike in impact phenomena among terrestrially collected meteorites, which are interpreted overwhelmingly as fragments of objects in the main asteroid belt. (Those with measured orbits typically have aphelia in, or close to, the main belt). The predictions were tested by measuring the age distribution of impact-related properties in these asteroidal meteorites. Examples of the results, shown in Figure 11, reveal neither a Ryder-like spike around 3.9 Ga ago, nor a "Ryder gap" (an absence of impact melts from before 4.0 Ga ago). Instead, meteorite impact melts and shock-altered ages typically show a gentle swell of impact phenomena. Among ordinary chondrites, the "swell" ranges from ~4.2 to ~3.5 Ga ago, with a poorly defined peak around 3.7 to 4.0 Ga ago, and in Vesta-related HED achondrites, a still broader swell from ~4.5 to ~3.4 Ga ago, with a possible shallow minimum at 4.2 to 4.3 Ga ago and a peak round 3.7 Ga ago. None of these data show evidence of the classic terminal cataclysm/LHB phenomenon, i.e., an extraordinary Ryder-like spike at 3.8–4.0 Ga, or "Ryder gap" in impacts from ~4.4 to ~4.0 Ga.

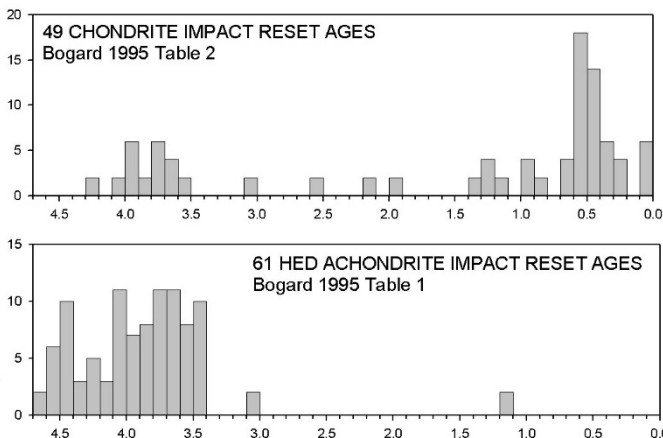

**Figure 11.** Age distributions of asteroidal meteorite ages for 49 chondrite and 61 HED achondrite ages, reported by Bogard (1995 [114]). The data show no sharp spike at 4.0 to 3.8 Ga ago, but rather an extended impact history expending from ~4.5 or ~4.3 Ga ago to ~3.5 Ga ago. The spike in chondrite ages near 0.5 Ga ago is related to a now-known impact fragmentation or cratering event on the L chondrite parent body, dated at 470 Ma ago (Schmitz et al. 2003 [115]; Trieloff et al. 2006 [116]).

In a more sophisticated search among the asteroid cratering record, Cohen (2013 [117]) made age measurements among impact melt clasts in the H (howardite) chondrite meteorites, considered to be candidates for origin on Vesta (or on Vesta-like parent bodies; cf. Cruikshank et al. 1991 [118]). As the second largest asteroid (at ~525 km diameter), Vesta is a good candidate for preserving a record of early cratering in the asteroid belt. As shown in Figure 12, Cohen's reported ages for twelve clasts in howardites (her Figure 7A) showed no anomalous spike at 4.0 to 3.85 Ga ago, and indicated that most impact related samples date from *more recently than* 3.9 Ga ago, with the 12 samples ranging in impact plateau age from 3.74 ± 0.05 Ga to 3.21 ± 0.08 Ga, with a roughly defined peak at about 3.70–3.74 Ga ago, and with one outlier plateau age of 4.04 ± 0.80 Ga. Interestingly, *none* of her 12 samples' ages fell in the range of the putative cataclysm (i.e., 4.0 to 3.85 Ga ago). Cohen concluded that the results indicate "a scenario ... similar to the late heavy bombardment of the Moon," even though the peak in impact-related ages occurs in a period *after* the 3.85 Ga cessation of the "classic" lunar terminal cataclysm by ~110 Ma to ~640 Ma. Others (including an unnamed reviewer of this manuscript) have argued that the Vesta impact data apply to smaller impacts than the lunar impact

melt data. However, nature does not allow us to imply effects of more or fewer small-scale impacts without considering the effects at the other end of the impactor size distribution at the same time. Applying Ryder's rule, Cohen concluded that " ... howardite impact-melt clast ages reinforce the notion of a dynamically unusual episode of bombardment across the inner solar system beginning at around 4.0 Ga ago." Yet the basic problem remains: How can we have an interplanetary bombardment of the inner Solar Systems at 3.9 Ga ago (especially if we invoke giant planets moving through the asteroid belt), and yet have zero indication of any anomaly in Vesta samples between 3.85 and 4.0 Ga ago? An alternative interpretation of these data could legitimately emphasize that they show no evidence of a unique cataclysmic bombardment on the HED parent body (or bodies) during putative lunar terminal cataclysm (which, according to the Nice model (Gomes et al. 2005 [109]) and other recent dynamical models, was supposed to have affected the whole inner Solar System).

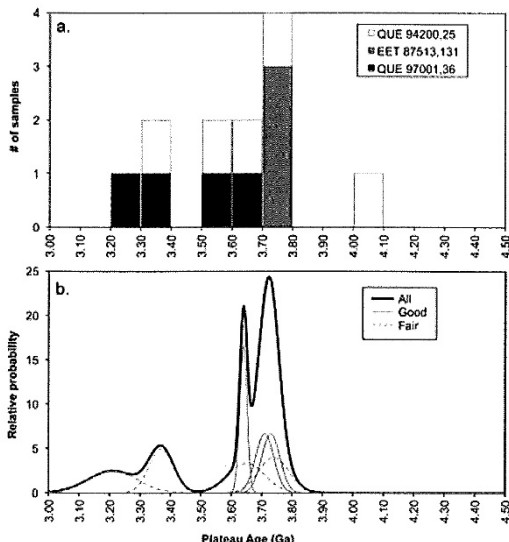

**Figure 12.** Age distribution of 12 impact melt clasts in H chondrite meteorites, believed to come originally from asteroid Vesta. The data show no measured ages between 3.8 and 4.0 Ga ago, the date for the "classic" terminal cataclysm. Lower figure shows ideogram for data judged "good" or "fair." Note that time flows from right to left, contrary to most of our diagrams. (Cohen 2013 [117]).

Taking all the above asteroid data together, we could describe them as showing a gradual, smeared-out swell in *surviving* impact-related asteroidal meteorites with a "soft" maximum that is later than 3.9 Ga—more like 3.8 to 3.6 Ga ago, and with measurable differences between one meteorite class and another. No Ryder-like, 150 Ma spike or classic terminal cataclysm starting at 4.0 or 3.9 Ga ago, as discussed by Turner and Cadogan (1973 [5]), Tera et al. (1973 [2]; 1974 [3,4]), Ryder (1990 [11]), and Gomes et al. (2005 [109]), is evident in the asteroid data. The asteroidal impact data also offer at least some indicators of an elevated impact history among asteroids before 4.0 Ga ago (Bogard 1995 [114], Hartmann 2003 [52]). A plausible conclusion is that pre-4.0 Ga impacts did occur in the belt, but that survival of evidentiary samples was controlled by stochastic differences in impact history of various individual parent bodies, plus different geologic and impact conditions in various parts of the asteroid belt, as well as on the lunar surface (see Section 16).

If a spike-like terminal impact cataclysm occurred on the Moon, in which most impacts and basin formation occurred between 3.85 and 4.0 Ga ago, then the asteroid belt did not experience it. That leads, in turn, to a serious problem with any dynamical models that attribute a lunar cataclysm at 3.9 Ga ago to scattering of impactors from the outer Solar System, and especially a scenario of Jupiter migrating through the belt after, say, 4.2 Ga ago. The resulting problem for our planetary science community is significant. First, is there adequate *empirical* evidence to support a lack of impacts in the inner Solar System before 4.0 Ga ago, as opposed to an early intense bombardment that, on average, declined with

lengthening half-life during the first 300–900 Ma? Second, if we accept that the empirical evidence refutes a unique global lunar cataclysmic bombardment at 3.9 Ga ago, then the lunar history paradigm as derived from the radiometric/petrologic approach—that a half dozen or more of the most prominent basin impact occurred within ~150–200 Ma—is probably wrong. Third, it appears that the constraint on the origin of life is not a special, brief cataclysm at 3.9 Ga ago, but rather the random, scattered, last large impacts occurring individually on Earth (with similar, separate events on the Moon).

Contrary to the above ideas, the Gomes et al. (2005 [109]) advocacy for the Nice model emphasized a disruption of the belt at 3.9 Ga:

> *"The asteroid belt was also strongly perturbed, with [asteroids] supplying a significant fraction of the LHB impactors . . . Our model predicts that the asteroid belt was depleted by a factor of ~10 during the LHB."*

Has the asteroid belt really been so disrupted? The answer is unclear. As early as 1982, Gradie and Tedesco (1982 [119]) noted that the asteroid main belt preserves a first-order zonal structure, based on different spectroscopic classes of observed asteroids. Continued observations at Mauna Kea Observatory in the 1980s (e.g., Hartmann et al. 1982 [120], Cruikshank et al. 1985 [113]) supported this. For example, if we go beyond Mars (semi-major 1.5 AU), we encounter a sparse group of asteroids spectroscopically classed as "E-type," congregating with semi-major axes ~1.8 to ~2.0 AU. The more populated part of the belt has a fairly sharply defined inner edge at ~2.2 AU. The common class S asteroids (mostly ordinary chondrites?) dominate from about 2.2 to 2.7 AU. Beyond about 2.7 AU, roughly the midpoint of the belt, the albedo of asteroids drops dramatically in the outer belt. The spectroscopic classifications in the outer belt resemble a broader range of low-albedo objects throughout the outer Solar System, believed to have some relationship to carbonaceous chondrite meteorites. There is some tendency toward redder colors (i.e., very dark, reddish brown colors) as we move farther out in the outer Solar System.

At the very least, this surviving, crude zonal structure of the main belt, manifested by spectral types, constrains any plausible history of chaotic disruption of the asteroid belt. It is not clear that the modern dynamical models meet such constraints or explain completely how Jupiter could pass first inward, then outward, through the belt without destroying the crude zonal structure—or, alternatively, whether there is a mechanism to establish the observed zonal structure after such events.

Dynamicists had already been looking at the crude zonal structure. Wetherill (1977 [8,9]), in his early dynamical models, had already suggested that some of the planetesimals that formed near Earth at ~1 AU were scattered by close encounters with growing Earth, ending up interacting with Mars, thus creating a small fraction of "1 A.U. asteroids" being preserved on the inner edge of the asteroid main belt. Wetherill's Monte Carlo calculations included 41 planetesimals that ended up in the asteroid belt with semi-major axes inside 3 AU, of which three were relatively isolated objects on the inner edge of the belt, with semi-major axes ≲2.0 AU and inclinations between 10° and 20°. Wetherill (1977b [9], caption of his Figure 5) asserted that these bodies were "calculated to survive for the age of the solar system," and he noted (his Figure 6) that this was the exact region in which the Hungaria "family" of asteroids is located toda y.

Within a few years after Wetherill's work, the Hungaria asteroids were found to comprise the spectroscopic class labeled E in some of the early asteroid classification systems—and this E class matched the spectral properties of enstatite achondrite meteorites (Tholen and Barucci 1989 [121]; Gradie et al. 1989 [122]; Clark et al. 2004 [123]). A diagram by Gradie et al. (1989 [122], their Figure 1) showed a "map" of asteroid spectroscopic classes, with a peak in E + R classes comprising ~70% of the asteroids at semi-major axis *a* < 2 AU (R being a "rare" class, and S class objects, thought to be ordinary chondrites, comprising ~30%). Interestingly, the enstatite meteorites (both achondrite and chondrite) have isotope ratios closer to the Earth/Moon composition than any other meteorite class, supporting Wetherill's suggestion that these E asteroids may have been scattered into the innermost belt from 1 AU. A more recent study of E asteroids by Clark et al. (2004 [123]), adding

new infrared spectroscopic observations to previous data, suggests three sub-classes with somewhat different mineral mixtures. Clark proposed that one sub-class matched enstatite achondrite ("aubrite") meteorites and the Hungaria asteroids "within 3%," but others possibly representing enstatitic objects not found in meteorite collections.

The key property of enstatite meteorites in our context is their isotopic composition, which appears to relate them to Earth and Moon, and hence to 1-AU planetesimals; in terms of direct sample-return missions from the E asteroid classes, however, their isotope compositions are, of course, unknown (unless they are the source of enstatite meteorites). There is some geochemical evidence that the basin-forming impactors resembled enstatite meteorites, however. For example, among a group of impact melt breccias from the Apollo 17 site, Norman et al. (2002 [124]) cite "at least two separate impact events." They state that the:

> *"KREEP-rich melt breccias representing the Apollo 17 poikilitic suite are enriched in highly siderophile elements (3.6–15.8 ppb Ir) with CI-normalized patterns that are elevated in Re, Ru and Pd relative to Ir and Pt. The restricted range of lithophile element compositions combined with the coherent siderophile element signatures indicate formation of these breccias in a single impact event involving an EH [high enstatite] chondrite asteroid . . . probably as melt sheet deposits from the Serenitatis Basin. One exceptional sample, a split from melt breccia, has a distinctive lithophile element composition and a siderophile element signature more like that of ordinary chondrites, indicating a discrete impact event . . . "*

They go on to say that their data are "more consistent with a late cataclysm than a smoothly declining accretionary flux," but they note the possibility of "Late accretion of enstatite chondrites during a 3.8–4.0 Ga cataclysm . . . "

Becker et al. (2006 [125]) presented geochemical evidence suggesting that the objects creating lunar basins had isotopic properties of high-siderophile enstatite chondrites. They discuss 3.89 Ga-old "impact melt breccias" from Apollo 17 and say that their data on HSE (highly siderophile elements) are "suggesting they represent binary mixtures of a high-HSE component, presumably a LL or EH chondrite-like impactor." Interestingly, Bottke et al. (2010 [13]) cited Becker et al. (2006 [125]) when Bottke et al. stated that "HSE studies suggest the extraterrestrial material mixed into Earth's primitive mantle after the Moon-forming impact . . . was dominated by enstatite chondrite-like projectiles."

Similarly, Joy et al. (2012 [126]; 2012 [127]) found evidence of enstatite-like materials in fragments of basin-forming impactors. They analyzed 20–225 μm-scale particles in "regolith breccias" from the Apollo 16 site, "consolidated into rocks between ~3.8 and 3.4 Ga," i.e., after the Imbrium impact and assumed to be related to Imbrium ejecta. They wrote (2012 [127], p. 1427):

> *"Oxygen isotope compositions of [the mafic] fragments are within the ranges exhibited by ordinary chondrite and CC chondrite [but because] of the large uncertainties associated with analyses of such small particles . . . the error bars statistically overlap the terrestrial fractionation line . . . and compositions of lunar rocks, achondritic meteorites, and enstatite chondrites. The Mg-rich [mafic fragment] compositions imply that those isotope compositions were established in a region of the solar nebula with near-solar redox conditions, similar to carbonaceous and enstatite chondrites . . . "*

To return to the crude zonal structure of the asteroid belt, later dynamical models, in the "Grand Tack" category, have proposed a possible explanation for zonal structure in spite of giant planet migration. For example, Walsh, Morbidelli, and three others (2011 [128]) describe Jupiter migrating inward as far as 1.5 AU from the Sun, during which it "initially empties but then repopulates the asteroid belt . . . " In this view, a crude zonal structure is created when a number of "bodies originating between 1 and 3 AU" end up surviving, in Wetherill-like fashion, primarily in the innermost edge of the main asteroid belt, while "bodies originating between and beyond the giant planets" are plastered onto the outer part of the belt. This would explain the very low-albedo outer Solar System carbonaceous objects, with their extremely non-terrestrial isotope ratios, which populate the outer belt, "butting up"

against the moderate albedo inner belt asteroids (ordinary chondritic and achondritic bodies, with closer-to-terrestrial isotope ratios).

A few more comments on the E asteroids and the Hungaria "family" in the innermost belt are relevant. Such families are believed to be fragments of single ancient asteroids, broken apart by an energetic collision. Here, however, we have placed "family" in quotes, because Clark et al. (2004 [123]), reviewing the literature on the Hungarias, stated that " . . . the consensus seems to be that the Hungaria asteroids are not a dynamical family." If they are not a single dynamical family, that would support the model by Wetherill (1977 [8,9]) in which 1-AU E asteroids could have implanted into the innermost part of the asteroid belt by individual scattering events. Clark et al. (2004 [123]) list the 23 known E-class asteroids and stated that they divide into two groups, the classic "Hungaria" with semi-major axes from 1.88 to 1.98 AU and the others, in the region from 2.10 to 2.72 AU.

## 12. Pre-4.0 Impact Melts in Upland Breccias, circa 2008–Present

Lurking behind the problems discussed in the above Sections is the assumption that we refer to as Ryder's rule: lack of observed impact melts = lack of impacts. Since the beginning of the terminal cataclysm/LHB paradigm, this assumption has neglected the possibility that a pre-4.0 Ga destructive mechanism (such as early intense cratering from lunar formation to 4.0 Ga ago, along the lines of Hartung 1974 [46] and Hartmann 1975 [34]) may have preferentially destroyed most of the pre-4.0 Ga impact melt samples (while still allowing numbers of early crustal, pre-4.0 Ga plutonic rocks to be collected on the lunar surface, as will be further discussed in Section 16).

Logically, the absence of impact melts from before 4.0 Ga does not describe any event at 3.9 Ga, nor does it require absence of impacts before 4.0 Ga. Rather, it demonstrates only *lack of survival* of impact melts from before 4.0 Ga (due either to lack of impact events in the first place, as Ryder assumed, or to preferential destruction of the oldest impact melts). Ryder's rule is currently being challenged by numbers of reported detections of rock ages (including some lunar impact melts) from that period, especially in clusters of reported ages around 4.21 Ga and 4.33 Ga, among materials in upland breccias. Interpretations of the meaning of such data were not always consistent. Section 9 already touched briefly on ages of ~4.2 Ga for certain Apollo 16 samples (including melt rocks) near Nectaris. A detailed discussion of the recent data is beyond the scope of this paper, but examples include:

- Fernandes et al. (2008 [129]) found a "4.2 Ga impact age in samples from Apollo 16 and 17" (their title), and emphasized that the recurrence of the ~4.2 age suggested major impacts did occur before Ga ago. They conclude " . . . it is suitable to say that a significant impact event(s) occurred at ~4.2 Ga . . . ."

- Nemchin et al. (2008 [130]) studied 105 zircons from Apollo 14 and 17 breccias, interpreting U–Pb ages as referring to magmatic events, but possibly initiated by impacts. Of these zircons, they found (their p. 685) that "only a few are *younger* than 4.00 Ga [and] only one has an age of ca. 3.85 Ga" (around the time of the putative cataclysm). Instead, at *both sites* they found strong peaks at 4.33 to 4.35 Ga (involving about 40 of the samples), and also possibly 4.2 Ga (their Figure 10). They proposed that large impacts at those two times triggered significant pulses of KREEP magmatism. Their interpretation favors a model with high cratering rates before 4.0 Ga, rather than very low rates as in the classic Ryder cataclysm model.

- Pidgeon et al. (2010 [131]) referred to the U–Pb zircon ages and discussed the strong peak in ages 4.338 ± 0.005 Ga among Apollo 14 samples and 4.341 ± 0.003 Ga among Apollo 17 samples. They concluded that the Moon experienced "a massive impact . . . at ~4.34 Ga," and that this impact "was possibly the largest in lunar history since crystallization of the magma ocean".

- Based on certain Apollo 16 sample ages ~4.2 Ga, Norman and Nemchin (2012 [102]) stated, "Crystallization ages of lunar impact melt rocks provide the primary evidence for a spike in the impact flux at ~3.9 Ga. Here, we report U–Pb isotopic ages of accessory phases (apatite, zirconolite) in lunar melt breccia 67,955 that confirm a crystallization age of ~4.2 Ga and reveal a younger

overprint possibly related to entrainment of the breccia by one or more younger basins such as Imbrium".

- Grange et al. (2013 [132]) presented a distribution of U–Pb ages of lunar zircon grains from Apollo 12, 14, 15, and 17 landing sites, confirming the largest peak at about 4.33 Ga ago, but with a second-largest peak at 4.20 Ga ago, and a third largest peak at ~4.23 Ga ago, suggesting these may all represent basin-scale impacts. "If this is true . . . the meteorite flux that affected the early moon . . . lasted from about 4350 Ma to 3900 Ma, with a decrease in either size or frequency of impacts along this time interval. No major impact able to produce a melt sheet [in which zircons formed] took place after 3900 Ma".

- Norman et al. (2015 [103]) argued that the Apollo 16 clasts with ~4.2 Ga ages have chemical and other affinities with the Imbrium region of the Moon, not the Nectaris (Apollo 16) region. They affirm that the 4.2 Ga age represents a basin-forming impact, but they suggest that the source, instead of being Nectaris, is a pre-Imbrium basin formed near the Imbrium impact site but then obliterated by the Imbrium impact. In this view, Imbrium ejecta, collected at the Apollo 16 site, entrained impact melts from the earlier basin. As we have already discussed (see Section 7 and Hartmann 2003 [52], p. 589ff), Imbrium is so big that impact melt fragments from pre-Imbrium basins in that location could plausibly have been excavated by the Imbrium impact.

Note that if the 3.9–4.0 Ga regions of the lunar highland surface are near-saturated or super-saturated with impact craters, and if the impact rate was even higher before 4.0–4.1 Ga than later, then we would expect pre-4.1 Ga impact melt samples, which formed in the near-surface layers, to be reduced to small clasts. This is just what is currently being observed (see further discussion in Section 16). The accumulating evidence of pre-4.0 Ga, impact-melt-producing large impacts, suggests, at a minimum, that the "classic" terminal cataclysm/LHB paradigm, with no large impacts before 4.0 Ga ago, must be reassessed.

## 13. Biological, Geological, and Meteoritical Issues, circa 1988–present

A danger with the spread of a questionable, occasionally redefined concept such as LHB (and a motivation for the present study) is that even as it declines in acceptance in its parent field, it may remain a paradigm being cited as a constraint in other fields. This is true in the present case. The spike in impacts at 3.9 Ga ago continues to be widely invoked as a factor in the origin of life on Earth, as well also a factor in terrestrial geology. Here are examples:

- Maher and Stevenson (1988 [133], even before Ryder's "strong form" of the terminal cataclysm paradigm) spoke of the "impact frustration of the origin of life" (their title, a phrase later widely adopted). They placed the earliest opportunity for biogenesis on land at 4.0 to 3.7 Ga ago.

- In their 2000 paper, Cohen, Swindle, and Kring [69], in asserting the isotopic "support for the lunar cataclysm" at 3.9 Ga, noted that "Coincidentally, the earliest isotopic evidence of life on Earth is also ~3.9 Ga. If a swarm of impactors at 3.9 Ga returned the surface of Earth to a hot and energetic state, the rise or evolution of life on Earth would have been affected."

- Levison et al. (2001 [12]), in developing the planet migration idea, stated that "Certainly, the end of the LHB marks the beginning of the epoch when the sustained origins of life became possible on the earth." This could be true whether "LHB" is defined as a single spike after ~500 Ma of low impact, or, as in more recent usage, merely the tail end of a still more intense early bombardment. This comment ignores the idea that if many basins, and much of the upland cratering, pre-dated 4.0 Ga, the environment at 3.9 Ga ago could have been much more benign than in the classic "single spike" terminal cataclysm model.

- Koeberl (2006 [134]), in a major review of "The record of impact processes on the early Earth" (his title), accepted "convincing evidence that the Moon experienced an interval of intense bombardment with a maximum at ca. 3.85 ± 0.05 Ga," and concluded "The consequences for the

Earth must have been devastating, although the exact consequences are the subject of debate . . . So far, no unequivocal record of a late heavy bombardment on the early Earth has been found".

- Ćuk (2012 [135]), in the first sentences of an abstract, states "The Moon has suffered intense impact bombardment ending at 3.9 Gyr ago, and this bombardment probably affected all of the inner solar system," and comments " . . . the last episode of bombardment at about 3.85 Gyr ago," while stating that it "was less extensive than previously thought".

- Perkins (2014 [136]), writing in *Science*, reviewed recent Czech experiments indicating that conditions during cosmic impacts can transform simple precursor materials into then nucleobases in RNA, stating that "During a period aptly dubbed the Late Heavy Bombardment, which began about 4 billion years ago and lasted some 150 million years, large objects pummeled our planet and Moon, as well as Mercury, Venus, and Mars." The article quotes an astrobiologist commenting on the Czech work, saying that it " . . . nicely correlates the Late Heavy Bombardment and the energy it delivered to Earth about 4 billion years ago with the formation of RNA and DNA nucleobases . . . ".

- Spray (2016 [57]), writing in the *Annual Review of Earth and Planetary Sciences* about the important issue of regolith lithification mechanisms, presented the "Late Heavy Bombardment" as a known "period between 4.1 and 3.8 Ga, when at least the inner solar system underwent enhanced bombardment by asteroids and comets" (his p. 142).

- Li and Hsu (2018 [137]) introduced a study of impact history of the L chondrite parent body by stating "Impact rates were much higher in the early stages of solar system evolution, as recorded by the well-known Moon-forming giant impact at 4.4–4.5 Ga . . . and the Late Heavy Bombardment (~3.8–4.2 Ga)." Regarding the LHB statement, they cited Kring and Cohen (2002 [72]) and Marchi et al. (2012 [138]).

- Lowe and Byerly (2018 [139]) reviewed "The terrestrial record of Late Heavy Bombardment" (their title and capitalization), giving a valuable overview of earlier terrestrial data on impact spherule beds, offering a " . . . terrestrial record of large impacts from at least 3.47 to 3.22 Ga and from 2.63 to 2.49 Ga." They concluded that " . . . high impact rates either continuous or as impact clusters persisted until at least the close of the Archean at 2.5 Ga." As will be discussed further in Section 17, Bottke et al. (2012 [140]) and Bottke and Norman (2017 [141]) had already also cited the existence of these clusters, and extended the LHB definition to include a gradually declining impact flux as late as 2.5 Ga ago. Picking up on that semantic lead from the planetary community, Lowe and Byerly (2018 [139]) supported a long-extended "LHB," stating that " . . . between about 3.47 and 3.22 billion years ago, the Earth was struck by at least 8 and probably many more large asteroids between 20 and 50 km in diameter and some perhaps much larger. These objects were part of a continuing rain of extraterrestrial objects striking the Earth that commenced at the start of the Late Heavy Bombardment some time before 4.0 Ga, gradually declined over the next billion and a half years, and ended sometime after 2.5 Ga." From the point of view of the present paper, that last sentence is ironic, because their description of LHB is, in fact, precisely contrary to the original definition of LHB.

To summarize this section, there is cause for concern if biologists, terrestrial geologists, and other scientists use the terminal cataclysm and "Late Heavy Bombardment" paradigm as a known constraint on terrestrial geology, terrestrial life's origins, and asteroid cratering. The concern is that this usage of "terminal cataclysm" and "LHB" happened at the very time the planetary science community, which originated the idea, was pushing it "off the table." Here we see that established paradigms have a ripple effect and may take years to fade out, even while rejected by workers within the original field of research.

On the positive side, there are grounds for a new synthesis. For example:

- Sleep et al. (1989 [142]) pointed out that terrestrial impacts of basin-forming scale could have boiled away large fractions of ocean water on primordial Earth. Thus, it was possible to suggest

that instead of being concentrated in one cataclysmic LHB destructive era at 3.9 Ga, causing a one-time wipe-out of earlier biochemical "experiments" dating from the 400 Ma "lull" in impacts, cycles of destruction, and biogenesis may have happened many times. There may have been many independent terrestrial developments and destructions of molecular proto-life forms, dependent on the half-life of decline in the early impact rate and the stochastic intervals between pre-4.0 Ga large impacts versus the unknown timescale for development of life forms under more ideal conditions.

- Ryder (2002 [143]) re-evaluated the mass flux and effects of the terminal cataclysm on Earth during the putative cataclysm. He concluded that such effects were inadequate to destroy biological activity and require a re-booting of the origin of life: To cite his abstract, " . . . there is no justification for the claim that life originated (or re-originated) as late as 3.85 Ga in response to the end of hostile impact conditions".

- Similarly, Lowe and Byerly (2018 [139] p. 58) estimate, from the terrestrial spherule beds, cratering rates in the interval 3.47 to 3.22 Ga, as being ~32 to as much as ~320 times the current rate or perhaps higher. The estimates are crude, depending on the interpretations of how many different impacts are being detected and whether the Archean impacts occurred in concentrated spikes. They are not, however, out of line with lunar data. From the similarly crude lunar Apollo site curve of cratering versus sample age, Hartmann (1972 [33], Figure 5) estimated the cratering rate in that same interval as descending from 20 to about 8 times the current rate. Neukum et al. (2001 [38] Figure 11) show a similar result descending from about 13 to 4 times the current rate in that period. While the Neukum curve of declining flux showed a virtually constant cratering rate after ~3.0 Ga ago, papers by Quantin et al. (2007 [40]) and Hartmann et al. (2007 [42]) gave supporting evidence that the early lunar and terrestrial impact rate was still declining after 3.0 Ga ago.

Perhaps, then, we are moving toward a more direct understanding of observed rates of decline in impact rates in the Earth–Moon system, i.e., without forcing the empirical discussion into semantic paradigm boxes such as "terminal cataclysm" or "LHB." Support for two research areas would greatly improve our understanding of the Earth–Moon system, and ultimately planetary systems in general: (1) further analysis of the terrestrial impact spherule beds and (2) lunar sample return from the reportedly youngest lava flows on the Moon. Such studies would greatly improve the calibration of the crater chronometry system in the inner Solar System.

## 14. Doubts about the "Classic Cataclysm," circa 2003–2014

As a result of the history and issues sketched here, doubts about the four-decades-old "classic" paradigm terminal cataclysm/LHB, 3.9 Ga ago, began to surface more commonly by the early 2000s.

- Hartmann (2003 [52]) pointed out the inconsistencies of the classic cataclysm/LHB models with impact-related lunar meteorite and asteroidal data, concluding that "The hypothesis that a lunar or inner-solar-system-wide cratering cataclysm occurred ~3.9 Ga ago is not established, and comparison of Apollo/Luna samples, lunar meteorites, and asteroidal meteorites does not give compelling evidence for it".

- Norman et al. (2007 [144]) presented evidence for a 4.2 Ga age of Apollo melt rock, noting that "earlier discussions of the lunar impact record emphasizing the lack of events significantly older than 3.9 Ga may need to be revised".

- Norman (2009 [145]), under the title "The Lunar Cataclysm: Reality or 'Mythconception,'" described the "cataclysm hypothesis" as "far from proven".

- Lineweaver (2010 [146]), under the title "Crater-counting Evidence against the Late Heavy Bombardment Hypothesis" concluded "Our analysis does not support the LHB hypothesis as articulated by Ryder, nor do we find the data from impact breccias, glass spherules or lunar meteorites supportive of LHB".

- Ćuk et al. (2010 [147]), as cited again by Ćuk (2012 [135]), argued that "the record of the cataclysm does not match the predictions of the planetary migration-based models." Ćuk (2012 [135]) argued that the Imbrium and Orientale impactors were too big to fit into the previous population of impactors and were produced by "a non-asteroidal impactor population, while planetary migration predicts the late impactors to be derived from main-belt asteroids (Gomes et al. 2005) [109]." The latter reference was invoking the Nice model.

- Hartmann (2012 [148]) re-emphasized that accumulating empirical data conflicted with the "classic" LHB model, noting also that the Orientale impactor might have been a satellite of the Imbrium impactor (allowing near-simultaneous formation; see discussion in next section), and that the distinctive feature of the era 4.1–3.8 Ga ago was that before that interval, the cratering rate across the whole size distribution was so high that rocks created or placed in lunar near-surface layers had much reduced chance of surviving intact to be collected or ejected to Earth today, but after that interval, survival was more common.

- Morbidelli, Marchi, Bottke, and Kring (2012 [149]) pursued dynamical models that could allow intense cratering in the pre-4.0 Ga era, and concluded "We deduce that . . . the impact flux did not decline exponentially over the first billion years . . . but also there was no prominent and narrow impact spike ~3.9 Gy ago . . . [Also], the timeline of the lunar bombardment has a sawtooth-like profile, with an uptick . . . near ~4.1 Gy ago . . . "

- Fernandes et al. (2013 [150]) reported $^{40}$Ar–$^{39}$Ar ages including an Apollo 16 breccia with impact melt dating from 4.293 ± 0.044 Ga. Combining their results with those of other authors, they advocated a complex bombardment history with generally declining flux throughout the first 600 Ma.

- Merle et al. (2013 [151]) reported U–Pb ages of zircons in samples interpreted as excavated from Cone Crater in the Fra Mauro Imbrium ejecta blanket at the Apollo 14 site. These samples included have not only the youngest ages of 3.900 ± 0.027 Ga and 3.932 ± 0.023 Ga (Imbrium ages), but also larger spikes at ~4.21 and ~4.34 Ga from material underlying the Imbrium layer.

- Norman and Nemchin (2014 [152]) described an array of radiometric ages (some from impact melts) around 4.1–4.3 Ga and stated that "The U–Pb isotopic compositions of zirconolite and apatite in lunar melt rock 67,955 establish a basin scale impact melting event on the Moon at 4.2 Ga followed by entrainment of the melt rock in . . . ejecta [~300 Ma] later . . . The ages of lunar zircons and metamorphosed breccias also suggest that multiple large impacts occurred on the Moon at around [4.1–4.2 Ga ago], but the sizes of those events is difficult to establish...The strong version of the late cataclysm hypothesis in which all of the lunar basins formed between 3.8 and 4.0 G . . . appears untenable.@

## 15. Evolving Dynamical Models (and the Issue of Large Basins), 2001–Present

Due in part to such developments, one might fairly say that the dynamical models have retreated from the 150 Ma-long impact spike scenario that had been putatively "explained" by the Nice model (Gomes et al. 2005 [109]). More diverse models appeared. It is useful to divide these newer dynamical efforts into two subjects: (1) the time behavior of impact flux *after* 3.9–4.0 Ga ago, and (2) the time behavior of impact *before* 3.9–4.0 Ga. During the 2000s, dynamical models began extending the period of intense cratering in both directions, responding to doubts about the brief impact spike.

Morbidelli et al. (2001 [106]), as mentioned in Section 9, looked for dynamical mechanics that could provide "A plausible cause of the late heavy bombardment" (their title) by which they meant not specifically the classic Ryder-type spike, but, more generally, the population of leftover planetesimals after planet formation. These were seen as objects left on high inclination orbits in the inner Solar System, so that "the final depletion of this leftover population would cause an extended bombardment of all the terrestrial planets, slowly decaying with a timescale of the order of 60 Ma" (their abstract).

In order-of-magnitude terms, the Morbidelli et al. (2001 [106]) model fit what had been shown to exist in the lunar record after ~3.8 Ga ago, namely, the declining flux based on Apollo rock ages

at various landing sites, and declining cratering rates (Hartmann 1970a [31]; 1970b [32]; 1972 [33]; 1980 [34]; Hartmann and Wood 1971 [20]; Neukum 1983 [36]). Section 5 mentioned various estimates of the impact decay rate and half-lives during different time intervals. As examples, we have the following estimates of half-lives of decline in the cratering rate, taken in some cases from graphs in the referenced publications; these are mostly early, but we include similar recent results. There is scatter in the early results, but the suggestions of behavior of the impact flux decay are contrary to the terminal cataclysm paradigm, and in the same ballpark as the more sophisticated theoretical estimates of decay rate by Morbidelli et al. (2001 [106]) and others in the 2000s. For example:

- ~4.4 Ga ago (post-accretion impact rate, planetesimals being scattered into higher eccentricities and inclinations: half-life ~20 Ma (Hartmann 1980 [34], Figure 2)
- Before 4.1 Ga ago: average half-life ~80 Ma (Hartmann, 1972 [33]; 1980 [34])
- Before 4.1 Ga ago: half-life of a Mars-crosser population (calculated as a possible dominant impactor population: 80 Ma (Ćuk 2012 [135] Figure 5 plus text)
- ~4 Ga ago: half-life of decline ~70 Ma (Hartung 1974 [46])
- ~3.9 Ga ago (during Imbrium impact): half-life ~100 Ma (Neukum 1983 [36])
- ~3.8 to 3.0 Ga ago: average half-life ~300 Ma (Hartmann 1972 [33])
- ~3.3 Ga ago: half-life ~350 Ma (Neukum 1983 [36] equation)
- After ~3.4 Ga ago: half-life of "proto-Hungaria" population of impactors: 600 Ma (Ćuk 2012 [135] Figure 5 plus text)

Morbidelli et al. (2001 [106]) pointed out that their dynamical model scenario of a leftover population of planet-forming asteroids and comet nuclei would not explain a classic cataclysmic bombardment at 3.9 Ga ago, because that would require too high an initial mass of these planetesimals:

> *"If a terminal lunar cataclysm (a spike in the crater record ~3.9 Ga ago) really occurred on the moon, it was not caused by the highly-inclined leftover population . . . "*

The Morbidelli et al. (2001 [106]) work appeared to provide a plausible alternative to the cataclysm paradigm: in the absence of cataclysm, a population of highly inclined, leftover, 1–2 AU planetesimals that give essentially the decay that is actually observed from ~3.8 to ~3.2 Ga ago, but without the 3.9 Ga spike. The cratering rate prior to 3.9 Ga was left undetermined, but could have been still higher, instead of negligible as in the "classic" cataclysm/LHB scenario.

Other dynamicists attempted to salvage the cataclysm/LHB concept by two approaches: (1) seeking models that reduced the intensity of spike in the impact rate, and/or lengthened its duration, and/or by (2) redefining earlier terms to maintain the phrase "late heavy bombardment," but applying it to other scenarios. This redefinition was not intentionally obfuscatory, but rather reflected a view within the modelling community that "LHB" was a merely a term referring to whatever was happening around 3.9 Ga, as estimated from the latest models (even if it was a less "heavy" bombardment than the preceding bombardment and was less clearly "late"). The new models typically retained the possibility of late, outer Solar System planet migration (as being suggested by contemporaneous observations of extra-solar planetary systems). They also reduced the impact spike to a more and more gradual surge, spread out during several hundred million years.

Papers by Bottke et al. (2010 [13]; 2011 [153], abstracts; and 2012 [140] *Nature* paper) demonstrate how the new results were allowed by reviewers, editors, and journals to be referred to as "LHB" in spite of the disconnect with the original term.

As mentioned in earlier sections, Bottke et al. (2010 [13]) began with essentially the "classic" definition of terminal cataclysm and LHB, this time lasting 210 Ma:

> *"The Late Heavy Bombardment (LHB) is defined as a period 3.96–3.75 Ga when many lunar basins (e.g., Serenitatis, Imbrium) and impact melts were produced."*

Bottke et al. (2010 [13]) then pointed out that the giant planet migration would cause the $v_6$ resonance in the asteroid belt to move, changing the position of the inner edge of the belt. They introduced the term E-belt to refer to a hypothetical primordial portion of the innermost belt extending inward toward Mars, which would allow the period of bombardment to be the stated ~210 Ma.

Bottke et al. (2011 [153]) then refined that model, examining how giant planet migration would push the $v_6$ resonance across the asteroid belt. This work allowed the LHB to start at least another 120 Ma earlier than in previous models, and allowed for an extended tail. Bottke et al. (2011 [153]) thus proposed that with their new model, the redefined LHB would have lasted ~370 Ma to ~430 Ma (or 400 Ma, as they rounded it off):

> " . . . the depletion of the E-belt not only produced numerous Hungaria asteroids . . . but it also created a long-lived tail to the LHB . . . In the LHB phase, we assumed the Nice model occurred . . . If we are correct, the LHB lasted 400 Ma, with its end set by Orientale (3.72–3.75 Ga), [and] this puts the start of the LHB at 4.12–4.15 Ga."

By 2012, Bottke et al. [140] extended the duration of the LHB even further, arguing that Archaean and early Proterozoic spherule beds on Earth mark additional, post-Orientale multi-ring-basin-scale impacts caused by E-belt objects, scattered by planet migration, thus producing an "LHB" during which basin-scale impacts occurred in the Earth–Moon system for at least 1600 Ma:

> "Here we report that the LHB lasted much longer than previously thought, with most late impactors coming from the E belt [producing not only] 10 lunar basins between 3.7 and 4.1 Gyr ago [but also] 15 terrestrial basins between 2.5 and 3.7 Gyr ago . . . (Bottke et al. (2012 [140] p. 78))."

The difference in age span between the lunar and terrestrial data on big impacts reflected (1) the preservation of older impact melt clasts on the Moon than on the Earth, and (2) Earth's greater area, by a factor ~16, such that Earth, for any given time interval and impactor flux in the Earth/Moon system, has a larger chance of experiencing an impact.

Regarding the Bottke et al. concept of the E-belt, we noted in Section 11 that modern spectroscopic observations of asteroids, taken together with Wetherill's (1977 [8,9]) models, suggest that the innermost edge of the asteroid belt may have been the only surviving repository for enstatitic asteroids scattered from the 1 AU region, with Earth/Moon-like isotopic chemistry. It is therefore important to note that the Bottke et al. (2010 [13]) "E-belt" definition referred not to "E-nstatitic" asteroids in the inner belt, but rather the possible "E-xtended" configuration of the inner edge of belt. (Anecdotal comment: During the drafting of this article, the author thought he recalled that around the 1980s, the group of asteroids on the innermost edge of the asteroid belt were occasionally referred to as the "E belt," because of the concentration E-class asteroids, with possible spectroscopic connection with enstatite-like materials. However, Clark R. Chapman, Stuart Weidenschilling, and Richard Binzel, who were much closer to asteroid spectroscopic classification and dynamics at that time, informed the author in independent private communications (2018) that they have no recollection of a term, "E-belt," being used to refer to a spectroscopically defined E-class asteroid concentration. The faulty memory may have been a construct only of the author's imagination after reading Wetherill's work.)

Though not discussed in the Bottke et al. papers, their sequence of LHB models, with no sharp spike at 3.9 Ga ago and an increasingly gradual decline in impact rate, eventually reaching 2.5 Ga ago, began to look more and more like the measured lunar cratering data and hypotheses from the 1960s, 70s, and 80s, and also resembled the Morbidelli et al. (2001 [106]) model: impact rates declining from $>10^2$ times the present rate at 3.8 Ga ago to lower levels at ~3.2 to ~2.9 Ga, before "leveling out" to impact rates no more than a few times today's rate. While most recent empirical models of impact rate versus time suggest an essentially flat rate since 3 Ga ago, coordinated papers by Quantin et al. (2007 [40]) and Hartmann et al. (2007 [42]) suggested a continuing decline after ~3.0 Ga by a factor about 2 to 4 (see Sections 5 and 13)—which would leave the idea of an *end* or *cutoff* for "late bombardment" completely ill defined.

To sum up, dynamical modeling publications circa 2001–2012 maintained the terminology of a cataclysmic bombardment or LHB, but year after year, especially after ~2005, extended its length fisbackward to at least 4.1 Ga ago and forward to ~2.5 Ga ago, reducing the term "LHB" to a scenario with no resemblance to the original meanings of the terms "terminal cataclysm" or "LHB" that had persisted for three decades. While not criticizing the science of the models, we suggest that this semantic style (drastic re-definition of a phenomenon from a 150 Ma spike to a 1600 Ma gradual surge, but reporting all versions as the "LHB"), sowed seeds of confusion throughout various scientific communities. After all, if Imbrium, Serenitatis, Nectaris, Crisium, and other basins did not all form during a roughly 150 Ma impact spike in impacts ~3.9 Ga ago, following a ~400 Ma period of impact quiescence, then a global multi-basin-forming "terminal cataclysm" and a "*late heavy* bombardment," as portrayed by researchers such as Turner and colleagues [5,6] in the early 1970s, Ryder [11] in 1990, Levison et al. [12] in 2001, the Nice model authors [107–109] in 2005, and Bottke et al. [13] in 2010, and treated as an option by Tera, Papanastassiou and Wasserburg [3–5] , and by Wetherill in the 1970s [7–9] simply did not happen. Yet the now ill-defined terminology was maintained.

To return to the formation of the basins themselves, and the possible role of an E belt, Bottke et al. (2011 [153]) argued that the Hungaria population of E class asteroids is the modern remnant of a once more massive E belt, and they proposed that the original E-belt population can be estimated from this circumstance, leading to model-dependent calculations of specific numbers of lunar impactors of various sizes during their LHB:

" . . . we predict the E belt created 8 ± 3 LHB lunar basins, with 2 ± 1 being Imbrium or Orientale-sized. Considering that 3 ± 2 lunar basins should come from the classical main belt (this) limits the scope of the LHB and probably forces Pre-Nectarian basins to come from elsewhere (e.g., leftover planetesimals)."

In this vein, Bottke et al. (2012 [140]) make an interesting assertion that some sort of LHB had to happen, because if the decline in cratering rate begins as far back as planetary formation, their models cannot be right:

"Our model results also demonstrate the need for an LHB. [Since] the E belt model[s] have steep decay rates as the LHB begins . . . shifting the start of the LHB to early solar system times, say 4.5 Gyr ago, would . . . take away the E belt's ability to produce late lunar basins . . . "

Thus, Bottke et al. (2012 [140]) stated: "Accordingly, if the giant planets reach their orbital configurations much earlier than 4.2 Ga ago, the E belt and main asteroid belt cannot form basins like Imbrium and Orientale at 3.7–3.9 Ga." The idea was that the two large, perhaps largest basins, Imbrium and Orientale, were the last to form, and too close together in time to be statistically reasonable if basin formation had been spread out over a longer time period.

Such arguments, of course, involve statistics of small numbers, and also require assumptions about dates and the formation rates. We have seen, however, that the assigned dates of lunar basins other than Imbrium are highly problematic.

Let us investigate the issue of largest multi-ring basins more closely. First, to clarify basin sizes, Orientale appears to be slightly smaller than Imbrium: The three rings of Imbrium at 1340, 970, and 670 km in diameter; the outer three rings of Orientale are a marginally defined out ring at 1300 km, plus more prominent rings at 930, and 629 km in diameter. Orientale has two more inner rings, at 480 and 320 km. If we were to judge basin sizes by innermost *visible* rings, Orientale would be only half the size of Imbrium. However, Imbrium may have had such inner rings; the inner part of Imbrium today is buried in the Mare Imbrium lavas, making the size comparison difficult, and the innermost ring we have listed is a ring of isolated peaks protruding from the lavas. To add relevant comparisons, the somewhat vestigial outer ring of Crisium has a diameter of 1060 km, and the prominent, but partial Altai scarp ring of Nectaris corresponds to diameter 840 km. (These measurements are all from Hartmann and Kuiper (1962 [82])).

As for Imbrium and Orientale being largest basins, the putative South Pole–Aitken impact basin, on the far side, is commonly described as being the earliest and largest impact basin, at ~2500 km in diameter. Oceanus Procellarum has been occasionally discussed as an even earlier, primordial impact basin, possibly 2600 km or more in diameter, now impact-degraded and partly overlapped by much of Imbrium impact structure. The GRAIL team discovered a "ring" of gravitational anomalies around Oceanus Procellarum, with a diameter of roughly 3000 km, and concluded that they mark fractures with lava intrusions (Andrews-Hanna et al. (2014 [154])). They then argued that Oceanus Procellarum could not be an impact basin on the basis that these features comprise about five or six linear segments, whereas impact basins should have more perfectly circular rim structure. This last argument is inconclusive, however, because the outer two rings ("Cordillera" and "Rook") of Orientale have several remarkably linear segments, associated with the fault fractures that allow the inner parts of the basin to slump, forming the overall "ring" structures. In the case of a Procellarum-sized impact, the lunar sphere itself was disrupted, and the mechanics and geophysics of how a lunar-sized body would re-create its spherical shape after a Procellarum-sized impact-excavation is poorly understood. The mare lava strips along the bases of several of the Orientale ring scarps, incidentally, suggest igneous extrusions (hence, intrusions as well) can occur as part of the formation of some basins. This supports the idea that some non-impact-melt igneous rock materials, such as clustered around 4.33 and 4.21 Ga ago, may date from multi-ring basin-forming events.

If we are to base arguments on the narrow time interval between the relatively well-measured age of Imbrium and the uncertain age of Orientale, it is important to discuss Orientale's age. No missions have landed on its ejecta blanket to give us reliable dates, but Orientale must be *older than the oldest nearby lava plains*, since no Orientale-radial ejecta are apparent on their surfaces. The nearest lava plains are southwest Mare Oceanus Procellarum and Mare Humorum, within the range of Orientale ejecta. Hiesinger et al. (2000 [76]; 2003 [155]) have made careful crater counts on various lunar mare units, including these two regions. Their age vs. crater density relation is calibrated with the rock ages reported from other various Apollo landing sites. Of 17 crater-count-dated mare lava units in Mare Humorum, they find one outlier at 3.94 Ga, and the next-oldest are two units in the 3.75 to 3.70 Ga age range (Hiesinger et al. (2003 [155]), their Table 8).

If we treat the Hiesinger measurements more like noisy samples of a more homogeneous mare, we can note that the average of the eight oldest unit ages that they measured was 3.69 Ga. For comparison, in our Table 3, the four oldest 1σ likely ages for Imbrium range 3.91 to 3.79 Ga, and the four youngest 1σ likely ages for Imbrium range from 3.90 to 3.75 Ga. If we assume that the Hiesinger et al. oldest units in Mare Humorum are truly unscathed by Orientale ejecta, then it seems fair to say that the Orientale age could range from simultaneous with Imbrium to ~200 My after Imbrium. In Oceanus Procellarum, 60 units were dated by crater counts and the oldest four units were assigned dates of 3.59, 3.57, 3.53, and 3.48/3.74 Ga (Hiesinger et al. (2000 [76]), Table 3), allowing Orientale to be more like 150–200 Ma after Imbrium. If these results were for some reason more reliable than the Humorum results, they would weaken the constraint, allowing Orientale age to range from simultaneous to as much as ~300 Ma older than Imbrium.

From the present author's point of view, while the crater count *relative densities* may be good to around 10%, the calibration of lunar crater count *absolute ages* are probably not good to three significant figures, which, if valid, add perhaps 50–100 Ma uncertainties to the uncertainty of this discussion.

To take another approach to the Orientale age problem, the closest Apollo rock sampling to Orientale was the collection from Apollo 12 mare lavas, somewhat farther from Orientale than the above. The Lunar and Planetary Institute web site "Lunar Sample Overview" [156] for the Apollo 12 site states, "The basalts at the *Apollo 12* site formed 3.1 to 3.3 billion years ago . . . " If we assume that these dates should supersede the crater count dates, and that these units do not show signs of Orientale ejecta, then the Orientale impact is constrained only to range from simultaneous to Imbrium to be a comfortable ~300–500 Ma younger than Imbrium."

To sum up the Orientale age: An empirical conclusion is that the broad Orientale age estimates overlap with the Imbrium age estimates, and that Orientale is anywhere from 0 Ma younger than Imbrium (formed by an Imbrium impactor satellite?—see next paragraph) to as much 200–500 Ma younger than Imbrium. From our present state of knowledge, it seems not stochastically or observationally implausible that if the impact rate was gradually declining, as described above, Orientale could have formed within a few hundred Ma after Imbrium, and the same, declining flux produced a modest number of additional, basin-scale impacts on Earth, with its 16 times greater area (and still greater gravitational cross-section) during the next 1200 Ma.

If one still wants to argue that Imbrium and Orientale may have improbably similar estimated ages, there is one more possibility that needs to be considered: As mentioned above and in the previous Section 14, the Orientale impactor might not have been independent of the Imbrium impactor, but rather a satellite of it (Hartmann (2012 [148])). If such a pair of bodies (~100–150 km diameter range) were orbiting within perhaps ten diameters of each other during approach to the Moon, a double impact would have been quite plausible on the 3475-km-wide Moon. The Orientale impact could thus have been within hours of the Imbrium impact. Hartmann also suggested that if the satellite body missed the Moon during the Imbrium impact, but stayed on an Earth-crossing heliocentric orbit, it might have impacted the Moon within 10 Ma or 100 Ma after the Imbrium impact, based on cosmic ray exposure ages of meteorites blown off the Moon, as well as interplanetary meteorites (Gladman et al. 1995 [66]; Eugster 2003 [67]). Bottke (private communication, 2019) pointed out, however, that during a return approach to the Earth–Moon system, it would have been ~20 times more likely to hit Earth. If the two basins did form within 100–200 Ma of each other, perhaps independently, Orientale ejecta 100 Ma later than Imbrium may be mixed (laterally or vertically) with Imbrium ejecta in many regions. This may help account for the spread in petrology types and reported ages of the ~3.9 Ga materials, as well as the range of ages attributed to Imbrium.

A third note on the relation of Orientale and Imbrium: Wetherill (1975 [7]) made an early examination of time interval between Imbrium and Orientale, already being alleged to be suspiciously coincidental in time. Utilizing the size frequency distribution of craters and impactors, he concluded that:

*"Using the mass-distribution law ... about 3 × 10²⁰ g of smaller bodies would strike the moon in the following 200 m.y. [after Imbrium], in agreement with the observed Orientale impact and the crater densities of the highland plains units."*

To return to the general thrust of the Bottke et al. (2012 [140]) paper, they stated that:

*"The known ancient [terrestrial impact spherule] beds argue for an intense, protracted phase of late terrestrial bombardment. Curiously, these enormous blasts have no obvious source, even though many occurred relatively soon after the formation of the 930-km-diameter lunar basin Orientale ... "*

Contrary to this statement, the time intervals among large impacts on Earth and Moon do not seem compellingly inconsistent. Stuart-Alexander and Howard (1970 [90]) listed 27 basin-scale impacts, and Hartmann and Wood (1971 [20]) listed 31 multi-ring basins. To these we can add the South Pole–Aitken far-side basin, widely considered to be the oldest and largest basin, at a somewhat poorly defined diameter about 2500 km. And we can recall that Oceanus Procellarum possibly marks a ~3000 km-scale primordial impact feature. If we accept ~30 multi-ring basins formed in the first 600 Ma (about 5 such lunar basins per 100 Ma), the *average* time interval between impacts of this scale would be ~20 Ma. With declining impactor flux, the interval may be considerably longer by 3.9 Ga ago, with statistical variations. Now let us add the observation by Bottke et al. (2012 [140]) of 15 terrestrial basin-scale impacts between 2.5 and 3.7 Ga ago (with possibly more unobserved, due to oceans and plate tectonics?) on the 16-times-larger surface of Earth in the next 1200 million years after Orientale. In this situation, is it implausible that one or two large basin-scale impacts could happen on the Moon around 3.9–3.8 Ga ago?

To investigate this with our crude statistics of two big late lunar basins, let us try out an assumption of 1 Imbrium/Orientale-size basin in each 200 Ma around 3.8 Ga ago on the Moon, a rate of ~0.5 such basins/100 Ma. To engage the Bottke et al. (2012 [140]) small-statistics assertions, we should then have 16 times higher impact numbers on Earth, hence an order-of-magnitude formation rate of ~8 Imbrium/Orientale scale basins per 100 Ma on Earth. Bottke et al. report 15 spherule-producing terrestrial basin impacts in 1200 Ma, or only ~1.2 per 100 Ma on Earth. Our crude discussion assumes that the impacts producing the Archaean spherule fields being found on Earth were of giant Imbrium/Orientale scale. However, if the terrestrial spherules come in part from smaller basins, then because of the size distribution of basins and craters, the terrestrial impact rate for that size basin would be higher than our estimate. Given the small numbers of giant basins on planetary bodies, their size distribution is uncertain, but we can apply a conservative power law that applies among the largest craters (number going as diameter to exponent −2). Thus if we assume that terrestrial impact basins of 1/2 or 1/3 of Imbrium/Oriental sizes (as well as the "Imbriums" and "Orientales" themselves) contribute to the Archaean terrestrial spherule populations, we would have a formation rate for such basins of ~5 to 11 per 100 My, of just the right order of magnitude to resolve the issue raised by Bottke et al. (2012 [140]).

Another factor in the gross uncertainty of the above discussion is the Bottke et al. [141] assumption that the effective sampling area on Earth is 16 times that of the Moon may be too high, since plate tectonic subduction may have removed some evidence of Earth's earliest basins and their strewn fields of spherules. Thus, as is often the case with numerical models, our "model" can be tweaked to allow various desired results—but it seems premature to dismiss the possibility of two large lunar basins in the time interval of ~3.9 to 3.7 Ga ago. To resolve the issue, we need to pursue empirical dating, i.e., clear sampling of the Orientale rim structure, ejecta blanket, and impact melts.

Another step in the evolution of the dynamical modelling discussions came when Morbidelli, Marchi, Bottke, and Kring (2012 [149]) presented an important re-examination of the models of evolution of lunar cratering rates, comparing the dynamical models with empirical evidence from lunar crater densities on radiometrically dated surfaces. As described in Section 5, data of that sort give a curve for accumulated cratering density vs. age after ~3.8 or 3.9 Ga ago. Morbidelli et al. used the empirical curve from a review of the lunar data by Neukum and Ivanov (1995 [37]), and then compared that curve with a calculated crater accumulation model based on scattering from the putative E belt (Bottke et al. 2012 [140]). They found that if they extrapolated the Neukum-Ivanov curve back from 3.9 to 4.5 Ga, and then assumed that the E belt was destabilized at 4.1 Ga ago, the two curves fit each other very well. As noted in our Section 14, their paper concluded that "there was no prominent and narrow spike ~3.9 Gy ago," but they did allow for an "uptick ... near ~4.1 Gy ago ... "

A certain irony pervaded the period of 2010–2012, as might be illustrated by another personal anecdote. Around the early 2000s, when I questioned the Ryder gap and re-suggested the possibility of an intense early declining bombardment between ~4.4 and ~4.0 Ga, I was assured by many colleagues (both in audience comments after my talks and in private discussions) that the dynamical models had proven that all the early planetesimals were swept up in the first 100 Ma. A problem with this frequent comment was semantic: The mass accumulation rate during planet accretion was so high that the impact rate could decline by a factor of $10^3$ in the first 100 Ma—thus satisfying the language that early "all planetesimals had been swept up"—but still leave an impact rate orders of magnitude higher than today, in the period ~4.4 to 4.1 Ga ago. The irony was that, ca. 2011, the same colleagues were scrambling to fill in the once sacrosanct Ryder gap with sources of intense cratering.

Morbidelli et al. (2012 [149]), after their useful summary of the evidence around 2012, moved on to a new, different model, on the grounds that (1) " ... no source on inner solar system projectiles has yet been found that decays over 1 Gy ... with the rate implied by Neukum and Ivanov [1995]," and (2) lunar geochemical data constrain the "total mass of projectiles that should have hit the Moon in the bombardment history of Neukum and Ivanov [1995]." The argument was still highly model dependent, with parameters such as the date of destabilization of the E belt by giant planet migration,



the number of objects in the original E belt measured relative to the number assumed by Bottke et al. (2012, [140]), and the total mass and sources of interplanetary material accreted by the Moon after its formation, which depends in turn on model calculations dependent on impact velocities. In their conclusion, Morbidelli et al. (2012 [149] p. 149) suggested:

> " ... *the need for a break, or inflection point, in the bombardment curve [versus time] at sometime in the 4.1–4.2 Gy interval* ... *This discontinuity defines the signature of a lunar cataclysm."*

Here again we see retention of a (watered-down) term "cataclysm," along with movement away from the hypothetical "Ryder gap" in impacts from ~4.4 to ~4.0 Ga ago (Ryder 1990 [11]), which had been invoked by earlier geochemists and dynamicists as a firm constraint in early dynamical models (Cohen et al. 2000 [69]; 2005 [71]; Kring and Cohen 2002 [72]; Levison et al. 2001, [12]; Gomes et al. 2005 [109]).

Fritz et al. (2014 [157]) expanded the study of early bombardment history to extrasolar planetary systems. They described how the various cratering and dynamical histories could constrain habitability in other planetary systems. Their conclusions adopted further movement away from the "terminal cataclysm spike" to a more spread-out cratering history, referred to by these authors with a new term, "the Heavy Bombardment Era (HBE)."

As with Morbidelli et al. (2012 [149]), Fritz et al. (2014 [157]) thus seemed to move back toward the pre-Apollo "Early Intense Bombardment" concept, but with a declining flux and a more complex history, while preserving aspects of the "Grand Tack" model (migration of giant planets into the inner Solar System), and the Nice model. Fritz et al. assumed what they referred to as a "single spiked saw-tooth like bombardment timeline". They proposed that such a timeline was:

> " ... *characterized by two episodes: an early one, due to the decay of the planetesimal population left over from the accretion of terrestrial planets and a late one, dominated by the E-belt projectiles. The early (first ~300 Myr) bombardment could be a generic feature of terrestrial planets* ... *; in contrast the late (after the initial ~300 Myr) bombardment is specific to the evolution of the giant planet [sic] in the Solar System* ... *".*

Marchi et al. (2014 [158]) presented new calculations of impact rates from scattered asteroids. They discussed results from 5000 different runs of their numerical model, helping to clarify statistical aspects of the scattering process. Their results showed a decline in cratering from 4.4 to 4.0 Ga (and afterwards) with no cataclysmic spike and no "Ryder gap" in impact rate before 4.0 Ga.

The Marchi et al. (2014 [158]) curve, if seen as a dominant part of the story, appeared once more to move the discussion back toward a declining cratering rate before 4.0 Ga ago, such as the statement by Tera et al. (1973 [2]) that "basin forming event(s) must have been even more prevalent prior to 4.0 AE." Indeed, as shown in Figure 13, their curve for asteroid impact flux vs. time overlapped cruder, pre-Nice model proposals of impact flux versus time that had been suggested in the 1970s, based on then-current lunar cratering data and first-order theoretical ideas of planetesimal scattering with gradually increasing half-lives associated with increasing planetesimal inclinations and eccentricities (Hartmann 1970 [31,32]; Safronov 1972 [50]; Wetherill 1977 [8,9]; Neukum 1983 [36]; Hartmann et al. 2000 [35]; Hartmann 2015 [159]). Marchi et al. allowed that a gradual wave of late bombardment could be inserted into their curve, if needed:

> " ... *given that large impactors could have struck at relatively late times (for example the expected surge of projectiles at 4.15 Gyr ago via the LHB), zircon production* ... *could have occurred for many hundreds of millions of years, as observed* ... *We argue that the peak of Hadean zircon ages at 4.1–4.2 Gyr reflects the onset of the LHB* ... *Indeed, we find that a scenario with an LHB spike at significant [sic] younger ages, say 3.9 Gyr, is inconsistent with Hadean zircon age distributions."*

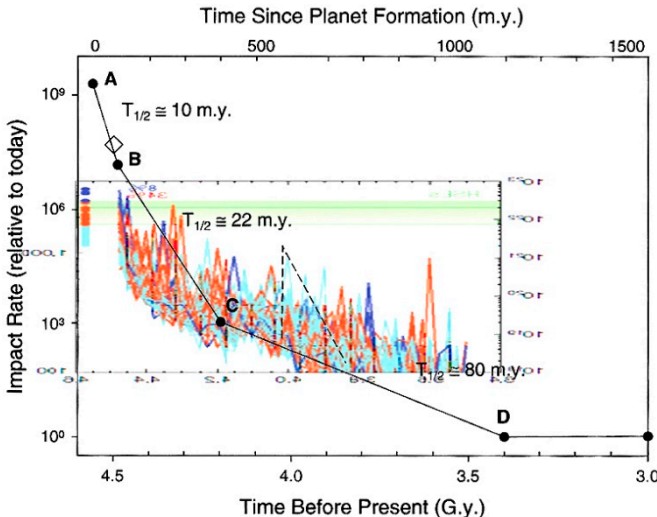

**Figure 13.** A numerical model from Marchi et el. (2014 [158], colored data) showing computed decay curves of impact rate from main belt asteroids, is here superimposed on a graph by Hartmann et al. (2000 [35]) showing early-proposed lunar flux curve (ABCD) based on crater counts after 3.9 Ga ago, and extrapolated back before 4.0 Ga ago with theoretically estimated half-lives (background larger diagram). Marchi graph (small diagram) was reversed left to right so that time flows from left to right as in larger diagram. Scales of the time axis and the impact rate were adjusted in Photoshop to be equal. (Diagram by W.K.H., presented at 2014 history session at DPS [Division of Planetary Sciences, American Astronomical Society], but not published).

The wordings in the papers of 2010–2014 illustrate the manifest desire at that time to explain some concept consistent with the terminal cataclysm/LHB paradigm, in spite of the admitted inability to find consistent evidence on when the phenomenon occurred. It is hard to deny this ongoing tendency, in which many modelers assumed a priori the reality of some sort of phenomenon associated with the term "cataclysm" or the acronym LHB, and then continued to use the original term for the (varying) result of their current model.

Even more recent papers, especially in the lunar and terrestrial geology literature, continue to adopt the "LHB" concept as a constraint. Many researchers outside the dynamical community thus assume, reasonably, that the term still refers to the original definition. For example, lunar geologists Martellato et al. (2017 [98]) stated that "Lunar samples constrained the age of the Serenitatis basin to 3.89 Ga, within the Nectarian period, when the Moon was undergoing a heavy bombardment by 100 km size asteroids."

Anecdote: In searching for a way to convey my discomfort about these issues, I was struck by the subtle, semantic relevance of an old joke, related by my father. It illustrates succinctly what is wrong with the current "LHB" usage. The joke involves a legendary American story about boyhood George Washington (praised in myth as never telling a lie): He admits to his father that he had cut down the family cherry tree with his new ax. According to the joke, a modern museum proudly displays the "original ax" that young Washington had used . . . while admitting that the handle had been replaced four times, and the blade replaced twice.

Morbidelli and seven coauthors (2018 [160]) supported somewhat stepwise monotonic decline in impact flux, and gave the previously amorphous ideas a name (perhaps a useful step in having them be taken seriously): "the accretion tail scenario." In their abstract they wrote:

*"There is an intense debate between two possible interpretations of the data: in the cataclysm scenario there was a surge in the impact rate approximately at the time of Imbrium formation, while in the accretion tail scenario the lunar bombardment declined since the era of planet formation and the latest basins formed in its tail-end . . . We use updated numerical simulations of the fluxes of asteroids, comets*

*and planetesimals leftover from the planet-formation process ... In summary, given the currently available data, models and knowledge, our preference goes to the accretion tail scenario."*

## 16. Connections to the Origin of the Moon

Another emerging constraint on the new dynamical models involves isotopic chemistry. In a complex way, it relates not just to the putative cataclysm/LHB period, but to very early events, such as the origin of the Moon. The initial modern giant impact hypothesis of lunar origin proposed an impactor that was the 2nd largest (or 3rd, 4th ... ) local planetesimal around 1 AU (Hartmann and Davis (1974 [161]), (1975 [162]); Cameron and Ward (1976 [163])). However, after the recognition of likely scattering of outer Solar System objects by resonant effects (e.g., Levison et al. (2001 [12]); Morbidelli et al. (2005 [107]), Gomes et al. (2005 [109])), there was a tendency to assume—either explicitly or implicitly—that the potential Moon-forming impactors came from the outer Solar System, or at least from well beyond 1 AU. According to meteorite data, this meant that they would have very different isotope chemistries than Earth and Moon. According to numerical models of a moon-forming impact, the Moon should end up with a measurably higher percentage of the impactor than the post-impact Earth. Therefore, if the impactor had *substantially* different isotope chemistry than Earth, the Moon should have measurably different isotope ratios than Earth—which is not observed.

This created a new conundrum for the dynamical models. On the one hand, if the Jupiter migration and scattering was chosen to happen at 3.9 Ga ago, so as to explain the LHB, a dramatic spike in impacts on the Moon and asteroids should have occurred at that time—which is not observed in lunar meteorite and asteroidal meteorite data. Worse yet, Fassett and Minton (2013, [164]) argued that the planet migration in the Nice-like scenarios needed to be short lived, or it would have pumped up terrestrial planet eccentricities enough to cause collisions among them. But if the Jupiter migration and scattering of primordial bodies was proposed to occur at ~4.5 Ga ago (perhaps facilitating a giant-impact lunar origin), then the likely Earth-impactor would have formed far from Earth, producing a moon with substantially different isotope chemistry from Earth—which is also not observed. So strong was the belief that dynamical models required non-local impactors that an "isotope crisis" was declared regarding the lunar origin impact hypothesis (Melosh 2009 [165]), rather than questioning the assumptions of dynamical models themselves. (See Philosophical Transactions of the Royal Society (A), 372, 20132049 for papers from the "isotope crisis" conference held in England in 2013.)

In answer to such problems of lunar origin, Hartmann (1986 [166]; 2014 [167]) emphasized that we have direct empirical evidence of the existence of parent bodies (large planetesimals) with Earth-like isotopic chemistry. Namely, as already discussed in Sections 11 and 15, enstatite meteorites have nearly the same isotope chemistry as Earth and Moon (much closer than any other known meteorites types, including martian meteorites). Herwartz et al. (2014 [168]) presented new, higher precision isotope measurements of lunar rocks, finding a slight offset in lunar O isotopes from Earth isotopes in the direction of enstatite meteorites. In short, a moon-forming impact from a large, enstatite-like parent body would resolve most (but not all) of the proposed isotope crisis.

Thus, the history of enstatitic parent bodies plays into the history of Earth-Moon bombardment. As described in Section 11, there have been suggestions of enstatitic impactors forming the large lunar multi-ring impact basins around or before 3.9 Ga ago, and Wetherill (1977 [8,9]) had already concluded that near-1 AU planetesimals (with Earth-like enstatitic isotope ratios) could have been stored in the innermost asteroid belt. Then, as described in Section 15, dynamical models by Bottke et al. (2010 [13]; 2012 [140]) suggested that early "extended" inner belt objects could have been disturbed and scattered by the giant planet migrations. Many of the scattered objects from that region could thus have been enstatite-like objects, originating near 1 AU. Hence, if the Grand Tack migration of giant planets occurred around 4.5 Ga instead of 3.9 Ga ago, it could plausibly have perturbed enstatite-like E-belt bodies (originally from 1 AU) back into the inner Solar System. As noted in Section 11, Norman et al. (2002 [124]) and Becker et al. (2006 [125]) suggested from lunar sample analysis that certain basin-forming impactors, well after lunar formation, involved enstatite chondrite-like projectiles.

Such results suggest that large planetesimals with Earth-like isotopic chemistry were common in the inner Solar System, allowing a resolution of the "isotope crisis," while at the same time allowing (and perhaps constraining) the fundamental feature of the current dynamical models—an early migration of Jupiter and other giant planets.

One more aspect of primordial intense cratering deserves comment. Introductory geology textbooks are fond of showing the evolution of complex terrestrial areas, like the Alps, with a series of diagrams starting with flat-lying strata, which then are crumpled into the observed complexity. At some conferences on the Moon's crustal evolution, even as recent as the Second Conference on the Lunar Highlands Crust (2012, Bozeman, Montana), the present writer, reporting as a journalist, sensed a tendency to start with similar, layered primordial stratigraphy. This assumed a settling-out of the dense minerals of the magma ocean and floating of a low-density anorthositic surface layer in the upper kilometers. This subliminally assumed a "Ryder's rule," where the cratering rate was near zero by the time magma ocean solidified—so that deep-seated lowest layer geological structure should remained buried. Such an idealized lunar stratigraphy, however, may never have existed if an early intense cratering rate has declined since the beginning, with early creation of a kilometers-deep megaregolith. The anorthositic surface layer suggests some layering survived, but those layers are highly brecciated. The existing crustal structure, megaregolith layering, and degree of mixing of plutonic rocks such as norites into the upper regolith may thus offer some observational constraints on the balance between chaotic impact-mixing and the tendency toward layering at the time of magma ocean solidification and formation of the lunar crust.

## 17. Dismantling of the Terminal Cataclysm/LHB Paradigm, 2015–2018

In 2015, two conferences were held that indicated a dismantling of the "classic" Terminal Cataclysm/LHB paradigm, which had been defined by Levison et al. (2001 [12]), as cited more briefly in several earlier Sections:

> "The "Late Heavy Bombardment . . . was a phase in the impact history of the Moon that occurred roughly 4.0 to 3.8 Gyr ago. It was during the LHB that the lunar basins with known dates were formed. The LHB was either the tail end of accretion or it may have been a spike in the impact rate ("terminal cataclysm," Tera et al. 1974) at that time. In either case, it marks the final epoch when the dominant surface geology of the Moon was created by large impacts . . . "

Again, (as cited earlier), defined with admirable conciseness by Bottke et al. (2010 [13]):

> " . . . as a period 3.96–3.75 Ga when many lunar basins (e.g., Serenitatis, Imbrium) and impact melts were produced."

The two 2015 "revisionist" conferences were where, in the writer's perception, the *dominant* discussion first turned against the paradigm of 3.9 Ga spike in basin formation. First was the "Workshop on Early Solar System Impact Bombardment III," 4–5 February 2015, in Houston. This saw many (inconclusive) presentations about attempts to reconcile the dynamical models with the radiometric data. The emphasis du jour was on a gradual swell in distributions of impact indicators among asteroidal meteorites, around 4.1 to 3.7 Ga ago, which might be reconciled with some sort of extended, late surge in planetesimal bombardment, which modelers had been suggesting (see Sections 15 and 16). Representative of the mood, Swindle and Kring (2015 [169]), in their abstract, stated that "...the extreme version of a lunar cataclysm envisioned by G. Ryder is clearly not accurate."

The second 2015 "revisionist" conference of interest, on 25–26 July at the Meteoritical Society meeting, was a "Workshop on the First 1 Ga of Impact Records: Evidence from Lunar Samples and Meteorites." The present author was a co-organizer and attended. Many of the same attendees from the previous conference were present, and we debated issues such as uncertainties in current radiometric dating of lunar samples, as well as meaning of lunar and meteorite radiometric asteroidal ages and again, there was again a *dominant* sense in the room that basin dating needed re-examination and that

the classic terminal cataclysm paradigm, with most major basins forming in a spike at 3.9 Ga ago after an impact lull. An amusing, but illuminating, good-natured bon mot from Eric Asphaug referred to the recent spate of constantly mutating dynamical models as "models masquerading as scientific progress." At both "revisionist conferences" the present writer raised the question of whether the spike at 3.9 Ga ago and the preceding impact lull could now be considered to be "off the table," and there was no significant dissent from the attendees.

Subsequent to those meetings, critiques of the terminal cataclysm/LHB paradigm accumulated with increasing frequency and acceptance. Examples are:

- Boehnke and Harrison (2016 [53], cited briefly in Sections 5 and 7) noted that much of the evidence recently marshalled in favor a terminal cataclysm comes from Ar dating. They argued that "diffusive loss of $^{40}$Ar from a monotonically declining impactor flux coupled with the early and episodic nature of lunar crust formation" produces spikes in $^{40}$Ar/$^{39}$Ar ages at ~3.9 Ga, producing an appearance of "Illusory Late Heavy Bombardments" (their title). This paper thus had some ancestry in Hartung's (1974, [46]) mechanism producing an illusory peak in rock ages, although Hartung was not referenced.

- Denevi (2017 [170]) remarked in a review article that "New work has shown that Imbrium ejecta likely litter the Apollo 17 landing site, so the samples thought to date Serenitatis impact may instead date the Imbrium basin. If so, we're left with diminished evidence that the late heavy bombardment occurred at all—and increased evidence of the monumental effect that the Imbrium impact had on the nearside lunar landscape." Denevi's article is titled "The New Moon"; it is interesting to note that while some of the "newness" comes from new data, much of the "newness" refers to the reinterpretation of early data.

- A review article by Zellner (2017 [171]) emphasized that decline of the cataclysm paradigm affects ideas about the origin of terrestrial life. The abstract states " . . . most evidence supports a prolonged lunar (and thus, terrestrial) bombardment from ~4.2 to 3.4 Ga and not a cataclysmic spike at ~3.9 Ga." The article noted the ongoing revisions of the Nice model, and her Section titled "A 'Cataclysm' No More," states that:

When taken together, lunar orbital data, terrestrial, lunar, and asteroid sample data, and dynamical modelling of solar system evolution, suggest an extended lunar bombardment from ~4.3 to ~3.5 Ga with evidence for impacts older than ~3.85–3.9 . . . in contrast to previous reports . . . Additionally, these sample ages provide evidence for a series of impacts lasting hundreds of millions of years, and not a single "cataclysm" that created all of the large basins on the nearside of the Moon in a short period of time.

- Bottke and Norman (2017 [141]) published a major overview of "Late Heavy Bombardment" (their title), in *Annual Reviews of Earth and Planetary Sciences*, taking a cautious view of the situation. Altering the definition of LHB once again, they stated that "Late Heavy Bombardment refers to impact events that occurred after stabilization of the planetary lithospheres such that they could be preserved as craters and basins." Taken literally, this could mean that LHB covers the entire history of the Moon after about 4.4 Ga ago. However, adopting Ryder's rule that lack of detection of impact melts means lack of impacts, they supported a period of "relative quiescence" in impacts from ~4.4 Ga until ~4.2 to ~4.0 Ga, and a "discrete episode of elevated impact flux" from ~4.2–~4.0 Ga to ~3.5 Ga, i.e., an impact surge or spike lasting at least 500 to 700 Ma. Adding "evidence from Precambrian impact spherule layers" (Bottke et al. 2012 [140]), they suggested a long-lived tail of terrestrial impactors lasted to ~2.0–2.5 Ga, so that the LHB duration would be considered to be 1500 to 2200 Ma. In their conclusions, Bottke and Norman refer to increasingly commonly heard statements that "'I do not believe in the Late Heavy Bombardment,'" and then they comment:

*"It is not clear precisely what this means, but it likely refers to doubts that the Moon and other worlds were hit by a spike of large impact events between ~3.7 and ~3.9 Ga. Given the evidence provided here,*

*we agree that the original basis for a strong version of the Terminal Cataclysm hypothesis has been substantially weakened. With this said, however, it is worth considering that two nearly 1000 km lunar basins, Imbrium and Orientale, formed on the Moon during this short interval. . . . it is unavoidable that at least some Archean-era impacts on Earth may have been comparable to Orientale-formation events on the Moon. Now that is a late heavy bombardment!"*

Michael, Basilevsky, and Neukum (2018 [30]) reviewed the data on the early bombardment of the Moon, stating:

*"We conclude that the statistics of sample ages contradict the terminal cataclysm scenario in the bombardment of the moon . . . Thus, our general conclusion is that the terminal cataclysm proposed by Tera et al. (1973, 1974) . . . did not occur."*

The proposed concept of the term "LHB" had now become sufficiently altered, compared to the original concept, that it could be invoked to fit a wide variety of new concepts. Regarding the state of the dynamical models, Bottke and Norman (2017 [141]) concluded:

*"Dynamical models that include populations residual from primary accretion and destabilized by giant planet migration can potentially account for the available observations, although all have pros and cons. The most parsimonious solution to match constraints is a hybrid model with discrete early, post-accretion [impactors] and later, planetary instability-driven populations of impactors."*

It could be argued, however, that this is not as parsimonious as a simple decline in the primordial impact rate in the Solar System, reaching a point where impact melt samples begin to survive until the present.

By 2018, the apparent collapse of the terminal cataclysm/LHB paradigm, as we have defined it here, was being openly reported by science journalists. For example, the journal *Nature* carried an article by journalist Adam Mann (2018 [172]), under the title "Cataclysm's End," stating that:

*" . . . the once-popular theory has come under attack, and mounting evidence is causing many researchers to abandon it . . . the community is grappling with the fact that a key chapter of solar system history might be vanishing before their eyes."*

Some of these disputations about LHB cite then-recent detections of evidence for pre-4.0 Ga impacts at 4.2 or 4.3 Ga and interpret them to imply that the LHB *began* at the oldest cited date. Here we observe, however, that in archaeology, virtually every reference to "the first use of stone tools" or "the first use of iron," etc., is replaced some years later by an earlier example. In science, we never have a guarantee that an earliest detection is the first occurrence. Meanwhile, the now-ambiguous "LHB" persists semantically as long as authors are allowed to change the definition to fit current models.

## 18. A Possible New "Megaregolith Evolution" Model for Explaining Lunar Impact Melt Data

If we propose intense cratering and multi-ring basin formation in the entire interval 4.5 to 4.0 Ga ago, then how can we explain the paucity of pre-4.0 Ga impact melts compared to numbers of fragments of primordial igneous crustal rocks that survive from the 4.4–4.1 Ga era? This issue was raised against the declining flux "accretion tail scenario" and the question has been discussed without fruitful resolution during numerous early conferences. Here, it is suggested that the key process is megaregolith evolution, and that it has several under-appreciated effects that constrain the kinds of rocks collected from the lunar surface, explaining the difference in survival of primordial igneous rocks and impact melts.

As mentioned in Sections 1–3, significant aspects of Apollo mission planning—and hence sample interpretation—were based on an underlying assumption that the Moon lacked erosive effects, so that surface samples would produce "Genesis rocks" revealing the entire history of the Moon. Thus, in spite

of the intense attention to lunar samples and to dynamical models of impact flux vs. time, surprisingly little attention has been paid to the effects of impact history on megaregolith evolution and, in turn, on the effects of regolith evolution on *filtering* the samples that can be found lying on the lunar surface today. Liu et al. (2019 [89]), in their analysis of Imbrium, Crisium, and Serenitatis ejecta and dates, exemplify the new attention to this process.

As mentioned in Section 4, the initial introduction of the "megaregolith" term (Nash et al. 1971 [44]) pictured fractured material "perhaps kilometers in thickness" and Hartmann (1973 [45]) pictured megaregolith fragmentation extending to a depth on the order of ~2 km, allowing deep sequestration of pulverized impact melt fragments. As we go down into the megaregolith, the loose material gives way to more cementation into coherent breccias by various processes (Wieczorek et al. 2013 [84], Spray 2016 [57]), leading to lower porosity, and eventually the semi-coherent primordial crust, laced with fractures. As a result of the size distribution of interplanetary impactors, the greater the depth, the fewer the impactors that were big enough to disturb the material at that depth. While the surface experiences a "continuum" of sandblasting by small impacts, the deep crust undergoes the stochastic effects of the few largest impactors at specific sites. By definition, the (irregular) "base" of the megaregolith is at the range of effective depths of penetration and excavation of the larger craters, where a transition between fractured and coherent or crustal rock occurs. The transition is also probably marked not only by original solidified crust but also by preserved intrusive dikes and sills of solidified magma from deeper levels. That transition occurs at different depths, depending on the statistical vagaries of the size and dates of the biggest one or two impactors that may have hit at any given spot since the solidification of the crust at that spot. These effects explain the petrologic diversity of ejected rocks from a large basin in any given spot. For example, the region impacted by Imbrium was probably already heterogeneous with a non-homogeneous pattern of previous impacts, impact melts, and plutonic intrusions. (For further discussion and a model of megaregolith evolution, see Hartmann (1973 [45]; 1980 [34])) By 1991, megaregolith models of Warren et al. (1991 [173]) considered megaregolith to have a depth as much as 10 km.

In support of those theoretical ideas, lunar seismic data indicated a multi-kilometer deep surface layer of low density, porous material. It was initially cited as about 1–2 km deep, but corrected by Nakamura (2011 [174], paragraph 9) to at least 3.3 km. In general, wave velocity increases downward, and according to Kahn et al. (2013 [175], p. 338):

> "It is believed that the continuous increase in compressional velocity between the upper first km and 20 km depth is primarily ascribed to the effect of crack closure with increasing pressure . . . Thus, the continuous increase in velocity from the upper to mid-crust is because of the transition from regolith . . . to competent crustal materials as pressure increases."

Gravity data from the GRAIL mission gave similar results (Zuber et al. (2013 [83]), Wieczorek et al. (2013 [84])). They measured the bulk density of upper layers in the heavily cratered highland crust as much less than the density of solid rock. The Wieczorek et al. (2013 [84] p. 671) density estimate for the upper highland layers was:

> "2550 kg/m$^3$, substantially lower than generally assumed . . . with an average crustal porosity of 12% to depths of at least a few kilometers."

The seismic and gravity literature typically discusses the changes in density, porosity, and seismic velocity in terms of observed compression and change in rock composition, but without commenting on broader aspects of regolith evolution's effects on surface samples. Regolith evolution models, based on meteoroid size distributions, suggest that pulverization by impactors, from the near-continuous rain of micrometeorites to the rare basin-forming objects, eats its way downward as the impacts accumulate toward crater saturation, first at small scale and gradually at larger scale.

Crude thought experiments may help explain some of the effects of megaregolith evolution. Suppose (as thought experiment 1) you live in a 100 m$^2$ square room with a firm adobe floor, but each

day you dig a new 1 m$^3$ hole, $1 \times 1 \times 1$ m on a side, and then carefully put the soil back into the hole. After a hundred days or so, your floor would be converted into a uniform "regolith" of 1 m depth. In this thought experiment the room represents the finite lunar surface model and the holes represent impact craters in an overly simplified model where all impactors have the same size. No matter how long we run the experiment, the "regolith" would rarely grow to a depth of >1 m, because new impacts would rework, and ever-more finely pulverize the 1 m of "regolith." As thought experiment 1a, suppose with each hole, you put no soil back into the hole, but rather distribute it around the room. This means that the "regolith" would now have near zero depth in the most recent holes places (fresh "crater" floors) but would be deeper than 1 m in other places. As thought experiment 1b, suppose you throw 10% of the soil from each hole out of the window (presenting material blown off the Moon). The result would be slightly slower regolith evolution and shallower average depth.

As a radically different thought experiment 2, suppose the holes in the floor were distributed in size according to a more realistic power law for lunar craters larger than 2 km. Namely, as the diameter of the hole increases by a factor 2, the number of holes decreases by a factor of 4. Let the power law be defined such that the smallest hole is centimeter scale, and the largest hole is the size of the room (resulting in destruction of the room; i.e., destruction of the Moon). Soil from a given hole is spread around the room. In this situation, you are not allowed to add only holes of one size (centimeter, meter, etc.), or focus on their effects of holes in a given size range. You must add holes according to the entire power law. As the days go by, most of the holes are small, but the longer the experiment runs (within the stochastic framework), the larger the holes that penetrate through the "regolith" into previously undisturbed adobe (or some underlying concrete foundation layer) below, and the deeper the "regolith" gets.

Similarly, in meteoritic/asteroidal reality, shown in Figure 14, as the crater density increases, the entire size–frequency distribution (SFD) rises at all diameters. Thus, given the nature of the SFD curve, a point comes when all sizes of craters of diameter D > 2 km approach saturation at about the same time (Hartmann 1980 [34]). As shown in Figure 14, this is because the slope of the SFD curve at D ≳ 2 km (but not at D < 1 km) is virtually parallel to the saturation curve. Thus, during an era when the early bombardment was intense enough, saturation happened on a freshly created surface within short intervals of tens or hundreds of Ma, and it happened simultaneously at all crater diameters ≳2 km, producing explosive growth of regolith. In Figure 14 this effect is shown with lunar isochron diagrams, which plot crater densities (SFDs) achieved on a newly formed surface after specified time intervals (100 Ma, 1 Ga, 10 Ga, etc.).

Figure 14a was plotted for surfaces observed today, showing that only surfaces older than ~4 Ga have reached saturation at diameters D ≳ 2 km. However, the measured cratering rate 3.8 Ga ago was ~150–200 times the present rate, which, as mentioned in Section 2, must also be the minimum *average* rate before 3.9. Figure 14b shows a conservatively measured estimate of 100 times higher-than-today cratering rates at 3.8 Ga ago. Here we see craters of few hundred meters diameter reaching saturation in only 10 Ma, suggesting tens of meters of regolith formed on that timescale. Figure 14c shows the dramatic result for a cratering rate 1000 times higher than today. This could have prevailed before 4.0 Ga ago, based on a simple extrapolation of the Apollo-measured cratering rate curve, from 3.8 Ga backwards to earlier time Ga, at a decay rate with half-life ~60 Ma, according to the "accretion tail" and "megaregolith evolution" hypotheses. For that cratering rate, saturation would occur at all multi-kilometer crater sizes, with explosive megaregolith creation, within about 100 Ma. Under these hypotheses, pulverization of materials to multi-kilometer megaregolith depths would occur within ~100 Ma intervals on surfaces created before ~4 Ga ago. These surfaces could include the floors and ejecta blankets of basins formed before 4.0 or 4.1 Ga ago, thus degrading the appearance of basins older than ~4 Ga.

Figure 15 illustrates a geometric method (Hartmann (1980 [34])) for estimating regolith depth as a function of surface age using the isochron diagram. Since the diagram is basically a histogram giving the crater density in each $\sqrt{2}$ diameter bin, it allows (as opposed to the popular cumulative

plot) evaluation of effects independently in each diameter bin. As explained in the caption, we add up the percentage of surface area covered by craters in each bin, starting at largest sizes. Thus, we can explore what happens at various crater sizes and depths as impact numbers take crater densities into the supersaturation regime. Comparison of the two halves of Figure 15 illustrates the dramatic increase in size of craters saturating the surface (hence regolith depth) as we go from a 4.0 Ga-old near-saturated surfaces back in time to supersaturated surfaces, with multi-kilometer craters covering 100% and even 200% of the surface area.

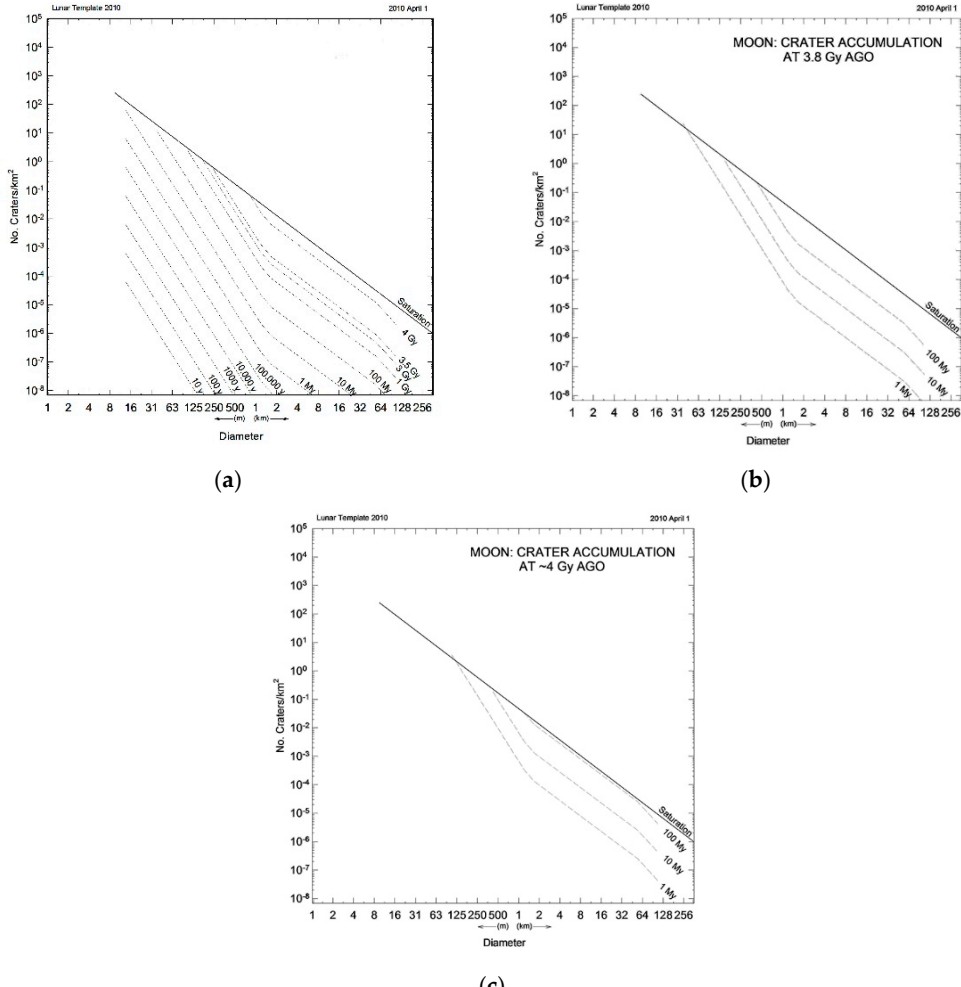

**Figure 14.** (**a**) A 2010 iteration of lunar isochron diagram, plotting crater density (craters/km$^2$) versus age of a given surface. Diagonal line at the top shows the empirically measured saturation equilibrium crater densities in the Solar System (based on Hartmann (1984 [19])); maximum crater densities oscillate by a factor ~2 around this line. (**b**) Isochron diagram as plotted for conditions ~3.8 Ga ago, using the cratering rate ~10$^2$ times the present rate. Such a cratering rate has been measured at that time from crater counts at dated Apollo landing sites (see Neukum et al. (2001, [38])). Cratering rates were declining half-life of roughly 10$^2$ years, so that the 100 Ma isochron has the most relevance, showing that craters of diameter ~500 m to ~1 km would reach saturation during a ~100 Ma interval at that time, producing a regolith many tens of meters deep. (**c**) Same as (**b**) but plotted for conditions ~4.1 Ga (or more?) ago, using cratering rate ~10$^3$ times the present rate, which may have existed then, according to the "accretion tail" model of declining flux before 4.0 Ga ago. This diagram illustrates, at least schematically, how bombardment at $\gtrsim$10$^3$ times the current rate, causes saturation simultaneously at crater diameters >2 km within ~100 Ma, causing explosive growth of kilometers-deep megaregolith during such early conditions.

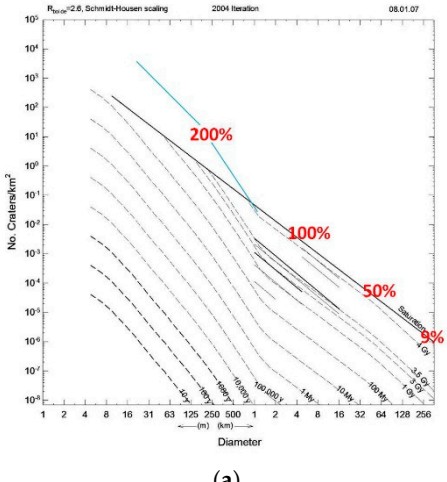
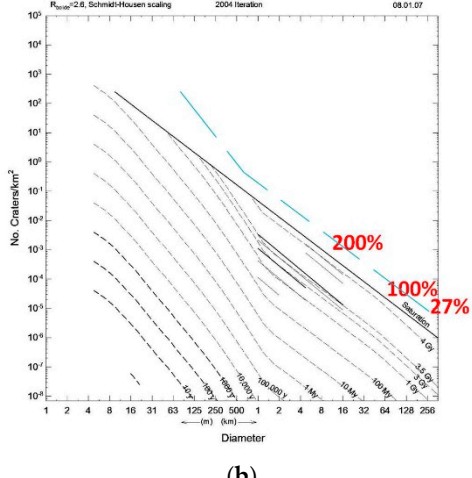

(**a**)                                                       (**b**)

**Figure 15.** This 2004 iteration of the isochron diagram schematically illustrates the method for deriving multikilometer deep regolith on old surfaces. The "%" figures indicate the percentage of the surface covered by craters larger than the indicated diameter. (**a**) This figure shows that for a surface 4 Ga in age (blue line), craters in the largest single bin cover only 9% of the surface, but craters of diameter as large as D ~ 4–8 km cumulatively cover 100% of the surface, and craters of a few hundred meters cover a total of 200% of the surface. (**b**) If impact rates were hundreds of time greater before 4 Ga ago (as discussed in the text), supersaturation of the lunar surface is virtually assured. In a hypothetical case where total cratering (blue dashed line) reaches ~8× the saturation level (plausibly within few hundred Ma on a surface formed >4 Ga ago; see Figure 14), 100% saturation is reached among 100 km-scale craters During supersaturation, *observed* crater counts rarely rise above 2× the saturation line, because of crater overlap, but we can plot the total number of craters formed (dashed line) and study the consequences. If 100 to 200% of the area is covered by craters of D ≳ 10 to 100 km, implying megaregolith depths in the range of at least ~3 km to as much as ~30 km. Deepest layers of megaregolith may be consolidated into coherent breccias.

Figure 16 shows examples of early work on megaregolith evolution as a function of time, according to the "accretion tail" and "megaregolith evolution" hypotheses. Figure 16a shows the assumed impact rate decay curve, and Figure 16b shows calculated curves giving depth of accumulated megaregolith as a function of time on a surface formed at any given time before the present. According to that model, if we imagine a surface formed at 4.3 Ga ago (such as the floor of a pre-Imbrium basin of that age), megaregolith depths of kilometers could easily have accumulated by 4.0 Ga ago. This, in turn, could have pulverized an impact melt lens formed in (or just below) that basin floor, as well as impact melts scattered on the surface in ejecta blankets from early basins, formed before the Imbrium impact scattered 3.9 Ga impact melt ejecta across the Apollo collecting area. According to those curves in Figure 16b, regolith reaches perhaps 10 km depth for a surface formed 4.25 Ga ago, and ~1 km depth for a surface formed ~4.2 Ga ago, roughly100 m depth for a surface formed 4.0–4.1 Ga ago, but only 40–60 m depth for a surface formed 3.9 Ga ago. Obviously, such a 1st-order model could be refined with better lunar data.

The curves in Figure 15 help explain why early crustal igneous rock samples from, say, 4.4 Ga ago can be found in lunar samples but impact melts before 4.0 Ga are rare, and they help explain why only small clasts of impact melts are showing up from large impact events at ~4.21 and ~4.35 Ga ago. For example, if an impact melt lens formed within the top few km under a fresh basin floor created, say, 4.4 Ga ago, then the finite volume of its impact melts would have been pulverized toward the 60-μm range mean particle size observed in the modern lunar regolith, or at best converted to relatively small clasts in breccias. Small clasts are just beginning to be detected and measured, but we note that at such sizes (especially in lunar glass spherules), argon losses may affect the validity of Ar-based radiometric

ages estimates (Zellner and Delano 2015 [176]). Radiometric dating issues increase as we go to small fragment sizes.

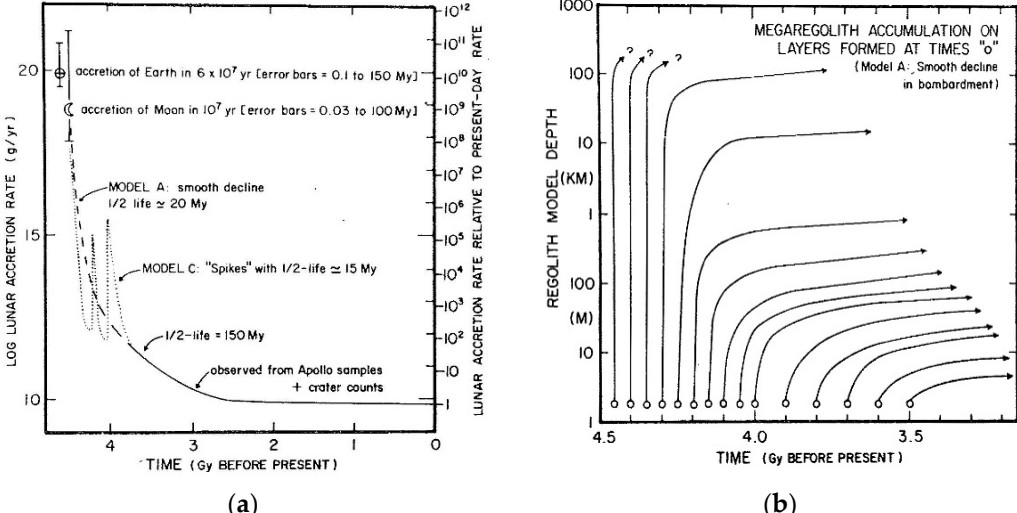

**Figure 16.** (**a**) The 1980 models of declining impact rate since planetary formation used to estimate megaregolith growth in different eras. "Model A" assumes constantly declining flux with assumed rate of increase of half-life for surviving planetesimals. "Model C" allows impact spikes around 4.0 Ga ago. These models contrast with then-prevailing assumption of low impact rate between ~4.4 and 3.9 Ga ago. (**b**) The 1980 model of regolith depth as a function of time, generated on surfaces formed at any given time (horizontal axis). The model assumes the smoothly declining impact rate shown as "Model A" in figure (**a**). This model is based on depths of craters of diameter D, such that the total area of covered by craters of size >D is 100–200% of the area of the Moon (see Figure 15). Under the assumed conditions, regolith would be ~1 km deep on surfaces formed at 4.2 Ga ago, but only about 30 m deep on surfaces formed at 3.9 Ga ago.

In the case of primordial igneous crustal rock, however, a global reservoir exists at the base of the megaregolith. Thus, every crater sufficiently deep to penetrate the megaregolith in a given area ejects primordial igneous rocks over certain areas of the Moon. A recent, large crater such as Copernicus (diameter 93 km, depth 3.8 km, transient excavation depth estimated at 20–30 km), or Eratosthenes (diameter 59 km, depth 3.6 km), or Aristillus (diameter 55 km, depth 3.6 km), may scoop into the coherent primordial crust and eject samples onto the surface. These three examples are especially relevant because they lie in or near the Imbrium basin; here, the Imbrium impact itself may have swept away much of the pre-3.9 Ga megaregolith that had formed during the early intense bombardment before the Imbrium impact, so that the crater's transient cavity more easily tapped into coherent parts of the buried crust.

The above scenario depends on the depth and geometry of what I have called "impact melt lenses" on or beneath the floors of freshly excavated basins. Numerical modeling of the explosive formation of basins (e.g., Artemieva and Shuvalov (2008 [60])) has not clearly resolved the depths and lateral profiles of the impact models. Substructure of the basin itself appears to depend on parameters such as impact velocity and angle. Some models estimate the total volume of impact melt, but uncertainties arise regarding the disposition of the melt relative to the transient cavity, and the fraction of melt that is thrown out into ejecta blankets (Zhu et al. (2017 [177])). Some models suggest a central deep plug of impact melt, surrounded by a thinner sheet of impact melt. In the case of the Imbrium impact, impact melt sheets extending to ~20 km deep have been mentioned.

It's clear that the well-known lunar mare surfaces are not impact melt sheets. This was known in pre-Apollo literature from the fact that most mare surfaces are broken by rims of craters like Archimedes (in Imbrium) which formed after the basin-forming impact but were partially flooded (sometimes

nearly buried) later by the mare lavas. Substantial time intervals between basin formation and mare surface lava emplacement were indicated. Mare basalt sample dates confirmed that the mare lavas postdated the basin-forming impacts.

To summarize, in modest-sized and large basins that formed before ~4.3 Ga ago, impact melt sheets were likely mostly pulverized and/or converted into highland breccia clasts by intense bombardment that ate through the upper kilometer or tens of kilometers. The period ~4.3 to ~3.9 Ga ago may mark a transition period where more basin impact melt fragments begin to survive, depending on the thickness, depth, and age of the initial impact melt lens. Meanwhile, however, recent Copernicus-class craters, depending on the patchy variation in megaregolith depth, may penetrate the megaregolith, ejecting and scattering primordial crustal rocks that exist at the base of global megaregolith.

The net effect of megaregolith evolution is thus to filter and constrain what can be collected on the surface. Rocks lying around on the Moon are not at all the uniform sampling of the whole history of the Moon, as seemed plausible during pre-Apollo planning, discussed in Section 1. An important factor in understanding the surface samples comes from the work of Thompson (1974 [178]), Thompson et al. (1979 [179]), Ghent et al. (2005 [92]), and Basilevsky et al. (2013 [91]). Thompson et al. (1979 [179]) analyzed radar, thermal infrared, and visual photogeologic properties of lunar craters and found that craters of diameter D < 12 km (depth d ≲ 2 km) produce typically rocky boulder fields in lunar mare lava plains, but not in the highlands. But larger craters of D > 12 km and d ≳ 2 km, craters in both maria and highlands, suggested higher boulder density. Also, as discussed in Section 9, Ghent et al. (2005 [92]; 2014 [93]) and Basilevsky et al. (2013 [91]) showed that roughly meter-scale rocks, once ejected onto the lunar surface, disappear within a few hundred million years. All this is supported also by cosmic ray exposure ages typically encountered among lunar samples, in the range of tens of Ma to a few hundred Ma (e.g., Eugster 2003 [67]). Combining these results with the above discussions, we see that the large, recent Copernicus or Tycho scale craters of the last few hundred million years have penetrated through the *currently existing* megaregolith cross sections, excavating and scattering most of the specimens returned by the Apollo and Luna programs. (At each collecting site, smaller impacts scatter shallower local samples.) Similarly, lunar meteorite samples that arrive on modern Earth are from geologically recent impacts. The lunar samples, thus, represent excavations from the *currently existing* substructure, but from globally scattered locations, where the 3.9-ish Ga spike from Imbrium (and Orientale?) may or may not be not dominant. These effects are schematically illustrated and described in Figure 17 and its caption.

In short, all lunar samples represent only the geologically current Moon, where the ages and petrology of samples are controlled by past megaregolith evolution, and the megaregolith evolution and depth varies from one region to another dependent on the stochastics of the few largest few basin-forming impacts (the 1, 2, 3 impacts that represent the extreme low end of the size-distribution curve of lunar craters). The "active geology" of the Moon involves not regional sedimentation, sub-aerial deposition, water, wind, or mountain building, but messy, stochastic impacts.

Why are these findings important? They mean that lunar sample collections do not give us a snapshot of the lunar surface or regolith at >4 Ga ago. The pre-Apollo expectation that rocks picked up on the lunar surface would randomly sample the entire geologic history of the Moon—an expectation lingering as a ghostly influence even in today's literature—was dramatically wrong.

Here we pause to note that these principles predict a different regolith-related history for asteroids than for lunar rocks (Hartmann (2003 [52])). In a given lunar basin-forming impact, most of the lunar impact melt materials that did not escape the Moon fell back into lunar near-surface layers, rarely more than roughly 10–20 km deep, as discussed above. The early basin impact melts and impact-shocked rocks thus had "no place to hide" during the intense early bombardment prior to ~4.1 Ga ago. In contrast, during near-catastrophic asteroid collisional disruptions during that same early period, fragment ejection velocities were frequently slow enough that most of the fragments generated by the impact reassembled and formed massive rubble piles (Michel et al. (2001 [180])), so that at least some impact melts and shocked rocks were sequestered at many tens of kilometer

depths (or more) in the now rubble-pile-like parent bodies. There, they could have been preserved for Ga-scale intervals. However, some of those asteroid-belt bodies were subject to later catastrophic collision in some cases, so that the early impact melt rocks from some parent bodies were now released in smaller rubble piles. Some of those rubble piles, perhaps after still further breakups into small objects, delivered the pre-4.0 Ga-old asteroidal impact melt materials to Earth. To put it in other words, intense bombardment on the Moon destroyed surface layers, but intense bombardment of the belt caused breakup and reassembly of some bodies. This explains why the asteroid signature retains more pre-4.0 Ga impact melts than the Moon, explaining why the time distribution of impact melt ages from different parent bodies varies from one meteorite class to another, and extends over a wider time range than the lunar samples (Swindle et al. (2013 [181])). In this scenario, we note again that the Imbrium impact was late enough that its impact melt lens was preserved, so that the Moon is peppered not only with crustal plutonic samples but also with 3.9-ish Ga impact melts.

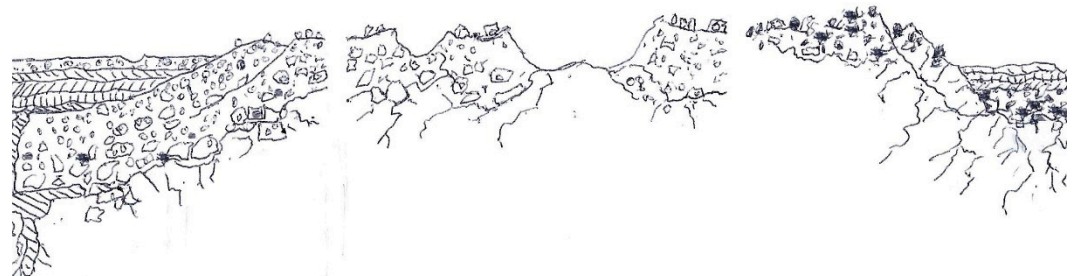

**Figure 17.** Schematic diagram showing aspects of lunar megaregolith evolution, existing on today's Moon. Black indicates impact melts; white objects indicate primordial crustal fragments. Hatched materials are intrusive mare basalts. Note that boulder-sized rock samples, once ejected onto the surface, last for no more than a few hundred Ma (see text). At left is a 4.3 Ga-old basin, where intense early bombardment has rounded the rim profile and, within 100 Ma or so after basin formation, quickly "eaten through" the impact melts lens near the original floor of the basin, after which the basin floor was covered by intrusive and extrusive basalts. Few impact melt specimens can be found from this basin. Middle panel shows upland, where thick regolith has formed. A large young crater has penetrated the megaregolith and ejected onto the surface samples of crustal rocks (but few impact melts except as micro-clasts in the upland breccias). At the right is the Imbrium basin, ~3.9 Ga old. The impact rates were sufficiently lower after 3.9 Ga ago that only a thin megaregolith was created on the floor of the basin, and some impact melts survive there. Impact melts scattered from this region during the Imbrium-forming impact and by more recent moderate-sized impacts have maintained 3.9 Ga-old Imbrium impact melt presence (as well as primordial crustal rock samples) on the lunar surface in much of the region sampled by Apollo and Luna landings.

## 19. Conclusions: Epistemology of a Paradigm, and Future Directions

In a spirit of collegial reflection, this section addresses not only current scientific issues of the first few Ga of the Solar System, but also the epistemological issues of why the cataclysm/LHB paradigm persisted for more than four decades in spite of questions and contrary evidence. Our current research situation involves the following points, derived from the above discussion.

- Pre-Apollo mission expectations that lunar rocks would represent the entire history of the Moon and Solar System were overly optimistic, and created a subtle, influential bias in the interpretations of post-Apollo lunar samples. By 1973–1990, these interpretations led to the paradigm that a brief, global, bombardment catastrophe occurred at ~3.9 Ga and created most of the present-day lunar multi-ring basins.

- It has been accepted from 1960s crater count evidence that the *average* lunar impact rate in pre-mare time ($\gtrsim$3.6 Ga ago) was *at least* $10^2$ higher than today. The major questions now are about the time dependence throughout that period.

- If the lunar highlands are supersaturated with impact craters, then the *average* cratering rate before 3.6–3.9 Ga ago was significantly higher than $10^2$ times the present rate.

- Comparison of Apollo rock sample ages with crater densities at various lunar landing sites by independent investigators indicates a sharply declining impact rate from ~3.8 Ga ago to ~3 Ga ago, followed by a more nearly constant or slowly declining rate. This is the observed tail end of *either* an LHB spike or a longer-term decline in pre-3.9 Ga cratering.

- The "classic" terminal cataclysm model, as defined here, derives primarily from work of Tera et al. (1973 [2]; 1974 [3,4]), Turner and Cadogan (1973 [5]), Turner et al. (1973 [6]), and Ryder (1990 [11]). For 30–40 years, the name "terminal cataclysm" has referred to a sudden, short-lived spike in impact rate, centered around 3.9 Ga ago, and lasting for about 150–200 Ma, during which most lunar multi-ring basins formed—with few impacts between ~4.45 and 4.0 Ga ago.

- The term "Late Heavy Bombardment," later "LHB," was introduced as early as a 1975 paper by the dynamicist, George Wetherill. Wetherill referenced this to the concept of Tera et al. (1974 [3]), but considered two "senses" of the term involving "storage" of planetesimals: (1) storage preventing impacts until ~3.9 Ga ago, or (2) storage resulting in leakage into a slowly declining impact rate until ~3.9 Ga ago.

- It seems fair to suggest that in the euphoria over the first "ground truth" lunar samples, there was too-willing acceptance of detailed interpretations of radiometric ages that were only 1, 2, or 3 error bars apart as representing different basin ages.

- Some Apollo-era igneous samples dating from ~4.5–~4.4 Ga were associated with primordial crustal history but were fewer than expected by Apollo mission planners. No impact melt samples have been found from the period (see Figure 4).

- Critiques of the emerging terminal cataclysm paradigm began as early as the 1970s, mostly from workers outside the lunar sample and dynamical communities—but were not very effective.

- Tera et al. (1973 [2]) suggested the Imbrium impact as a possible explanation of their terminal cataclysm and maintained this idea (with decreasing emphasis?) in later papers, referring also to some sort of global geochemical event. The connection with Imbrium, briefly but unsuccessfully resurrected by Haskin et al. (1998 [62]), appears to be re-resurrected in the current discussions of lunar history.

- Ryder (1990 [11]), with valuable data, showed that Apollo lunar impact melt rock samples exhibit a large, 150-Ma spike at about 3.9 Ga ago with few impact melt samples before that (see Figure 5). This paper also introduced an influential "Ryder's rule," that lack of impact melts around 4.5–3.9 Ga ago implies few impacts in that period. These concepts were seen as confirming the terminal cataclysm paradigm, and explicitly or implicitly affected (or even constrained) interpretations of lunar history for the next two decades. These ideas now appear to be incorrect.

- Data on lunar impact melt clast ages in KREEP-poor lunar meteorites (Cohen et al. (2000 [69]; 2002 [70]; 2005 [71])) show no anomalous Ryder-like spike at ~3.9 Ga ago.

- Data on *asteroidal* meteorites' impact-related samples also show no anomalous, Ryder-like spike at ~3.9 Ga ago, but rather a gentle "swell" from ~4.2 to ~3.5 Ga go, usually peaking around 3.7 Ga ago. This profile varies from one meteorite class to another (Swindle et al. (2013 [181])). These results may be a signature of survival processes that are different among various asteroids than on the Moon.

- Accumulating reports suggest impact melt clasts and other possibly impact-related materials from before 4.0 Ga, with radiometric ages clustering around ~4.21 and ~4.33 Ga ago. These have led to suggestions of basin-scale impacts at those times, contrary to the terminal cataclysm/LHB paradigm.

- The combination of current lunar and asteroidal data, from radiometric, dynamical, cratering, and petrologic sources suggests that the "classic" terminal cataclysm/LHB spike, accepted for ~45 years, never occurred. This conclusion has been reached independently in several recent review articles.

- The ~3.9 Ga-old Apollo impact melt spike, if interpreted as coming from Imbrium, implies that the Imbrium-scale impacts' ejecta blankets affect larger areas of the lunar surface than originally expected, and is consistent with some petrological variety in Imbrium ejecta blanket materials.

- Apollo/Luna samples have been used to suggest ages of 3.93 to 3.72 Ga for at least five of the largest, lunar front-side, multi-ring basins (see Table 1). This interpretation requires a cataclysmic bombardment around 3.9 Ga, but this conflicts with data from lunar meteorite impact melt clasts and asteroidal meteorites. The sample ages typically lie within 1–3 error bars of the Imbrium age, and may simply associate with Imbrium ejecta, in which case the assigned ages of basins are incorrect.

- The dating of most basins, and the Ryder's rule assertion that virtually no basin forming impacts happened before ~4.0 Ga ago, is called into question by reports of clusters of impact melts from ~4.35 Ga ago and circa 4.2 Ga ago.

- Crater count studies of impact rates and crater density ratios among basins such as Serenitatis, Nectaris, and Imbrium are inconsistent with published radiometric-based assertions that Serenitatis formed after Nectaris or (in a rare implication) even after Imbrium.

- The 2005 Nice model was defended as explaining the cataclysmic spike in bombardment at 3.9 Ga, but this is contradicted by the absence of such a spike in lunar and asteroidal meteorites.

- The dating of a spike at 3.9 Ga in dynamical models was not constrained by physics of the models, but (as properly stated in the models) was assumed to be 3.9 Ga ago.

- Dynamical models attempting to match the terminal cataclysm paradigm have evolved on a yearly timescale, producing varied and rapidly revised results, with the cataclysmic spike replaced by increasingly gradual surges. In successive models (2010–2012), the proposed durations of these surges have increased from 210 Ma to 1600 Ma. A more recent model (Morbidelli et al. (2018 [160])) allows for an early intense impact flux decreasing from lunar formation until 3.9 Ga ago, and continuing to decrease after that, as per the observational data.

- These same models continue to apply the term "Late Heavy Bombardment" ("LHB") to their results, even though the results are inconsistent with original definitions of these terms (used until 2010).

- In the view of this paper, the use of the terms "LHB" and "terminal cataclysm" should be restricted to their original definitions (which are inconsistent with observations).

- There is currently wide agreement that gravitational and resonant effects produced migration of Jupiter and other giant planets, which in turn may have scattered various classes of bodies from various sources in the Solar System. Giant planet migration seems supported by observations of extra-solar planetary systems with giant planets very close to their stars. The relation of these events to the 3.9 Ga-ago era is uncertain.

- The numbers of outer Solar System bodies, scattered by such planet migrations, are uncertain. Some may have been involved in creation of low-albedo captured satellites. The intensity and timing of any resultant inner Solar System bombardment remains highly uncertain.

- Most of the collectible rocks on the lunar surface today were placed there by impact excavation within the last several hundreds of Ma, and hence represent the subsurface structure of the *present-day* Moon.

- Megaregolith evolution has fragmented and mixed subsurface layers. As a result, existing sample collections are not intact lunar history collections, but are filtered by megaregolith-production processes.

- The Orientale basin structure offers clues to various impact basin formation processes. For example, (1) the GRAIL discovery of linear segments in the Oceanus Procellarum buried ring structure does not rule out impact as the origin of that huge feature (as argued by Andrews-Hanna et al. (2014 [154])) since the Orientale rings show similar linear segments, and (2) mare lavas along the bases of several of the Orientale ring scarps suggest igneous extrusions during basin formation,

so that some ancient "crustal" igneous rocks (in addition to impact melts) may date from specific multi-ring basin-forming events.

- The process of lunar megaregolith formation explains the paradox of paucity of lunar impact melts from before ~4.0–~4.1 Ga ago, but presence of primordial, crustal plutonic igneous rocks. Impact melts concentrate in quickly destroyed near-surface layers, but ancient crustal rocks are always available to be excavated from the base of the megaregolith.

- Asteroidal materials experienced megaregolith sequestration processes different from those on the Moon. Stochastic, catastrophic disruption of asteroidal parent bodies created large rubble- pile asteroids where early asteroid impact melts could be "stored" (to use the Wetherill term) at great depths until release by geologically more recent disruptive impacts. This explains differences in age distributions not only between lunar and asteroidal meteorites, but also among different petrologic classes of meteorites. These phenomena deserve more work.

- Enstatite-like planetesimals, originating near 1 AU and having Earth-like isotope ratios, may have been scattered into the inner edge of the belt during planetary formation, as first proposed by Wetherill (1977 [8,9]). They may, thus, have formed a fairly massive, extended "E belt" as pictured in the Morbidelli et al. (2001 [106]; 2018 [161]) and Bottke et al. (2011 [153]; 2012 [140]) dynamical models. One large, local, ~1 AU enstatite planetesimal may have impacted Earth, resulting in a lunar formation scenario that avoids the "isotope crisis," i.e., explaining isotope similarity between Earth and Moon. Some of these same bodies may have been prominent in the declining impact flux during the lunar basin-forming period from ~4.4 to ~3.8 Ga ago, explaining suggestions of enstatite-like chemistry. Enstatite-like planetesimals formed in Earth's zone deserve more work.

- Dynamical models may remain in an immature state until better empirical evidence is available. Most essential is more reliable dating of samples that will define the formation ages of several key lunar multi-ring basins. Examples: Serenitatis (difficult to find outcrops, but key to constraining the age of morphologically "old"-looking basins), Orientale (easy to find basin-dating outcrops in well-defined ring scarp faces; measurably different from Imbrium?), Nectaris (Altai scarp and proposed impact-melt pools may allow dating?). Excluded is South Pole-Aitken basin (contrary to current wisdom). Yes, it formed very early, but for that very reason it is difficult to guarantee an easy age measurement, because of smaller basin-scale impacts on the floor of the earlier large basin.

- Although the "classic" terminal cataclysm is now being abandoned by the planetary science community, other communities have a problem because the term and concept continue to appear as constraints in various other fields, for example in biology, where it has been used recently as a constraint on the origin of life at ~3.9 Ga.

- Consistent with the study of Kuhn (1962 [1]), paradigms should not be viewed as edifices of perfect, known science, but more as home ports that provide the maps and equipment for the launching of new efforts that may uncover new results.

- The most fruitful use of numerical models comes not as providing "answers" to problems, but by varying the individual uncertain parameters to constrain plausibility of various ideas.

- The history of the terminal cataclysm/LHB paradigm provides important lessons in science. First, a tendency has been visible to interpret new observational data and new theoretical models as constrained by the paradigm, rather than seeing the paradigm as constrained by the observations and models. Paradigms must fit data, not the other way 'round.

- The history of the terminal cataclysm/LHB paradigm also points to a troublesome separation of various communities in planetary science, defined by graduate student training rather than underlying natural phenomena. For example, studies of radiometric dating, dynamical models, cratering chronometry, planetary mapping, and petrologic results are often presented in different conferences or different (often overlapping) sessions. For example, we have the Lunar and Planetary Science Conference, Division of Planetary Sciences of the American Astronomical Society, Meteoritical Society, European Geophysical Union, American Geophysical Union, American Geological Society, etc. This system tends to concentrate work into different communities,

preventing interplay of data sets. In the author's view, recent, smaller, topical conferences have often offered more fruitful, interdisciplinary progress on significant problems (but perhaps less fruitful networking for freshly minted Ph.D.s).

• In spite of our problems as scientists and writers of scientific papers, the evolution of the terminal cataclysm/LHB paradigm offers an important example of the strength of the post-Renaissance, scientific method at work, including cooperative progress from different sub-disciplines, some wrong turns, and few-decades-timescale of self-correction, all based on empiricism. This contrasts with systems of thought such as various legal, religious, and some political/ideological systems that rely on an advocacy-basis, where different participants defend literally pre-scribed views.

**Funding:** This research received no external funding.

**Acknowledgments:** The writer gratefully thanks the International Space Science Institute in Bern, Switzerland, for visiting scientist invitations in 2015, 2016, 2017, and 2018, during which I pursued this study. At ISSI, thanks to Johannes Geiss, Rafael Rodrigo, Roger Bonnett, and Rudi von Steiger for helpful discussions on content and publication. Thanks also to numerous colleagues, including Alessandro Morbidelli, Bill Bottke, Marc Norman, Barbara Cohen, Patrick Michel, Tim Swindle, Dave Kring, Natalia Artemieva, Larry Nyquist, Heather Meyer, Hal Levison, Audrey Bouvier, Donald Lowe, Clark Chapman, and Richard Binzel for collegial correspondence, discussions, and comments during presentations about these issues (even when we did not always agree). Thanks to Caleb Fassett and two other unnamed professional reviewers for thoughtful and conscientious review of this long manuscript for *Geosciences*, and to special issue co-editor Nicolle E. B. Zellner for detailed editorial comments. Special thanks to Caleb Fassett for pointing out that the term "megaregolith" can be traced at least as far back as Doug Nash (1971, two years prior to my 1973 usage). Special thanks also to Elaine Owens at the Planetary Science Institute for editorial and formatting assistance, and help in locating sometimes-obscure historical references, and to Gayle Hartmann for always reliable editorial assistance. Many more references could have been mentioned in addition to the examples I have cited here; I apologize to authors that I have left out. An enormous number of international researchers have cooperated with each other, utilizing the scientific method of comparing, testing, and interpreting observational results, in order to work our way toward an understanding of lunar samples and the primordial impact record throughout the Solar System. Such an understanding helps us understand our place in the universe. Civilization owes all these researchers a great, still under-appreciated, debt.

**Conflicts of Interest:** The author declares no conflict of interest. Permissions for use of the historic figures in this paper have been obtained from original publishers or authors of the material.

## Appendix A. References

This paper was originally written with references in the style "Author (Year)" in alphabetical order. However, the reference style of this journal, Geosciences, is listing of references by number in order of appearnce. The author, the editors of this lunar "issue" of Geosciences, and the publisher MDPI, agreed that in a paper about evolution of ideas, such as this, the text should include the authors and dates of various statements, as well as the number alone. The bibliography of 182 references is thus presented below in both styles below for the benefit of readers.

Andrews-Hanna, J.C.; Besserer, J.; Head, J.; Howett, C.; Kiefer, W.; Lucey, P.; McGovern, P.; Melosh, H.; Neumann, G.; Phillips, R.; Smith, D.; Solomon, S.; Zuber, M. Structure and Evolution of the Lunar Procellarum Regio as Revealed by GRAIL Gravity Data. *Nature* **2014**, *514*, 68–71.

Artemieva, N.; Shuvalov, V. Numerical Simulation of High-velocity Impact Ejecta Following Falls of Comets and Asteroids onto the Moon. *Solar Sys. Res.* **2008**, *42*, 329–334.

Baldwin, R. B. *The Face of the Moon*; University of Chicago Press: Chicago, IL, USA, **1949**.

Baldwin, R. B. Ancient Giant Craters and the Age of the Lunar Surface. *Astronom. J.* **1969a**, *14*, 570–571.

Baldwin, R. B. Absolute Ages of the Lunar Maria and Large Craters. *Icarus* **1969b**, *11*, 320–331.

Baldwin, R. B. Was There a Terminal Cataclysm 3.9–4.0 × 10$^9$ years ago? *Icarus* **1974**, *23*, 157–166.

Basilevsky, A. T.; Head, J. W.; Hörz, F. Survival Times of Meter-sized Boulders on the Surface of the Moon. *Planet. Space Science* **2013**, *89*, 118–126.

Beals, C. S.; Innes, M. J. S.; Rottenberg, J. A. Fossil Meteorite craters. In *The Moon, Meteorites, and Comets*; Middlehurst, B. M., Kuiper, G. P., Eds.; University of Chicago Press: Chicago, IL, USA, **1963**; pp. 235–284.

Becker, H.; Horan, M.; Walker, R; Gao, S.; Lorand, J.-P.; Rudnik, R. Highly Siderophile Element Composition of the Earth's Primitive Upper Mantle: Constraints from New Data on Peridotite Massifs and Xenoliths. *Geochim. Cosmochim. Acta* **2006**, *70*, 4528–4550.

Boehnke, P.; Harrison, T. M. Illusory Late Heavy Bombardments. In Proceedings of the National Academy of Science, Los Angeles, CA, USA, **2016**; *113(39)*, 10802–10806.

Bogard, D. Impact Ages of Meteorites: A Synthesis. *Meteoritics* **1995**, *30*, 244–268.

Bottke, W. F.; Norman, M. D. The Late Heavy Bombardment. *Ann. Rev. Earth Planet. Science* **2017**, *49*, 619–647.

Bottke, W. F.; Vokrouhlický, D.; Nesvorný, D.; Minton, D.; Morbidelli, A.; Brasser, R. The E-Belt: A Possible Missing Link in the Late Heavy Bombardment. 41st Lunar and Planetary Science Conference, Houston, TX, USA, 1–5 March **2010**; LPI Contribution No. 1533, p. 1269.

Bottke, W. F.; Vokrouhlický, D.; Minton, D.; Nesvorný, D.; Morbidelli, A.; Brasser, R.; Simonson, B. The Great Archean Bombardment, or the Late Late Heavy Bombardment. 42nd Lunar and Planetary Science Conference, Houston, TX, USA, 7–11 March **2011**; LPI Contribution No. 1608, p. 2591.

Bottke, W. F.; Vokrouhlický, D.; Minton, D.; Nesvorný, D.; Morbidelli, A.; Brasser, R.; Simonson, B.; Levison, H. An Archean Heavy Bombardment from a Destabilized Extension of the Asteroid Belt. *Nature* **2012**, *485*, 78–81.

Cameron, A. G. W.; Ward, W. The Origin of the Moon. 7th Lunar and Planetary Science Conference, The Woodlands, TX, USA, **1976**; p. 120.

Clark, B. E.; Bus, S. J.; Rivkin, A. S.; McConnochie, T.; Sanders, J.; Shah, S.; Hiroi, T.; Shepard, M. E-type Asteroid Spectroscopy and Compositional Modeling. *J. Geophys. Res.* **2004**, *109(E2)*, CiteID E02001.

Cohen, B. The Vestan Cataclysm: Impact-Melt Clasts in Howardites and the Bombardment History of 4 Vesta. *Meteorit. Planet. Science* **2013**, *48*, 771–785.

Cohen, B. A.; Swindle, T. D.; Kring, D. A. Support for the Lunar Cataclysm Hypothesis from Lunar Meteorite Impact Melt Ages. *Science* **2000** *290*, 1754–1756.

Cohen, B. A.; Swindle, T. D.; Taylor, L. A.; Nazarov, M. A. $^{40}$Ar-$^{39}$Ar Ages from Impact Melt Clasts in Lunar Meteorites Dhofar 025 and Dhofar 026. 33$^{rd}$ Lunar and Planetary Science Conference, Houston, TX, USA, 11–15 March **2002**; Abstract No. 1252.

Cohen, B. A.; Swindle, T. D.; Kring, D. A. Geochemistry and $^{40}$Ar-$^{39}$Ar Geochronology of Impact-melt Clasts in Feldspathic Lunar Meteorites: Implications for Lunar Bombardment History. *Meteorit. Planet. Science* **2005**, *40*, 755–777.

Cruikshank, D. P.; Hartmann, W. K.; Tholen, D. J. Colour, Albedo and Nucleus Size of Halley's Comet. *Nature* **1985**, *315*, 122–124.

Cruikshank, D. P.; Tholen, D. J.; Hartmann, W. K.; Bell, J. F.; Brown, R. H. Three Basaltic Earth-approaching Asteroids and the Source of Basaltic Meteorites. *Icarus* **1991**, *89*, 1–13.

Ćuk, M. Chronology and Sources of Lunar Impact Bombardment. *Icarus* **2012**, *218*, 69–79.

Ćuk, M.; Gladman, B. J.; Stewart, S. T. Constraints on the Source of Lunar Cataclysm Impactors. *Icarus* **2010**, *207*, 590–594.

Denevi, B. The New Moon. *Physics Today* **2017**, *70(6)*, 39–44.

Deutsch, R.A.; Stöffler, D. Rb-Si Analysis of Apollo 16 Melt Rocks and a New Age Estimate for the Imbrium Basin Chronology and the Early Heavy Bombardment of the Moon. *Geochim. Cosmochim. Acta* **1987**, *51*, 1951–1964.

Eugster, O. Cosmic-ray Exposure Ages of Meteorites and Lunar Rocks and Their Significance. *Chemie der Erde Geochem.* **2003**, *63*, 3–30.

Fassett, C. I.; Minton, D. A. Impact Bombardment of the Terrestrial Planets and the Early History of the Solar System. *Nature Geosci.* **2013**, *6*, 520–524.

Fassett, C. I.; Head, J.; Kadish, S.; Mazarico, E.; Neumann, G.; Smith, D.; Zuber, M. Lunar Impact Basins: Stratigraphy, Sequences, and Ages from Superposed Impact Crater Populations Measured from Lunar Orbiter Laser Altimeter (LOLA) Data. *J. Geophys. Res.* **2012**, *117*, Cite ID E00H06.

Fernandes, V.; Artemieva, N. Impact Ejecta Temperature Profile on the Moon – What Are the Effects on the Ar-Ar Dating Method? 43rd Lunar and Planetary Science Conference, The Woodlands, TX, USA, 19–23 March **2012**; Abstract No. 1659.

Fernandes, V.A. Garrick-Bethel, I., Shuster, D.L.; Weiss, B. Common 4.2 Ga Impact Age in Samples from Apollo 16 and 17. In Workshop on Early Solar System Bombardment, Houston, scheduled Nov. 19–26, 2008 [129] (meeting delayed until 2009 due to hurricane). Program book published **2008**; abstract #3028.

Fernandes, V. A.; Fritz, J.; Weiss, B. P., Garric-Bethell, I.; Shuster, D. L. The Bombardment History of the Moon as Recorded by 40Ar-39Ar Chronology. Meteoritics and Planetary Science, **2013**, 48, 1–29.

Fielder, G. *Structure of the Moon's Surface*; Pergamon Press: London, UK, **1961**.

Fischer-Gödde, M.; Becker, H. What Is the Age of the Nectaris Basin? New Re-Os Constraints for a Pre-4.0 Ga Bombardment History of the Moon. 42nd Lunar and Planetary Science Conference, The Woodlands, TX, USA, 7–11 March **2011**; LPI Contribution No. 1608, p. 1414.

Fritz, J.; Bitsch, B.; Kührt, E.; Morbidelli, A.; Tornow, C.; Wünnerman, K.; Fernandes, V. A.; Grenfell, J. L.; Rauer, H.; Wagner, R.; Werner, S. C. Earth-like Habitats in Planetary Systems. *Planet. Space Science* **2014**, *98*, 254–267.

Ghent, R. R.; Leverington, D. W.; Campbell, B. A.; Hawke, B. R.; Campbell, D. B. Earth-based Observations of Radar-dark Crater Haloes on the Moon: Implications for Regolith Properties. *J. Geophys. Res.* **2005**, *110(E2)*, CiteIDE02005.

Ghent, R. R.; Hayne, P. O.; Bandfield, J. L.; Campbell, B. A.; Carter, L. M.; Allen, C. C.; Paige, D. A. Constraints on the Recent Rate of Lunar Ejecta Breakdown. *Geology* **2014**, *42*, 1059–1062

Gladman, B. J.; Burns, J. A.; Duncan, M. J.; Levison, H. F. The Dynamical Evolution of Lunar Impact Ejecta. *Icarus* **1995**, *118*, 302–321.

Gomes, R.; Levison, H. F.; Tsiganis, K.; Morbidelli, A. Origin of the Cataclysmic Late Heavy Bombardment Period of the Terrestrial Planets. *Nature* **2005**, *435*, 466–469.

Gradie, J. C.; Tedesco, E. F. Compositional Structure of the Asteroid Belt. *Science* **1982**, *216*, 1405–1407.

Gradie, J. C.; Chapman, C. R.; Tedesco, E. F. Distribution of Taxonomic Classes and the Compositional Structure of the Asteroid Belt. In *Asteroids II*; Binzel, R. P., Gehrels, T., Matthews, M. S., Eds.; University of Arizona Press: Tucson, AZ, USA, **1989**; pp. 316–335.

Grange, M. L.; Nemchin, A. A.; Pidgeon, R. T.; Merle, R. E.; Timms, N. E. What Lunar Zircon Ages Can Tell? 44th Lunar and Planetary Science Conference, The Woodlands, TX, USA, 18–22 March **2013**; LPI Contribution No. 1719, p. 1884.

Grieve, R. A. F.; Cintala, M. J. A Method for Estimating the Initial Impact Conditions of Terrestrial Cratering Events, Exemplified by its Application to Brent Crater, Ontario. In Proceedings of the 12th Lunar and Planetary Science Conference, Houston, USA, 16–20 March 1981; Lunar and Planetary Institute, Houston, TX, USA, **1981**; pp. 1607–1621.

Grinspoon, D. H. Large Impact Events and Atmospheric Evolution on the Terrestrial Planets. Ph.D. Thesis, University of Arizona, Tucson, AZ, USA; **1989**.

Hartmann, W. K. Terrestrial and Lunar Flux of Large Meteorites in the Last Two Billion Years. *Icarus* **1965**, *4*, 157–165.

Hartmann, W. K. Early Lunar Cratering. *Icarus* **1966**, *5*, 406–418.

Hartmann, W. K. Terrestrial, Lunar, and Interplanetary Rock Fragmentation. *Icarus* **1969**, *10*, 201–213.

Hartmann, W. K. Preliminary Note on Lunar Cratering Rates and Absolute Time Scales. *Icarus* **1970a**, *12*, 131–133.

Hartmann, W. K. Lunar Cratering Chronology. *Icarus* **1970b**, *13*, 299–301.

Hartmann, W. K. Paleocratering of the Moon: Review of Post-Apollo Data. *Astrophys. Space Science* **1972**, *12*, 48–64.

Hartmann, W. K. Ancient Lunar Mega-Regolith and Subsurface Structure. *Icarus* **1973**, *18*, 634–636.

Hartmann, W. K. Lunar Cataclysm: A misconception? *Icarus* **1975**, *24*, 181–187.

Hartmann, W. K. Dropping Stones in Magma Oceans: Effects of Early Lunar Cratering. In Proceedings of the Conference on the Lunar Highlands Crust, Houston, TX, USA, 14–16 November **1979**; Pergamon Press: New York, NY, USA, **1980**; pp. 155–171.

Hartmann, W. K. Does Crater "Saturation Equilibrium" Exist in the Solar System? *Icarus* **1984**, *60*, 56–74.

Hartmann, W.K. Moon Origin: The Impact Trigger Hypothesis. In *Origin of the Moon*; Hartmann, W. K., Phillips, R. J., Taylor, G. J., Eds.; Lunar and Planetary Institute: Houston, TX, USA, **1986**; pp. 579–608.

Hartmann, W. K. A Satellite-Asteroid Mystery and a Possible Early Flux of Scattered C-class Asteroids. *Icarus* **1987**, *71*, 57–68.

Hartmann, W.K. Additional Evidence about an Early Intense Flux of C Asteroids and the Origin of Phobos. *Icarus* **1990**, *87*, 236–240.

Hartmann, W. K. Megaregolith Evolution and Cratering Cataclysm Models B Lunar Cataclysm as a Misconception (28 Years Later). *Meteor. Planet. Science* **2003**, *38*, 579–593.

Hartmann, W. K. Empirical Studies of Lunar Bombardment around 3.6-4 Gy Ago. Workshop on the Early Solar System Bombardment II, Houston, TX, USA, 1–3 February **2012**; LPI Contribution 1649, pp. 28–29.

Hartmann, W. K. The Giant Impact Hypothesis: Past, Present, (and Future?). *Phil. Trans. Royal Soc. A*, **2014**, *372(2024)*, DOI: 10.1098/rsta.2013.0249.

Hartmann, W. K. Reviewing "Terminal Cataclysm": What Does it Mean? Workshop on Early Solar System Impact Bombardment III, Houston, TX, USA, 3–4 February **2015**; LPI Contribution No. 1826, p. 3003.

Hartmann, W. K.; D[61aubar, I. J. Martian Cratering 11. Utilizing Decameter Scale Crater Populations to Study Martian History. *Meteor. Planet. Science* **2017**, *52*, 493–510.

Hartmann, W. K.; Davis, D. R. Satellite-sized Planetesimals. IAU Colloquium on Planetary Satellites, Ithaca, NY, USA, 18–21 August **1974**; Abstract.

Hartmann, W. K.; Davis, D. R. Satellite-sized Planetesimals and Lunar Origin. *Icarus* **1975**, *24*, 504–515.

Hartmann, W. K.; Gaskell, R. Planetary Cratering 2: Studies of Saturation Equilibrium. *Meteor. Planet Science* **1997**, *32*, 109–121.

Hartmann, W. K.; Kuiper, G. P. Concentric Structures Surrounding Lunar Basins. *Comm. Lunar Planet Lab.* **1962**, *1*, 51–66.

Hartmann W. K.; Wood, C. A. Origin and Evolution of Multi-ring Basins. *The Moon* **1971**, *3*, 3–78.

Hartmann, W. K.; Strom, R.; Weidenschilling, S.; Blasius, K.; Woronow, A.; Dence, M.; Grieve, R.; Diaz, J.; Chapman, C.; Shoemaker, E.; Jones, K. Chronology of Planetary Volcanism by Comparative Studies of Planetary Craters. In *Basaltic Volcanism on the Terrestrial Planets* (Basaltic Volcanism Study Project); Pergamon Press: Elmsford, NY, USA, **1981**; pp. 1050–1127.

Hartmann, W. K.; Cruikshank, D. P.; Degewij, J. Remote Comets and Related Bodies: VJHK Colorimetry and Surface Materials. *Icarus* **1982**, *52*, 377–407.

Hartmann, W. K.; Ryder, G.; Dones, L.; Grinspoon, D. The Time-Dependent Intense Bombardment of the Primordial Earth/Moon System. In *Origin of the Earth and Moon*; Canup, R. M., Righter, K., Eds.; University of Arizona Press: Tucson, AZ, USA, **2000**; pp. 493–512.

Hartmann, W. K.; Quantin, C.; Mangold, N. Possible Long-term Decline in Impact Rates 2: Lunar Impact-melt Data Regarding Impact History. *Icarus* **2007**, *186*, 11–23.

Hartung, J. B. Can Random Impacts Cause the Observed $^{39}$Ar/$^{40}$Ar Age Distribution for Lunar Highland Rocks? *Meteoritics* **1974**, *9*, 349.

Haskin, L. A.; Korotev, R. L.; Rockow, K. M.; Joliff, B. L. The Case for an Imbrium Origin of the Apollo Thorium-rich Impact-melt Breccias. *Meteor. Planet. Science* **1998**, *33*, 959–975.

Head, J. W. Serenitatis Multi-ringed Basin: Regional Geology and Basin Ring Interpretation. *Moon and the Planets* **1979**, *21*, 439–462.

Herwartz, D.; Pack. A.; Friedrichs, B.; Bischoff, A. Identification of the Giant Impactor Theia in Lunar Rocks. *Science* **2014**, *344*, 1146–1150.

Hiesinger, H.; Jaumann, R.; Neukum, G.; Head, J. W. Ages of Mare Basalts on the Lunar Nearside. *J. Geophys. Res.* **2000**, *105(E12)*, 29,239–29,275.

Hiesinger, H.; Head, J.; Wolf, U.; Jaumann, R.; Neukum, G. Ages and Stratigraphy of Mare Basalts in Oceanus Procellarum, Mare Nubium, Mare Cognitum, and Mare Insularum. *J. Geophys. Res.* **2003**, *108(E7)*, Cite ID 5065.

James, O. B. Petrologic and Age Relations of the Apollo 16 Rocks – Implications for Subsurface Geology and the Age of the Nectaris Basin. In Proceedings of the 12th Lunar and Planetary Science Conference, Houston, TX, USA, **1982**; pp. 209–233.

Jessberger, E. K.; Kirsten, T.; Staudacher T. One Rock and Many Ages – Further K-Ar Data on Consortium Breccia 83215. Proceedings of 8th Lunar and Planetary Science Conference, Houston, USA, 14–18 March 1977; Lunar and Planetary Institute, Houston, TX, USA, **1977**; pp. 2567–2580.

Joy, K. H.; Zolensky, M.; Ross, D.; McKay, D.; Kring, D. Direct Detection of Projectile Relics on the Moon. Workshop on the Early Solar System Bombardment II, Houston, TX, USA, 1–3 February **2012a**; LPI Contribution No. 1649, pp. 34–35.

Joy, K.; Nagashima, K.; Huss, G.; Ross, D.; McCoy, D.; Kring, D. Direct Detection of Projectile Relics from the End of the Lunar Basin-forming Epoch. *Science* **2012b**, *336*, 1426–1429.

Kahn, A.; Pommier, A.; Neumann, G. A.; Mosegaard, K. The Lunar Moho and the Internal Structure of the Moon: a Geophysical Perspective. *Tectonophysics* **2013**, *609*, 331–352.

Koeberl, C. The Record of Impact Processes on the Early Earth: a Review of the First 2.5 Billion Years. In *Processes on the Early Earth*; Reimold, W. J., Gibson, R. L., Eds.; Geological Society of America: Washington, DC, USA, **2006**; DOI: 10.1130/2006.2405(01).

Koeberl, C.; Anderson, R. R. Memorial: Jack B. Hartung, March 10, 1937-August 28, 2015. *Meteor. Planet. Science* **2015**, 50, 2137–2139.

Kring, D. A.; Cohen, B. A. Cataclysmic Bombardment Throughout the Inner Solar System 3.9–4.0 Ga. *J. Geophys. Res.* **2002**, *107(E2)*, CiteID 5009.

Kuhn, T. S. *The Structure of Scientific Revolutions*. University of Chicago Press: Chicago, IL, USA; **1962**.

Levison, H. F.; Dones, L.; Chapman, C. R.; Stern, S. A.; Duncan, M. J.; Zahnle, K. Could the Lunar Late Heavy Bombardment Have Been Triggered by the Formation of Uranus and Neptune? *Icarus* **2001**, *151*, 286–306.

Li, Y.; Hsu, W. Multiple Impact Events on the L-Chondrite Parent Body: Insight from SIMS U-Pb Dating of Ca-Phosphates in the NWA 7251 L-Melt Breccia. *Meteor. Planet. Science* **2018**, *53*, 1081–1096.

Lineweaver, C. H. Crater-counting Evidence Against the Late Heavy Bombardment Hypothesis. Astrobiology Science Conference 2010: Evolution and Life: Surviving Catastrophes and Extremes on Earth and Beyond, League City, TX, USA, 20–26 April **2010**; LPI Contribution No. 1538, p. 5226.

Liu, T.; Michael, G.; Engelmann, J.; Wünnemann, K.; Oberst, J. Regolith Mixing by Impacts: Lateral Diffusion of Basin Melt, *Icarus* **2019**, *321*, 691–704.

Lowe, D. R.; Byerly, G. R. The Terrestrial Record of Late Heavy Bombardment. *New Astronomy Reviews* **2018**, *81*, 39–61.

Lunar and Planetary Institute, Universities Space Research Association, Lunar Sample Overview, Apollo 12 Mission, https://www.lpi.usra.edu/lunar/missions/apollo/apollo_12/samples/ (accessed on 13 May 2019).

Maher, K. A.; Stevenson, D. J. Impact Frustration of the Origin of Life. *Nature* **1988**, *331*, 612–614.

Mann, A. Bashing Holes in the Tale of Earth's Trouble Youth {Cataclysm's End]. *Nature* **2018**, *553*, 393–395.

Marchi, S.; Bottke, W.; Kring, D.; Morbidelli, A. The Onset of the Lunar Cataclysm as Recorded in its Ancient Crater Populations. *Earth Planet. Sci. Lett.* **2012**, *325–6*, 27–38.

Marchi, S.; Bottke, W.; Elkins-Tanton, L.; Bierhaus, M.; Wuennemann K.; Morbidelli, A.; Kring, D. Widespread Mixing and Burial of Earth's Hadean Crust by Asteroid Impacts. *Nature* **2014**, *511*, 578-582.

Martellato, E.; Vivaldi, V.; Massironi, M.; Cremonese, G.; Marzari, F.; Ninfo, A.; Haruyama, J. Is the Linné Impact Crater Morphology Influenced by the Rheological Layering on the Moon's Surface? Insights from Numerical Modeling. *Meteor. Planet. Science* **2017**, *52*, 1388–1411.

McGetchin T. R.; Settle, M.; Head, J. W. Radial Thickness Variation in Impact Crater Ejecta: Implications for Lunar Basin Deposits. *Earth Planet. Science Lett.* **1973**, *20*, 226–236.

Melosh, H. J. *Impact Cratering: A Geologic Process*; Oxford University Press: New York, NY, USA, **1989**.

Melosh, H. J. An Isotopic Crisis for the Giant Impact Origin of the Moon? 72nd Annual Meeting of the Meteoritical Society, Nancy, France, 13–18 July **2009**; p. 5104.

Merle, R. E; Nemchin, A.; Grange, M.; Whitehouse, M. Stratigraphy of the Fra Mauro Formation Defined by U-Pb Zircon Ages of Breccia Samples from Apollo 14 Landing Site. 44th Lunar and Planetary Science Conference, The Woodlands, TX, USA, 18–22 March **2013**; LPI Contribution No. 1719, p. 1833.

Meyer, H. H.; Robinson, M. S.; Denevi, B. W.; Boyd, A. K. A New Global Map of Light Plains from the Lunar Reconnaissance Orbiter Camera. 49th Lunar and Planetary Science Conference, The Woodlands, TX, USA, 19–23 March **2018**; LPI Contribution No. 2083, id.1474.

Michael, G.; Basilevsky, A.; Neukum, G. On the History of the Early Meteorite Bombardment of the Moon: Was There a Terminal Lunar Cataclysm? *Icarus* **2018**, *302*, 80–103.

Michel, P.; Benz, W.; Tanga, P.; Richardson, D. C. Collisions and Gravitational Reaccumulation: Forming Asteroid Families and Satellites. *Science* **2001**, *294*, 1696–1700.

Morbidelli, A., Petit, J.-M.; Gladman, B.; Chambers, J. A Plausible Cause of the Late Heavy Bombardment. *Meteor. Planet. Science* **2001**, *36*, 371–380.

Morbidelli, A.; Levison, H. F.; Tsiganis, K.; Gomes, R. Chaotic Capture of Jupiter's Trojan Asteroids in the Early Solar System. *Nature* **2005**, *435*, 462–465.

Morbidelli, A.; Marchi, S.; Bottke, W. F.; Kring, D. A. A Sawtooth-like Timeline for the First Billion Years of Lunar Bombardment. *Earth Planet. Science Lett.* **2012**, *355*, 144–151.

Morbidelli, A.; Nesvorny, D.; Laurenz, V.; Marchi, S.; Rubie, D.; Elkins-Tanton, L.; Wieczorek, M.; Jacobson, S. The Timeline of the Lunar Bombardment: Revisited. *Icarus* **2018**, *305*, 262–276.

Nakamura, Y. Timing Problem with the Lunar Module Impact Data as Recorded by LSPE and Corrected Near-surface Structure at the Apollo 17 Landing Site. *J. Geophys. Res. Planets* **2011**, *116(E12)*, CiteID E12005.

NASA Johnson Space Center Curatorial Document. Available online: http://curator.jsc.nasa.gov/lunar/lsc/15415.pdf (accessed on 10 May 2019).

Nash, D. E.; Conel, J. E.; Fanale, F. P. Objectives and Requirements of Unmanned Rover Exploration of the Moon. *The Moon* **1971**, *3*, 221–230.

Nemchin, A. A.; Pidgeon, R. T.; Whitehouse, M. J.; Vaughan, J. P.; Meyer, C. SIMS U-Pb Study of Zircon from Apollo 14 and 17 Breccias: Implications for the Evolution of Lunar KREEP. *Geochim. Cosmochim. Acta* **2008**, *72(2)*, 668–689.

Neukum, G. Meteoritenbombardement und Datierung Planetarer Oberflächen. Habilitation Dissertation for Faculty Membership, Ludwig-Maximilians University of Munich, Germany; **1983**.

Neukum, G.; Ivanov, B. Crater Size Distributions and Impact Probabilities from Lunar, Terrestrial-planet, and Asteroid Cratering Data. In *Hazards Due to Comets and Asteroids*; Gehrels, T., Ed.; University of Arizona Press: Tucson, AZ, USA, **1995**; pp. 359–416.

Neukum, G.; Ivanov, B. A.; Hartmann, W. K., Cratering Records in the Inner Solar System in Relation to the Lunar Reference System. In *Chronology and Evolution of Mars*; Kallenbach, R.; Geiss, J.; Hartmann, W. K.; International Space Science Institute: Bern, Switzerland, **2001**; (also published in *Space Science Rev.* **2001**, *96*, 55–86).

Norman, M. D. The Lunar Cataclysm: Reality or "Mythconception?" *Elements* **2009**, *5*, 23–38.

Norman, M.; Nemchin, A. Large Impacts at 4.2 Ga from Uranium-Lead Dating of Lunar Melt Breccia. Workshop on Early Solar System Impact Bombardment II, Houston, TX, USA, 1–3 February **2012**; LPI Contribution No. 1649, pp. 59–60.

Norman, M. D.; Nemchin, A. A. A 4.2 Billion Year Old Impact Basin on the Moon: U-Pb Dating of Zirconolite and Apatite in Lunar Melt Rock 67955. *Earth Planet. Science Lett.* **2014**, *388*, 387–398.

Norman, M. D.; Bennett, V. C.; Ryder, G. Targeting the Impactors: Siderophile Element Signatures of Lunar Impact Melts from Serenitatis. *Earth Planet. Sci. Lett.* **2002**, *202(2)*, 217–228.

Norman, M. D.; Shih, C.-Y.; Nyquist, L. E.; Bogard, D. D.; Taylor, L. A. Early Impacts on the Moon: Crystallization Ages of Apollo 16 Melt Breccias. 38th Lunar and Planetary Science Conference, League City, TX, USA, 12–16 March **2007**; LPI Contribution No. 1338, p. 1991.

Norman, M. D.; Taylor, L.; Shih, C.-Y.; Nyquist, L. Crystal Accumulation in a 4.2 Ga Lunar Impact Melt. *Geochim. Cosmochim. Acta* **2015**, *172*, 410–429.

Nunes, P.D.; Tatsumoto, M.; Unruh, D. M. U-Th-Pb Systematics of Some Apollo 17 Lunar Samples, and Implications for a Lunar Basin Excavation Chronology. In Proceedings of the 5th Lunar Science Conference, Houston, TX, USA, 18–22 March **1974**; pp. 1487–1514.

Perkins, S. From Hell on Earth, Life's Building Blocks. *Science* **2014**, *346(6215)*, 1279.

Petro, N. E.; Pieters, C. M. Modeling the Provenance of the Apollo 16 Regolith. *J. Geophys. Res.* **2006**, *111(E9)*, CiteID E09005.

Pidgeon, R. T.; Nemchin, A. A.; Grange, M. L.; Meyer, C. Evidence for a "Lunar Cataclysm" at 4.34 Ga from Zircon U Pb Systems. 41st Lunar and Planetary Science Conference, The Woodlands, TX, USA, 1–5 March **2010**; LPI Contribution No. 1533, p. 1126.

Podosek, F. A.; Huneke, J.; Gancarz, A.; Wasserburg, G. The Age and Petrography of Two Luna 20 Fragments and Inferences for Widespread Lunar Metamorphism. *Geochim. Cosmochim. Acta* **1973**, *37*, 423–425.

Quantin, C.; Allemand, P.; Mangold, N.; Delacourt, C. Ages of Valles Marineris (Mars) Landslides and Implications for Canyon History. *Icarus* **2004**, *172*, 555–572.

Quantin, C.; Mangold, N.; Hartmann, W. K.; Allemand, P. Possible Long-term Decline in Impact Rates 1: Martian Geological Data. *Icarus* **2007**, *186*, 1–10.

Robbins, S. J. Revised Lunar Cratering Chronology for Planetary Geological Histories. 44th Lunar and Planetary Science Conference, The Woodlands, TX, USA, 18–22 March **2013**; LPI Contribution No. 1719, p. 1619.

Ryder, G. Lunar Samples, Lunar Accretion and the Early Bombardment of the Moon. *Eos* **1990**, *71*, 313, 322–323.

Ryder, G. Mass Flux in the Ancient Earth-Moon System and Benign Implications for the Origin of Life on Earth. *J. Geophys. Res.* **2002**, *107(E4)*, CiteID 5022.

Ryder, G.; Spudis, P. D. Chemical Composition and Origin of Apollo 15 Impact Melts. In Proceedings of the 17th Lunar and Planetary Science Conference, Houston, TX, USA; **1987**; pp. E432–E446.

Safronov, V. Evolution of the Protoplanetary Cloud and Formation of the Earth and the Planets. Translated NASA Technical Document 677, **1972** (originally published by Nauka Press: Moscow, USSR, 1969).

Schaeffer, G. A.; Schaeffer, O. A. $^{39}$Ar-$^{40}$Ar Ages of Lunar Rocks. In Proceedings of the 8th Lunar Science Conference, Houston, TX, USA; **1977**; pp. 2253–2300.

Schmitz, B.; Häggström, T.; Tassinari, M. Sediment-Dispersed Extraterrestrial Chromite Traces a Major Asteroid Disruption Event. *Science* **2003**, *300*, 961–964.

Schultz, P. H.; Crawford, D. A. Origin and Implications of Non-radial Imbrium Sculpture on the Moon. *Nature* **2016**, *535*, 391–394.

Shoemaker, E. M. Interpretation of Lunar Craters. In *Physics and Astronomy of the Moon*; Kopal, Z., Ed.; Academic Press: New York, NY, USA, **1962**; pp. 283–359.

Sleep, N. H.; Zahnle, K., Kasting, J.F.; Morowitz, H. J. Annihilation of Ecosystems by Large Asteroid Impacts on the early Earth. *Nature* **1989**, *342*, 139–142.

Spray, J. G. Lithification Mechanisms for Planetary Regoliths: The Glue that Binds. *Ann. Rev. Earth Planet. Science* **2016**, 44, 139–174.

Spudis, P. D.; Wilhelms, D. E.; Robinson, M. S. The Sculptured Hills of the Taurus Highlands: Implications for the Relative Age of Serenitatis, Basin Chronologies and the Cratering History of the Moon. *J. Geophys. Res.* **2011**, *116*, CiteID E00H03.

Staudacher, T.; Dominik, B.; Jessberger, E. K.; Kirsten, T. Consortium Breccia 73255: $^{40}$Ar-$^{39}$Ar Dating. 9th Lunar and Planetary Science Conference, Houston, TX, USA, **1978**; pp. 1098–1100.

Staüdermann, F. J.; Heusser, E.; Jessberger, E.; Lingner, S. New $^{40}$Ar-$^{39}$Ar Ages of Apollo 14 Rocks. *Geochim. Cosmochim. Acta* **1991**, *55*, 2339–2349.

Stöffler, D.; Ryder, G. Stratigraphy and Isotope Ages of Lunar Geologic Units: Chronological Standard for the Inner Solar System. In *Chronology and Evolution of Mars*; Kallenbach, R., Geiss, J., Hartmann, W. K., Eds.; Kluwer Academic Publishers: Dordrecht, Netherlands, **2001**; pp. 105–164.

Stöffler, D.; Bischoff, A.; Borchardt, R.; Burghele, A.; Deutsch, A.; Jessberger, E. K.; Ostertag, R.; Palme, H.; Spettel, B.; Reimold, W. U.; Wacker, K.; Wänke, H. Composition and Evolution of the Lunar Crust in the Descartes Highlands, Apollo 16. *J. Geophys. Res.* **1985**, *90*, C449–C506.

Stöffler, D.; Ryder, G.; Ivanov, B. A.; Artemieva, N. A.; Cintala, M. J.; Grieve, R. A. F. Cratering History and Lunar Chronology. *Rev. Mineral. Geochem.* **2006**, *60*, 519–596.

Strom, R. G.; Marchi, S.; Malhotra, R. Ceres and the Terrestrial Planets Impact Cratering Record. *Icarus* **2018**, *302*, 104–108.

Stuart-Alexander, D. E.; Howard, K. A. Lunar Maria and Circular Basins B A Review. *Icarus* **1970**, *12*, 440–456.

Swann, G. A.; Bailey, N.; Batson, R.; Eggleton, R.; Hait, M.; Holt, H.; Larson, K.; Reed, V.; Schaber, G.; Trask, N.; Ulrich, G.; Wilshire, H. Geology of the Apollo 14 Landing Site in the Fra Mauro Highlands. U.S.G.S. Geological Survey Professional Survey Paper 880; U. S. Government Printing Office, Washington, D. C., USA, **1977**.

Swindle, T.; Kring, D. Was There a Concentration of Lunar and Asteroidal Impacts at ~4000 Ma? Workshop on Early Solar System Impact Bombardment III, Houston, TX, USA, 4–5 February **2015**; LPI Contribution No. 1826, p. 3030.

Swindle, T. D.; Kring, D. A.; Weirich, J. R. $^{40}$Ar/$^{39}$Ar Ages of Impacts Involving Ordinary Chondrite Meteorites. In *Advances in $^{40}$Ar/$^{39}$Ar Dating*; Jourdan, F., Mark, D. F., Verati, C., Eds.; Geological Society: London, UK, **2013**, pp. 333–348.

Taylor, S. R. *Planetary Science: A Lunar Perspective*; Lunar and Planetary Institute: Houston, TX, USA, **1982**.

Taylor, S. R. *Solar System Evolution: A New Perspective*; Cambridge University Press: Cambridge, UK, **1992**.

Taylor, G. J.; Warren, P.; Ryder, G.; Delano, J.; Pieters, C.; Lofgren, G. Lunar Rocks. In *Lunar Sourcebook*; Heiken, G. H., Vaniman, D. T., French, B., Eds.; Cambridge University Press: Cambridge, MA, USA, **1991**; pp. 183–284.

Tera F.; Papanastassiou, D. A.; Wasserburg, G. J. A Lunar Cataclysm at ~3.9 AE and the Structure of Lunar Crust. 4th Lunar and Planetary Science Conference, Houston, TX, USA, **1973**; p. 723.

Tera, F.; Papanastassiou, D. A.; Wasserburg, G. J. Isotopic Evidence for a Terminal Lunar Cataclysm. *Earth Planet. Science Lett.* **1974a**, *22*, 1–21.

Tera, F.; Papanastassiou, D. A.; Wasserburg, G. J. The Lunar Time Scale and a Summary of Isotopic Evidence for a Terminal Lunar Cataclysm. 5th Lunar and Planetary Science Conference, Houston, TX, USA, **1974b**; p. 792.

Thiessen, F.; Nemchin, A. A.; Snape, J. F.; Whitehouse, M. J.; Bellucci, J. J. Impact History of the Apollo 17 Landing Site Revealed by U-Pb SIMS Ages. *Meteor. Planet. Science* **2017**, *52*, 584–611.

Tholen, D. J.; Barucci, M. A. Asteroid Taxonomy. In *Asteroids II*; Binzel, R. P., Gehrels, T., Matthews, M. S., Eds.; University of Arizona Press: Tucson, AZ, USA, **1989**; pp. 298–315.

Thompson, T. W. Atlas of Lunar Radar Maps at 70-cm Wavelength. *The Moon* **1974**, *10*, 51B 85.

Thompson, T. W.; Roberts, W. J.; Hartmann, W. K.; Shorthill, R. W.; Zisk, S. H. Blocky Craters: Implications about the Lunar Megaregolith. *Moon and the Planets* **1979**, *21*, 319–342.

Trieloff, M.; Korochantseva, E.; Buikin, A.; Schwarz, W.; Hopp, J.; Lorentz, C.; Jessberger, E. L Chondrite Asteroid Breakup tied to Ordovician Meteorite Shower by Multiple Isochron 40Ar-39Ar Dating. 1st International Conference on Impact Cratering in the Solar System, Zurich, Switzerland, 6-11 August **2006**; Abstract.

Tsiganis, K.; Gomes, R.; Morbidelli, A.; Levison, H. Origin of the Orbital Architecture of the Giant Planets of the Solar System. *Nature* **2005**, *435*, 459–461.

Turner, G. The Early Chronology of the Moon: Evidence for the Early Collisional History of the Solar System. *Phil. Trans. Royal Soc. A, Math. Phys. Science* **1977**, *285(1327)* 97–103.

Turner, G.; Cadogan, P. H. $^{40}$Ar-$^{39}$Ar Chronology of Chondrites. *Meteoritics* **1973**, *8*, 447–448.

Turner, G.; Cadogan, P. H. The History of Lunar Bombardment Inferred from $^{40}$Ar-$^{39}$Ar Dating of Highland Rocks. In Proceedings of the 6th Lunar Science Conference, Houston, TX, USA, 17–21 March **1975**; pp. 1509–1538.

Turner, G.; Cadogan, P. H., Yonge, C. J. Argon Selenochronology. In Proceedings of the 4th Lunar Science Conference, Houston, TX, USA, **1973**; p. 1889.

Urey, H. C. *The Planets: Their Origin and Development*; Yale University Press: New Haven, CT, USA, **1952**.

Walsh, K. J.; Morbidelli, A.; Raymond, S. N.; O'Brien, D. P.; Mandell, A. M. A Low Mass for Mars from Jupiter's Early Gas-driven Migration. *Nature* **2011**, *475*, 206–209.

Warren, P. Early Lunar Cratering and a Reality Check on Efforts to Directly Age-date Lunar Magma Ocean Crystallization. Meteoritical Society Meeting, Workshop on the First Ga of Impact Records: Evidence from Luna Samples and Meteorites, Berkeley, CA, USA, 25–26 July **2015**; LPI Contribution No. 1884, p. 6022.

Warren, P. H.; Haack, H.; Rasmussen, K. L. Megaregolith Insulation and the Duration of Cooling to Isotopic Closure within Differentiated Asteroids and the Moon. *J. Geophys. Res.* **1991**, *96*, 5909–5923.

Wetherill, G. W. Late Heavy Bombardment of the Moon and Terrestrial Planets. 6th Lunar and Planetary Science Conference, Houston, TX, USA, **1975**; pp. 866–868.

Wetherill, G. W. Evolution of the Earth's Planetesimal Swarm Subsequent to the Formation of the Earth and Moon. 8th Lunar and Planetary Science Conference, Houston, TX, USA, **1977a**; Abstract 1005–1007.

Wetherill, G. W. Evolution of the Earth's Planetesimal Swarm Subsequent to the Formation of the Earth and Moon. In Proceedings of the 8th Lunar and Planetary Science Conference, Houston, TX, USA, **1977b**; pp. 1–16.

Whitaker, E.; Kuiper, G. P.; Hartmann, W. K.; Spradley, L. H. *Rectified Lunar Atlas*; University of Arizona Press with the Lunar and Planetary Laboratory: Tucson, AZ, USA, **1964**.

Wieczorek, M. A. and 15 others. The Crust of the Moon as Seen by GRAIL. *Science* **2013**, *339*, 671–674.

Wilhelms, D. E. The Geologic History of the Moon. U.S. Geol. Surv. Prof. Pap. 1348, **1987**.

Wilkening, L. L. Meteorites in Meteorites: Evidence for Mixing among the Asteroids. In *Comets, Asteroids and Meteorites*; Delsemme, A. H., Ed.; University of Toledo Press: Toledo, OH, USA, **1977**; pp. 389–396.

Zellner, N. E. B. Cataclysm No More: New Views on the Timing and Delivery of Lunar Impactors. *Orig. Life Evol. Biosph.* **2017**, *47*, 261–280.

Zellner, N. E. B.; Delano, J. W. $^{40}$Ar/$^{39}$Ar Ages of Lunar Impact Glasses: Relationships among Ar Diffusivity, Chemical Composition, Shape, and Size. *Geochim. Cosmochim. Acta* **2015**, *161*, 203–218.

Zhu, M.-H.; Wünnemann, K.; Artemieva, N. Effects of Moon's Thermal State on the Impact Basin Ejecta Distribution. *Geophys. Res. Lett.* **2017**, *22*, 11,292–11,300.

Zuber, M. T.; and 16 others. Gravity Field of the Moon from the Gravity Recovery and Interior Laboratory (GRAIL) Mission. *Science* **2013**, *339(6120)*, 668–671.

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
