# Peer review of "History of the Terminal Cataclysm Paradigm: Epistemology of a Planetary Bombardment That Never (?) Happened"

_geosciences, doi:10.3390/geosciences9070285_

Round 1
Reviewer 1 Report
see file attached

Reviewer 2 Report
Comments are in attached file

Reviewer 3 Report
This manuscript presents a very interesting review of the history of the theory of the Late Heavy Bombardment. The author, who is a specialist and has been involved in that story almost since the beginning, tells us the different aspects that spoke for or against the theory of the LHB, and how the definition of the LHB evolved, since the end of World War II. This manuscript could be a (large) book chapter. The structure itself helps the reader to follow the chronology and the different ideas and studies, which built this paradigm. This very dense and exhaustive review is entertaining as well, as the author gives many anecdotes. He actually tells us a scientific story, which is, albeit very long, very pleasant to read and very instructive. I enthusiastically recommend publication. I detected anyway many minor problems of copy-editing, which I list below:
1) in many places in the paper, strange characters appear, mostly as a C. I guess this is a problem of encoding. For instance lines 30, 32, 48 ("terminal cataclysm" starts with an A and finishes with a @), 63, 150, 402, 510, 598, 642, 662, 798, 904, 909, 980, 986, 996, 1061, 1077, 1150, 1165, 1166, 1189, 1201, 1264 (twice a strange character between the number and the uncertainty), 1439, 1775, 1781, 1784, 1812, 1849, 1917, 2294
2) the figures are usually of poor quality, e.g. Fig.2, Fig.5, Fig.13, Fig.14, Fig. 15
3) l.173: enesis -> Genesis
4) l.196: "and are taken" -> "are taken"
5) Fig.6: I guess the caption contains notes for the author, which he should remove
6) l.502: "a few Fa few tens of Ma"
7) l.883: the bullet should be removed
8) caption of Fig.9: numbers are missing
9) l.1091: a closing parenthesis without opening
10) l.1511: add a space between 401 and Gyr
11) l.1627: "the that"
12) 2 bullets to remove, l. 1859 and 1882
13) the Fig.14 has two captions
14) Caption of Fig.15: "The right diagram a case"
Round 2
Reviewer 1 Report
Ok
Author Response
We don't have a record of a Round 2 from the first Reviewer, but see email from Nicolle that responses to all reviews have been received.